


# Variability and stability of anthropogenic CO2 in Antarctic Bottom Waters observed in the Indian sector of the Southern Ocean, 1978-2018

Léo Mahieu[1], Claire Lo Monaco[2], Nicolas Metzl[2], Jonathan Fin[2], Claude Mignon[2]

[1]Ocean Sciences, School of Environmental Sciences, 4 Brownlow Street, Liverpool L69 3GP, UK
[2]LOCEAN-IPSL, CNRS, Sorbonne Université, Paris, France

*Correspondence to*: Léo Mahieu (Leo.Mahieu@live.fr); Claire Lo Monaco (claire.lomonaco@locean.upmc.fr)

**Abstract**

Antarctic bottom waters (AABWs) are known as a long term sink for anthropogenic $CO_2$ ($C_{ant}$) but is hardly quantified because of the scarcity of the observations, specifically at an interannual scale. We present in this manuscript an original dataset combining 40 years of carbonate system observations in the Indian sector of the Southern Ocean (Enderby Basin) to evaluate and interpret the interannual variability of $C_{ant}$ in the AABW. This investigation is based on regular observations collected at the same location (63° E/56.5° S) in the frame of the French observatory OISO from 1998 to 2018 extended by GEOSECS and INDIGO observations (1978, 1985 and 1987).

At this location the main sources of AABW sampled is the fresh and younger Cape Darnley bottom water (CDBW) and the Weddell Sea deep water (WSDW). Our calculations reveal that $C_{ant}$ concentrations increased significantly in AABW, from about +7 µmol.kg$^{-1}$ in 1978-1987 to +13 µmol.kg$^{-1}$ in 2010-2018. This is comparable to previous estimates in other SO basins, with the exception of bottom waters close to their formation sites where $C_{ant}$ concentrations are about twice as large. Our analysis shows that the $C_T$ and $C_{ant}$ increasing rates in AABW are about the same over the period 1978-2018, and we conclude that the long-term change in $C_T$ is mainly due to the uptake of anthropogenic $CO_2$ in the different formation regions. This is however modulated by significant interannual to pluriannual variability associated with variations in hydrological (Θ, S) and biogeochemical ($C_T$, $A_T$, $O_2$) properties. A surprising result is the apparent stability of $C_{ant}$ concentrations in recent years despite the increase in $C_T$ and the gradual acceleration of atmospheric $CO_2$.

The $C_{ant}$ sequestration by AABWs is more variable than expected and depends on a complex combination of physical, chemical and biological processes at the formation sites and during the transit of the different AABWs. The interannual variability at play in AABW needs to be carefully considered on the extrapolated estimation of $C_{ant}$ sequestration based on sparse observations over several years.

## 1 Introduction

$CO_2$ atmospheric concentration has been increasing since the start of the industrialization (Keeling and Whorf, 2000). This increase leads to an ocean uptake of about a quarter of $C_{ant}$ emissions (Le Quéré et al., 2018; Gruber et al., 2019a). It is widely acknowledged that the Southern Ocean (SO) is responsible for 40 % of the $C_{ant}$ ocean sequestration (Gruber et al., 2009; Khatiwala et al., 2009; Matear, 2001; McNeil et al., 2003; Orr et al., 2001). Ocean $C_{ant}$ uptake and sequestration have the benefit to limit the atmospheric $CO_2$ increase but also result in a



gradual decrease of the ocean pH (Gattuso and Hansson, 2011; Jiang et al., 2019). Understanding the oceanic $C_{ant}$
sequestration and its variability is of major importance to predict future atmospheric $CO_2$ concentrations, impact
on the climate and impact of the pH change on marine ecosystems (de Baar, 1992; Orr et al., 2005; Ridgwell and
Zeebe, 2005).
$C_{ant}$ in seawater cannot be measured directly and the evaluation of the relatively small $C_{ant}$ signal from the total
inorganic dissolved carbon ($C_T$; around 3 %; Pardo et al, 2014) is still a challenge to overcome. Different
approaches have been developed in the last 40 years to quantify $C_{ant}$ concentrations in the oceans. The 'historical'
back calculation method based on $C_T$ measurement and preformed inorganic carbon estimate ($C^0$) was
independently published by Brewer (1978) and Chen and Millero (1979). This method has been often applied at
regional and basin scale (Chen, 1982, 1993; Goyet et al., 1998; Körtzinger et al., 1998, 1999; Poisson and Chen,
1987; Lo Monaco et al., 2005a). More recently the TrOCA method (Tracer combining Oxygen, dissolved Carbon
and total Alkalinity) has been developed (Touratier and Goyet, 2004a, 2004b; Touratier et al., 2007) and applied
in various regions including the SO (e.g. Lo Monaco et al., 2005b; Sandrini et al., 2007; Pardo et al., 2014; Roden
et al., 2016; Shadwick et al., 2014; Van Heuven et al., 2011; Kerr et al., 2018). Comparisons with other data-based
methods show significant differences in $C_{ant}$ concentrations, especially at high latitudes and more particularly in
deep and bottom waters (Lo Monaco et al., 2005b; Vázquez-Rodríguez et al., 2009; Pardo et al., 2014). Thus, there
is a need to better explore the $C_T$ and $C_{ant}$ temporal variability in the deep ocean, especially in the SO where
observations are relatively sparse.
Antarctic bottom waters (AABWs) are of specific interest for the atmospheric $CO_2$ and heat regulation as they
play a major role in the meridional overturning circulation (Johnson et al., 2008; Marshall and Speer, 2012) .
AABWs represent a large volume of water by covering the majority of the bottom world ocean (Mantyla and Reid,
1995), and their spreading in the interior ocean through circulation and water mixing (Siegenthaler and Sarmiento,
1993) is a key mechanism for the long-term sequestration of $C_{ant}$ and climate regulation. The AABW formation is
a specific process occurring in few locations around the Antarctic continent (Orsi et al., 1999). In short, the AABW
formation occurs when the Dense Shelf Water (DSW) flows down along the continental shelf. The DSW density
required for this process to happen is reached by the increase in salinity (S) due to brine release from the ice
formation and by a decrease in temperature due to heat loss to either the ice-shelf or the atmosphere. Importantly,
AABW formation process is enhanced by katabatic winds that open areas free of ice called polynyas (Williams et
al., 2007). Indeed, katabatic winds are responsible for an intense cooling that enhance the formation of ice
constantly pushed away by the wind, leading to cold and salty surface waters in contact with the atmosphere. The
variable conditions of wind, ice production, surface water cooling and continental slope shape encountered around
the Antarctic continent lead to different types of AABW, hence the AABW characteristics can be used to identify
their formation sites.
The ability of AABW to accumulate $C_{ant}$ has been controversial since one can believe that the ice coverage limits
the invasion of $C_{ant}$ in Antarctic surface waters (e.g. Poisson and Chen, 1987). This is however not the case in
polynyas, and several studies have reported significant $C_{ant}$ signals in AABW formation regions, likely due to the
uptake of $CO_2$ induced by high primary production (Roden et al., 2016; Sandrini et al., 2007; Shadwick et al.,
2014; van Heuven et al., 2011, 2014). However, little is known about the variability and evolution of $CO_2$ fluxes
in AABW formation regions, and since biological and physical processes are strongly impacted by seasonal and
interannual climatic variations (Fukamachi et al., 2000; Gordon et al., 2010, 2015; Gruber et al., 2019b; McKee et



al., 2011), the amount of $C_{ant}$ stored in the AABWs may be very variable, which could bias the estimates of $C_{ant}$
trends derived from data sets collected several years apart (e.g. Williams et al., 2015; Pardo et al., 2017; Murata et
al., 2019).
In this context of potentially high variability in $C_{ant}$ uptake at AABW formation sites, as well as in AABW export,
circulation and mixing, we used repeated observations collected in the Indian sector of the Southern Ocean to
explore the variability in $C_{ant}$ and $C_T$ in AABW and evaluate their evolution over the last 40 years.
**2 Studied area**
**2.1 AABW samples during the last 40 years**
Most of the data used in this study were obtained in the frame of the long-term observational project OISO (Ocean
Indien Service d'Observations) conducted since 1998 onboard the R.S.V. Marion-Dufresne (IPEV/TAAF). During
these cruises, several stations are visited, but only one station is sampled down to the bottom (4800 m) south of
the Polar Front at 63.0° E and 56.5° S (hereafter noted OISO-ST11). This station is located in the Enderby Basin
on the Western side of the Kerguelen Plateau (Fig. 1) and coincides with the station 75 of the INDIGO-III cruise
(1987). In our analysis, we also included data from the station 14 of the INDIGO-I cruise (1985) and the station
430 of the GEOSECS cruise (1978) located near OISO-ST11 site (405 km and 465 km respectively). All the re-
occupations used in this analysis are listed in Table 1.
Table 1.
**2.2 AABWs circulation in the Atlantic and Indian sectors of the Southern Ocean**
The circulation in the SO is mainly governed by the ACC that flows Eastward, while the Coastal Antarctic Current
(CAC) flows Westward (Fig. 1). The ACC and the CAC influence the circulation of the entire water column,
including the AABWs. The main AABW formation sites are the Weddell Sea where are produced deep and bottom
waters (WSDW and WSBW, respectively; Gordon, 2001; Gordon et al., 2010), the Ross Sea (RSBW; Gordon et
al., 2015, 2009), the Adelie Land coast (ALBW; Williams et al., 2008, 2010) and the Cape Darnley Polynya
(CDBW; Ohshima et al., 2013). AABW formation in the Prydz Bay has also been observed (Rodehacke et al.,
2007; Yabuki et al., 2006) from three polynyas and two ice shelves flowing into the Prydz Channel (Williams et
al., 2016) and mixing with the CDBW. The CDBW and Prydz Bay bottom waters (hereafter called CDBW)
represent a significant AABW export (13 % of all AABWs exports; Ohshima et al., 2013).
The largest bottom water source of the global ocean is the Weddell Sea (Gordon et al., 2001). The exported WSDW
is a mixture of the Warm Deep Water (WDW) and the WSBW. The WSDW in the ACC and Weddell Gyre mixes
with the Circumpolar Deep Water (CDW). A part of the WSDW deflecting southward with the ACC in the Enderby
Basin reaches the Princess Elizabeth Trough (PET) region, East of the Kerguelen Plateau, where it mixes with
other types of AABWs (Orsi et al., 1999).
In the PET sector, the CAC transports a mixture of RSBW and ALBW and accelerates Northward along the Eastern
side of the Kerguelen Plateau (Mantyla and Reid, 1995; Fukamachi et al., 2010). Part of the ALBW-RSBW mixture
also reaches the Western side of the Kerguelen Plateau (Orsi et al., 1999; Van Wijk and Rintoul, 2014) and mix
with the CDBW. The mixture of CDBW and ALBW-RSBW either flows Westward with the CAC and dilutes with





the CDW (Meijers et al., 2010) or flow Northward (Ohshima et al., 2013) and mix with the older WSDW before
reaching the location of our time-series station in the eastern Enderby Basin.
Figure 1.
**2.3 AABW definition**
Nowadays, the distinction of water masses is usually performed according to neutral density ($\gamma^n$) layers. In the SO,
CDW and AABW properties are generally well defined in the range 28.15-28.27 kg.m$^{-3}$ and 28.27-bottom
respectively (Orsi et al., 1999; Murata et al 2019). However, to interpret the long-term variability of the properties
in the AABWs at our location, we prefer to adjust the AABW layer in a narrow band, and select the samples for
$\gamma^n$ > 28.35 kg.m$^{-3}$ (range starting at 4200m to 4600m depending on the year, see Fig. 3). $\gamma^n$ > 28.35 kg.m$^{-3}$
corresponds to the AABW characteristics observed at higher latitudes in the Indian SO sector (Roden et al., 2016).
**2.4 AABW composition at OISO-ST11**
At each formation site, AABWs experienced significant temporal property changes, mostly recognized at decadal
scale (e.g. freshening in the South Indian Ocean, Menezes et al., 2017) with potential impact on carbon uptake and
$C_{ant}$ concentrations during AABW formation (Shadwick et al., 2013). The Θ-S diagram constructed from yearly
averaged data in bottom waters (Fig. 2) shows that the AABW at OISO-ST11 is a complex mixture of WSDW,
CDBW, RSBW and ALBW. The coldest type of AABW was observed at the GEOSECS station at 60° S (-0.56
°C), probably because it experienced less mixing with CDW compared to the warmer type of AABW observed at
the INDIGO-1 station at 53° S (-0.44 °C). For the other cruises and years, Θ in AABW ranges from -0.51 to -0.45
°C with no clear indication on the specific AABW origin. The S range observed in the bottom waters at OISO-
ST11 (34.65-34.67), illustrates either changes in mixing with various AABW sources or temporal variations at the
formation site. Given the knowledge of deep and bottom waters circulation and characteristics (Fig.s 1 and 2) and
the significant $C_{ant}$ concentrations that we estimated at depth (Fig. 3), the main contribution at our location is likely
the younger and colder CDBW for which relatively high $C_{ant}$ concentrations have been recently documented
(Roden et al., 2016). From its formation region, the CDBW can either flow westward with the CAC or flow
northward in the Enderby Basin (Ohshima et al., 2013, Fig. 1). In the CAC branch, the CDBW mixes with the
CDW along the Antarctic shelf and the continental slope between 80°E and 30°E (Meijers et al., 2010; Roden et
al., 2016). On the western side of the Kerguelen Plateau, CDBW also mixes with RSBW and ALBW (Orsi et al.,
1999; Van Wijk and Rintoul, 2014). In this context, the $C_{ant}$ concentrations observed in the bottom layer at OISO-
ST11 are probably not linked to a single AABW source, but are likely a complex interplay of AABW from different
sources with different biogeochemical properties.
Figure 2.
**3 Material and methods**
**3.1 Validation of the data**
For 1998-2004, the OISO data were quality controlled in CARINA (Lo Monaco et al., 2010) and for 2005 and
2009-2011 in GLODAPv2 (Key et al., 2015; Olsen et al., 2016, 2019). The 3 additional stations from GEOSECS
and INDIGO were qualified in GLODAP-v1 (Key et al., 2004) and previously used for the first $C_{ant}$ estimates in



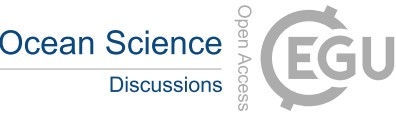

the Indian Ocean (Sabine et al., 1999). The data for INDIGO III (1987) have been revisited in GLODAP-v2 but
the correction applied on $A_T$ values leads to a suspicious offset and we decided to use the adjustment proposed in
GLODAP-v1 and confirmed in CARINA.
For the recent OISO cruises conducted in 2012-2018 not yet qualified in the GLODAP project, we have proceeded
to a data control mainly based on repeated observations in deep waters (CDW) where $C_{ant}$ concentrations are low
and subject to very small changes from year to year. At this location, the seasonal variations of all properties are
only observed in the mixed-layer, about 50 m in austral summer and 150 m in winter (Metzl et al., 2006). Therefore,
for deep water analysis, we used the observations available for all seasons (Table 1).

### 3.2 Biogeochemical measurements

Measurement methods during OISO cruises were previously described (Jabaud-Jan et al., 2004; Metzl et al., 2006).
In short, measurements were obtained using Conductivity-Temperature-Depth (CTD) casts fixed on a 24 bottles
rosette equipped with 12 L General Oceanics Niskin bottles. Θ (in °C) and S (no unit) measurements have an
accuracy of 0.01 of their respective units. $C_T$ and $A_T$ were sampled in 500 mL glass bottles and poisoned with 100
μL of $HgCl_2$ saturated solution to halt biological activity. Discrete $C_T$ and $A_T$ samples were analyzed onboard by
potentiometric titration derived from the method developed by Edmond (1970) using a closed cell. The accuracy
for $C_T$ and $A_T$ varies from 1 to 3.5 μmol.kg$^{-1}$ (depending on the cruise) and is determined by sample duplicates in
surface, at 1000 m and bottom waters. All measurements were calibrated with Certified Reference Materials
(CRMs) provided by A.G. Dickson laboratory (Scripps Institute of Oceanography). $O_2$ was determined by an
oxygen sensor fixed on the rosette. These values were adjusted using measurements obtained by Winkler titrations
using a potentiometric titration system (at least 12 measurements for each profile). Thiosulphate solution used in
Winkler titration was calibrated using iodate standard solution to provide the standard $O_2$ accuracy of 2 μmol.kg$^{-1}$
$^{-1}$. Nitrate ($NO_3$) and Silicate (Si) were measured onboard or offshore with an automatic colorimetric Technicon
analyser following the methods described by Tréguer and Le Corre (1975) until 2008, and the revised protocol
described by Aminot and Kérouel (2007) since 2009. Based on replicate measurements for deep samples we
estimate an error of about 0.3 % for both nutrients. $NO_3$ concentrations are not available for all the cruises used
in this analysis. The mean $NO_3$ concentrations in the AABW at OISO-ST11 is 32.8 ± 1.2 μmol.kg$^{-1}$ while the
average value derived from GLODAP-v2 database in bottom waters south of 50°S in the South Indian is 32.4 ±
0.6 μmol.kg$^{-1}$. The lack of $NO_3$ for few cruises has been palliated by considering a standard value of 33 μmol.kg$^{-1}$
with a limited impact on $C_{ant}$ determined by the C° method (from 0.1 μmol.kg$^{-1}$ to 1.7 μmol.kg$^{-1}$ on the mean
annual values).

### 3.3 $C_{ant}$ calculation using the TrOCA method

The TrOCA method was first presented by Touratier and Goyet (2004a, 2004b) and revised by Touratier et al.
(2007). Following the concept of the quasi-conservative tracer NO (Broecker, 1974), TrOCA is a tracer defined as
a combination of $O_2$, $C_T$ and $A_T$, following:
$$TrOCA = O_2 + a\left(C_T - \frac{1}{2}A_T\right), \tag{1}$$
where $a$ is the Redfield ratio.

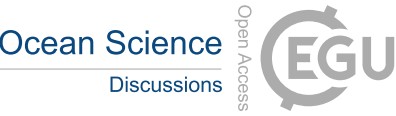

The temporal change in TrOCA is independent of biological processes and can be attributed to anthropogenic
carbon (Touratier and Goyet, 2004a). Therefore, $C_{ant}$ can be directly calculated from the difference between
TrOCA and its pre-industrial value TrOCA°:
$$C_{ant} = \frac{TrOCA - TrOCA^0}{a},$$ (2)
where TrOCA° is evaluated as a function of θ and $A_T$ (Eq. 3):
$$TrOCA^0 = e^{\left[b - (c).\theta - \frac{d}{A_T^2}\right]},$$ (3)
In these expressions, coefficients a, b, c and d were adjusted by Touratier et al. (2007) from free anthropogenic
$CO_2$ deep waters using the tracers $\Delta^{14}C$ and CFC-11 from the GLODAP-V1 database (Key et al., 2004). The final
expression used to calculate $C_{ant}$ is:
$$C_{ant} = \frac{O_2 + 1,279\left(C_T - \frac{1}{2}A_T\right) - e^{\left[7,511 - \left(1,087.10^{-2}\right).\theta - \frac{7,81.10^5}{A_T^2}\right]}}{1,279},$$ (4)
The consideration of the errors on the different parameters involved in the TrOCA method results in an uncertainty
of ±6.25 µmol.kg⁻¹ (mostly due to the parameter a, leading to ±3.31 µmol.kg⁻¹). As this error is relatively large
compared to the expected Cant concentrations in deep and bottom SO waters (Pardo et al., 2014) we will compare
the TrOCA results using another indirect method to interpret Cant changes over 40 years.
**3.4 $C_{ant}$ calculation using the preformed inorganic carbon method**
To support the $C_{ant}$ trend determined with the TrOCA method, $C_{ant}$ was also estimated using a back-calculation
approach noted C° (Brewer, 1978; Chen and Millero, 1979), previously adapted for $C_{ant}$ estimates along the
WOCE-I6 section between Africa and Antarctica (Lo Monaco et al., 2005a). This method consists in the correction
of the measured $C_T$ for the biological contribution ($C_{bio}$) and the preindustrial preformed $C_T$ ($C_{0,PI}$):
$$C_{ant} = C_T - C_{bio} - C_{0,PI},$$ (5)
$C_{bio}$ (Eq. 6) depends on carbonate dissolution and organic matter remineralization, taking account of the corrected
Redfield ratio from Kortzinger et al. (2001):
$$C_{bio} = 0.5\Delta A_T - (C/O_2 + 0.5N/O_2)\Delta O_2,$$ (6)
Where $C/O_2 = 106/138$ and $N/O_2 = 16/138$. $\Delta A_T$ and $\Delta O_2$ are the difference between the measured values ($A_T$ and
$O_2$) and the preindustrial values ($A_T^0$ and $O_2^0$). $A_T^0$ (Eq. 7) has been computed by Lo Monaco et al. (2005a) as a
function of Θ, S and the conservative tracer PO:
$$A_T^0 = 0.0685PO + 59.79S - 1.45\theta + 217.1,$$ (7)
PO (Eq. 8) has been defined by Broecker (1974) and depends on the equilibrium of $O_2$ with phosphate ($PO_4$). When
$PO_4$ data are not available, nitrate ($NO_3$) can be used instead (Anderson and Sarmiento, 1994):
$$PO = O_2 + 170PO_4 = O_2 + 170/16NO_3,$$ (8)
To determine $O_2^0$, it is assumed that the surface water is in full equilibrium with the atmosphere ($O_2^0 = O_{2,sat}$; Benson
and Krause, 1980) and after only impacted by the biological activity (Weiss, 1970). The correction of $O_{2,sat}$ has
been proposed by Lo Monaco et al. (2005a) to take account of the undersaturation of $O_2$ due to sea-ice cover. $\Delta O_2$
is, therefore, corrected by assuming a mean mixing ratio of the ice-covered surface waters k=50 % (Lo Monaco et
al., 2005a), and a mean value for $O_2$ undersaturation in ice-covered surface waters α=12 % (Anderson et al., 1991)
according to Eq. 9:





$\Delta O_2 = (1 - \alpha k)O_{2,sat} - O_2 = AOU$ ,       (9)
$C_{0,PI}$ in equation 5 is a function of the current preformed $C_T$ ($C_{0,obs}$) and a reference water term (Eq. 10):
$C_{0,PI} = C_{0,obs} + [C_T - C_{bio} - C_{0,obs}]_{REF}$ ,       (10)
Where the reference water term has been computed using an optimum multiparametric (OMP) model and defined
as 51 µmol.kg$^{-1}$ from North Atlantic deep water (Lo Monaco et al., 2005a) and $C_{0,obs}$ has been computed similarly
as $A_T^0$ (Eq. 11):
$C_{0,obs} = -0.0439PO + 42.79S - 12.02\theta + 739.8$ ,       (11)
For more details in the C° method, which has a final error of ± 6 µmol.kg$^{-1}$, especially on the determination of
reference water terms and on the errors of this method, please see Lo Monaco et al. (2005a).

## 4. Results

The vertical distribution of hydrological and biogeochemical properties observed in deep and bottom waters and
their evolution over the last 40 years are displayed in Fig. 3. The LCDW layer ($\gamma^n$ = 28.15-28.27 kg.m$^{-3}$) is
characterized by maximum AOU values (Fig. 3c), maximum $C_T$ concentrations (Fig. 3d) and minimum $C_{ant}$
concentrations (Fig. 3a). $C_{ant}$ concentrations were not significant in the LCDW until the end of the 1990s (<6
µmol.kg$^{-1}$), then our data show a sudden increase in $C_{ant}$ between January and December 1998, followed by
relatively constant $C_{ant}$ concentrations (10±3 µmol.kg$^{-1}$). In the core of AABW ($\gamma^n$ > 28.35 kg.m$^{-3}$), well identified
by low $\Theta$, low S, high $O_2$ and low AOU, $C_{ant}$ concentrations are higher than in the overlying deep waters (Fig. 3a)
and increased from 5±4 µmol.kg$^{-1}$ in 1978, 7±4 µmol.kg$^{-1}$ in the mid-1980s to 13±2 µmol.kg$^{-1}$ at the end of the
1990s and up to 19±2 µmol.kg$^{-1}$ in 2004 (Fig. 4a). Figure 4a also shows a very good agreement between the TrOCA
method and the C$^0$ method for both the magnitude and variability of $C_{ant}$ in the core of AABW. Our results show
a mean $C_{ant}$ trend in AABW of +1.6 µmol.kg$^{-1}$.decade$^{-1}$ over the full period and a maximum trend of the order of
+6.5 µmol.kg$^{-1}$.decade$^{-1}$ over 1987-2004 (Table 2). These trends are lower than the theoretical trend expected from
the increase in atmospheric $CO_2$. Indeed, assuming that the surface ocean $fCO_2$ follows the atmospheric growth
rate (+1.8 µatm.year$^{-1}$ over 1978-2018), the theoretical $C_{ant}$ trend at the AABW formation sites would be of the
order of +8 µmol.kg$^{-1}$.decade$^{-1}$. The observed slow $C_{ant}$ trends can be partly explained by the transit time for AABW
to reach our study site and the mixing of AABW with older CDW waters that contain less $C_{ant}$ (Fig. 3).
Figure 3.
To investigate changes in the accumulation of $C_{ant}$ in AABW, Fig. 4 shows the evolution of $C_T$, $A_T$, $O_2$, $\Theta$ and S
(properties used to estimate $C_{ant}$), as well as the "natural" component of $C_T$ ($C_{Tnat}$ calculated as the difference
between $C_T$ and $C_{ant}$). Over the full period, $C_T$ increased by 2.0±0.5 µmol.kg$^{-1}$.decade$^{-1}$, mostly due to the
accumulation of $C_{ant}$ (Table 2). Our data also show a significant decrease in $O_2$ concentrations by 0.8±0.4 µmol.kg$^{-1}$.decade$^{-1}$
over the 40-years period (Fig. 4c) that could be caused by reduced ventilation, as suggested by
Schmidtko et al. (2017) who observed significant $O_2$ loss in the global ocean. In the deep Indian SO sector, these
authors found a trend approaching -1 µmol.kg$^{-1}$.decade$^{-1}$ over 50 years (1960-2010), which is consistent with our
data. We did not detect any significant trend in $A_T$, $\Theta$ and S over the full period, but on shorter periods our data
show a significant decrease in $A_T$ from the mid-1980s to 2004 (Fig. 4d, Table 2) that is also observed in the
overlying deep waters (Fig. 3f). This could suggest reduced calcification in the upper ocean leading to less sinking
of calcium carbonate tests and hence a decrease in $C_{Tnat}$ (i.e. for this period the increase in $C_T$ was lower than the



accumulation of $C_{ant}$). This event is followed by an increase in $C_{Tnat}$ since 2004 associated to a rapid decrease in
$O_2$ (increase in AOU) and a decrease in $C_{ant}$ (Table 2). These recent trends were associated with a small increase
in θ (Fig. 4e, Table 2), but no significant trend in S (Fig. 4f). The increase in $C_{Tnat}$ is thus unlikely originating
solely from increased mixing with LCDW during bottom waters transport. Enhanced organic matter
remineralization is also unlikely since nitrate did not show any significant trend (Table 2).
Table 2.
Figure 4.
Importantly, our data show substantial interannual variations in AABW properties, which could significantly
impact the trends estimated from limited reoccupations (e.g. Williams et al., 2015; Pardo et al., 2017; Murata et
al., 2019). For example, we found relatively higher $C_{ant}$ concentrations in 1985 (10 µmol.kg$^{-1}$) compared to 1978
and 1987 (5 µmol.kg$^{-1}$). This is linked to a signal of low S in 1985 that could be due to a larger contribution of
fresher AABW or reduced mixing with saltier LCDW (Fig. 3h). Over the last decade (2009-2018), our data show
large and rapid changes in S that are partly reflected on $C_T$ and $O_2$, and that could explain the relatively low $C_{ant}$
concentrations observed over this period. Indeed, the S maximum observed in 2012 (correlated to higher θ) is
associated with a marked $C_T$ minimum (surprisingly almost as low as in 1987), as well as low $A_T$ (hence low $C_{Tnat}$),
and low nitrate concentrations. These anomalies point to a change in AABW characteristics rather than a change
in mixing with the underlying deep waters, and since they were associated with a decrease in $C_{ant}$ concentrations,
one may argue for an increased contribution of bottom waters ventilated far away from our study site (possibly
from the Ross Sea due to higher S, Fig. 2). A few years later our data show a S minimum (correlated to lower θ),
associated with a rapid increase in $C_T$ and a rapid decrease in $O_2$ between 2013 and 2016, suggesting the
contribution of a closer AABW such as the CDBW. The freshening of -0.01 in S that we observed on the Western
side of the Kerguelen Plateau was also observed on the Eastern side of the Plateau by Menezes et al. (2017). In
this region, Menezes et al. (2017) evaluated a change in salinity by about -0.008.decade$^{-1}$ from 2007 to 2016
(against -0.002.decade$^{-1}$ between 1994 and 2007), suggesting an acceleration of the AABW freshening in recent
years. However, they also reported a warming by +0.06 °C.decade$^{-1}$, while we observed cooler temperature in
2016-2018. This suggests that we sampled a different mixture of AABWs.
**5. Discussion**
**5.1. $C_{ant}$ concentrations**
In order to compare our $C_{ant}$ estimates with other studies, we separated the 40-years time-series into 3 periods: the
first period (1978-1987) corresponds to historical data when $C_{ant}$ is expected to be low; the second period (1998-
2004) starts when the first OISO cruise was conducted (and using CRMs for $A_T$, $C_T$ measurements) and lasts when
$C_{ant}$ concentrations in AABW are maximum (Fig. 4a); the third period consists in the observations performed in
late 2009 to 2018 when the observed variations are relatively large for S and small for $C_{ant}$. The mean $C_{ant}$
concentrations for each period are 7, 14 and 13 µmol.kg$^{-1}$, respectively, which is consistent with the results from
other studies (Table 3). The $C_{ant}$ values for 1978-1987 can hardly be compared to other studies because very few
observations were conducted in the 1980s in the SO Indian sector (Sabine et al. 1999) and because of potential
biases for historical data despite their careful qualification in CARINA (Lo Monaco et al., 2010). In addition, the
different methods used to estimate $C_{ant}$ can lead to different results, especially in deep and bottom waters of the





SO (Vázquez-Rodríguez et al., 2009). Overall, Table 3 confirms that $C_{ant}$ concentrations were low in the 1970s
and 1980s, and reached values of the order of 10 µmol.kg$^{-1}$ in the 1990s, a signal not clearly captured in global
data-based estimates (Gruber, 1998; Sabine et al., 2004; Waugh et al., 2006; Khatiwala et al., 2013).
The observations presented in this analysis, although regional, offer a complement to recent estimates of $C_{ant}$
changes evaluated between 1994 and 2007 in the top 3000 m for the global ocean (Gruber et al., 2019a). In the
Enderby Basin at the horizon 2000-3000 m, the accumulation of $C_{ant}$ from 1994 to 2007 is not uniform and ranges
between 0 and 8 µmol.kg$^{-1}$ (Gruber et al., 2019a). At our station, in the CDW (2000-3000m) the $C_{ant}$ concentrations
were not significant in 1978-1987 (-2 to 5 µmol.kg$^{-1}$) but increase to an average of 8.7±3.0 µmol.kg$^{-1}$ in 1998-
2018 (Fig. 3a) probably due to mixing with AABW that contain more $C_{ant}$. Interestingly, this value is close but in
the high range of the $C_{ant}$ accumulation estimated from 1994 to 2007 in deep waters of the south Indian Ocean
(Gruber et al., 2019a).
Not surprisingly, high $C_{ant}$ concentrations are detected in the AABW formation regions (Table 3). The highest $C_{ant}$
concentrations in bottom waters (up to 30 µmol.kg$^{-1}$) were observed in the ventilated shelf waters in the Ross Sea
(Sandrini et al., 2007). In the Adélie and Mertz Polynya regions, Shadwick et al. (2014) observed high $C_{ant}$
concentrations in the subsurface shelf waters (40-44 µmol.kg$^{-1}$) but lower values in ALBW (15 µmol.kg$^{-1}$) when
it mixed with older CDW. In WSBW, all $C_{ant}$ concentrations estimated from observations between 1996 and 2005
and with the TrOCA method (Table 3) lead to about the same values ranging between 13 and 16 µmol.kg$^{-1}$ (Lo
Monaco et al., 2005b; van Heuven et al., 2011). In bottom waters formed near the Cape Darnley (CDBW), Roden
et al. (2016) estimated high $C_{ant}$ concentrations in bottom waters (25 µmol.kg$^{-1}$) resulting from the shelf waters
that contain very high $C_{ant}$ (50 µmol.kg$^{-1}$). The comparison with other studies confirms that far from the AABW
formation sites, contemporary $C_{ant}$ concentrations are not exceeding 16 µmol.kg$^{-1}$ on average. However, higher
$C_{ant}$ concentrations are not unrealistic (Sandrini et al., 2007; Roden et al., 2016; this study in 2004) and likely
related to ventilation and water masses variability.
Table 3.
**5.2. $C_{ant}$ trends and variability**
Comparison of long-term $C_{ant}$ trends in deep and bottom waters of the SO is limited to very few regions where
repeated observations are available. To our knowledge, only 3 other studies evaluated the long-term $C_{ant}$ trends in
the Southern Ocean based on more than 5 reoccupations: in the South-western Atlantic (Rios et al., 2012) and in
the Weddell Gyre along the Prime meridian section (van Heuven et al., 2011, 2014). Temporal changes of $C_T$ and
$C_{ant}$ have also been investigated in other SO regions, but limited to 2 to 4 reoccupations (Murata et al., 2019;
Williams et al., 2015; Pardo et al., 2017). Given the $C_{ant}$ variability depicted at our location (Fig. 4a), different
trends can be deduced from limited reoccupations. As an example, Murata et al., (2019) evaluated the change in
$C_{ant}$ from data collected 17 years apart (1994–1996 and 2012–2013) along a transect around 62°S and found a
small increase at our location (< 5 µmol.kg$^{-1}$ around 60°E). This result appears very sensitive to the time of the
observation given that we found a minimum in $C_{ant}$ concentrations between 2011 and 2014 (Fig. 4a) associated
with a marked $C_T$ minimum (Fig. 4b). In addition, our results show that the detection of $C_{ant}$ trends appears very
sensitive to the time period considered (Table 2). As an extreme case, the $C_{ant}$ trend estimated for the period 1987-
2004 is +6.5 µmol.kg$^{-1}$.decade$^{-1}$ (close to the theoretical $C_{ant}$ trend of +8 µmol.kg$^{-1}$.decade$^{-1}$), but it reverses to -
3.5 µmol.kg$^{-1}$.decade$^{-1}$ for the period 2004-2018.



The long-term $C_T$ trend that we estimated in AABW in the eastern Enderby Basin (2.0±0.5 μmol.kg$^{-1}$.decade$^{-1}$) is
slightly faster than the $C_T$ trends estimated in the WSBW in the Weddell Gyre: +1.2±0.5 μmol.kg$^{-1}$.decade$^{-1}$ over
the period 1973-2011 and +1.6±1.4 μmol.kg$^{-1}$.decade$^{-1}$ when restricted to 1996-2011 (van Heuven et al., 2014).
Along the SR03 line (south of Tasmania) reoccupied in 1995, 2001, 2008 and 2011, Pardo et al (2017) evaluated
a $C_T$ trend of +2.4±0.2 μmol.kg$^{-1}$.decade$^{-1}$ in the AABW composed of ALBW and RSBW in this sector. This is
higher than the $C_T$ trends found at our location and in the Weddell Gyre, but surprisingly, this was not associated
with a significant increase in $C_{ant}$. The $C_T$ trend in AABW along the SR03 section was likely due to the intrusion
of old and $C_T$-rich waters also revealed by an increase in silicate concentrations during 1995-2011 (Pardo et al.,
2017). This is a clear example of decoupling between $C_T$ and $C_{ant}$ trends in deep and bottom waters as observed at
our location in the last decade (Table 2). For $C_{ant}$, our 40-years trend estimate (1.6±0.6 μmol.kg$^{-1}$.decade$^{-1}$) appears
close to the trend reported by Rios et al. (2012) in the South-Western Atlantic AABW from 6 reoccupations
between 1972 and 2003 (+1.5 μmol.kg$^{-1}$.decade$^{-1}$). However, if we limit our result to the period 1978-2002 or
1978-2004 (about the same period as in Rios et al., 2012), our trend is much larger (+3-4 μmol.kg$^{-1}$.decade$^{-1}$).
At our location, the $C_{ant}$ trend over 40 years (+1.6±0.6 μmol.kg$^{-1}$.decade$^{-1}$) explains most of the observed $C_T$
increase (+2.0±0.5 μmol.kg$^{-1}$.decade$^{-1}$). The residual of +0.4 μmol.kg$^{-1}$.decade$^{-1}$ reflects changes in natural
processes affecting the carbon content (different AABW sources, ventilation, mixing with deep waters,
remineralization or carbonates dissolution). Although this is a weak signal, the natural $C_T$ change ($C_{Tnat}$) mirrors
the observed decrease in $O_2$ by -0.80±0.4 μmol.kg$^{-1}$.decade$^{-1}$. This $O_2$ decrease detected in the Enderby Basin
appears to be a real feature that was documented at large scale for 1960-2010 in deep SO basins (Schmidtko et al.
2017), suggesting that the changes observed at 56.5°S-63°E are related to large-scale processes, possibly due to a
decrease in AABW formation (Purkey and Johnson, 2012).
**5.3. Recent $C_{ant}$ stability**
Although most studies suggest a gradual accumulation of $C_{ant}$ in the AABW, our time-series highlights significant
pluriannual changes, in particular over the last decade when $C_{ant}$ concentrations were as low as around the year
2000 (Fig. 4a) and decoupled from the increase in $C_T$ (Fig. 4b). This result is difficult to interpret because at our
location, away from AABW sources (Fig. 1), the temporal variability observed in the AABW layer can result from
many remote processes occurring at the AABW formation sites (such as wind forcing, ventilation, sea-ice melting,
thermodynamic, biological activity and air-sea exchanges), but internal processes during the transport of AABWs
(such as organic matter remineralization, carbonate dissolution and mixing with surrounding waters) must also be
taken into account. The apparent steady $C_{ant}$ feature suggests that AABWs found at our location has stored less
$C_{ant}$ in recent years. This might be linked to reduced $CO_2$ uptake in the AABW formation regions, as recognized
at large-scale in the SO from the late 1980s to 2001 (Le Quéré et al., 2007; Metzl, 2009; Lenton et al., 2012;
Landschützer et al., 2015). This large-scale response in the SO during a positive trend in the Southern Annular
Mode (SAM) is mainly associated to stronger winds driven by accelerating greenhouse gas emissions and
stratospheric ozone depletion, leading to warming and freshening in the SO (Swart et al., 2018), change in the
ventilation of the carbon-rich deep waters and reduced $CO_2$ uptake (Lenton et al., 2009). The reconstructed $pCO_2$
fields by Landschützer et al. (2015) suggest that the reduced $CO_2$ sink in the 1990s is identified at high latitudes
in the SO (see Fig. 2a and S9 in Landschützer et al., 2015). However, as opposed to the circumpolar open ocean
zone (e.g. Metzl, 2009; Takahashi et al., 2009, 2012; Munro et al., 2015; Fay et al., 2018), the long-term trend of





surface $fCO_2$ and carbon uptake deduced from direct observations are not clearly identified in the seasonal ice
zone (SIZ) and shelves around Antarctica, and thus in the AABW formation regions of interest to interpret our
results (Laruelle et al., 2018). There, surface $fCO_2$ data are sparse, especially before 1990, and cruises were mainly
conducted in austral summer when the spatio-temporal $fCO_2$ variability is very large and driven by multiple
processes at regional or small scales, such as primary production, sea-ice formation and retreat, circulation and
mixing. This leads to various estimates of the air-sea $CO_2$ fluxes around Antarctica depending on the region and
period and large uncertainty when attempting to detect long-term trends (Gregor et al., 2018).
In particular, in polynyas and AABW formation regions where $fCO_2$ is low and where katabatic winds prevail,
very strong instantaneous $CO_2$ sink can occur at the local scale (up to -250 mmol $C.m^{-2}.d^{-1}$ in Terra Nova Bay in
the Ross Sea according to DeJong and Dunbar, 2017). In the Prydz Bay region where CDBW is formed, recent
studies show that surface $fCO_2$ in austral summer vary in a very large range (150-450 µatm), with the lowest $fCO_2$
observed in the shelf region generating very strong local $CO_2$ sink (-221 mmol $C.m^{-2}.d^{-1}$ according to Roden et al.
2016). The carbon uptake was particularly enhanced near Cape Darnley and coincided with the highest $C_{ant}$
concentrations that Roden et al. (2016) estimated in the dense shelf waters that subduct to form AABW. In the
Prydz Bay coastal region, surface $fCO_2$ values in 1993-1995 were as low as 100 µatm (Gibson and Trull, 1999)
leading to a strong local $CO_2$ uptake of -30 mmol $C.m^{-2}.d^{-1}$ in summer. In addition, Roden et al. (2013) found a
large $C_T$ increase over 16 years (+34 µmol.kg$^{-1}$) in the Prydz Bay, which is much higher than the anthropogenic
signal alone (+12 µmol.kg$^{-1}$) and likely explained by changes in primary production that would have been stronger
in 1994. To our knowledge, this is the only direct observation of decadal $C_T$ change in surface waters in a region
of AABW formation (here the Prydz Bay) and it highlights the difficulty not only to evaluate the $C_T$ and $C_{ant}$ long-
term trends in these regions but also to separate natural and anthropogenic signals when this water reaches the
deep ocean. We attempted to detect long-term changes in $CO_2$ uptake in this region using the qualified $fCO_2$ data
available in the SOCAT database (Bakker et al., 2016), but our estimates (not shown) were highly uncertain due
to very large spatial and temporal variability. To conclude, all previous studies conducted near or in AABW
formation sites clearly reveal that these regions are potentially strong carbon sinks, but how the sink changed over
the last decades is not yet evaluated, and thus we are not able to certify that the recent $C_{ant}$ stability that we observed
in the AABW at our location is directly linked to the weakening of the carbon sink that was recognized at large-
scale in the SO from the 1980s to mid-2000s (Le Quéré et al., 2007; Landschützer et al., 2015).
Changes in the accumulation of $C_{ant}$ in AABW could also be directly related to changes in physical processes
occurring in AABW formation region. Decadal decreasing of sea-ice production and melting of sea-ice have been
documented in several regions including Cape Darnley polynyas (Tamura et al., 2016; Williams et al., 2016). The
consequent changes in Antarctic surface waters properties are transmitted into the deep ocean, notably the well-
recognized freshening of AABWs over the last decades (Rintoul, 2007). The warming of bottom waters was also
documented in the Enderby basin (Couldrey et al., 2013) as well as at a larger scale in all deep SO basins (Purkey
and Johnson, 2010; Desbruyères et al., 2016). Associated to a decrease in AABW formation in the 1990s (Purkey
and Johnson, 2012), these physical changes could explain the recent stability of $C_{ant}$ concentrations in AABW
observed at our location. As AABWs from different sources spread and mix with $C_T$-rich deep waters before
reaching our location (Fig. 1), less AABW formation and export would result in an increase in $C_T$ (increase in
$C_{Tnat}$) not associated with an increase in $C_{ant}$, and a decrease in $O_2$ (as observed in recent years in Fig. 4a,b,c).
Finally, it is also possible that the AABW observed in recent years at our location is the result of a larger





contribution of older RSBW and/or ALBW that have lower $C_{ant}$ and $O_2$ concentrations compared to CDBW formed
at Cape Darnley and Prydz Bay.
**6. Conclusion**
The distribution and evolution of $C_{ant}$ in the bottom layer of the SO are related to complex interactions between
climatic forcing, air-sea $CO_2$ exchange at formation sites, as well as biological and physical processes during
AABWs circulation. The dataset that we collected regularly in the Enderby basin over the last 20 years (1998-
2018) in the frame of the OISO project, together with historical observations obtained in 1978, 1985 and 1987
(GEOSECS and INDIGO cruises), allows the investigation of $C_{ant}$ changes in AABW over 40 years in this region.
Our results suggest that the accumulation of $C_{ant}$ explains most of (but not all) the observed increase in $C_T$. We
also detected a decrease in $O_2$ that is consistent with the large-scale signal reported by Schmidtko et al. (2017),
possibly due to a decrease in AABW formation (Purkey and Johnson, 2012). Our data further indicate rapid
anomalies in some periods suggesting that for decadal to long-term estimates care have to be taken when analyzing
the change in $C_{ant}$ from data sets collected 10 or 20 years apart (e.g. Williams et al., 2015; Murata et al., 2019).
Our results also show different $C_{ant}$ trends on short periods, with a maximum increase of 6.5 $\mu mol.kg^{-1}.decade^{-1}$
between 1987 and 2004 and an apparent stability in the last 20 years (despite an increase in $C_T$). This suggests that
AABWs have stored less $C_{ant}$ in the last decade, but our understanding of the processes that explain this signal is
not clear. This might be the result of the reduced $CO_2$ uptake in the SO in the 1990s (Le Quéré et al., 2007;
Landschützer et al., 2015), but this is not yet verified from direct $C_T$ or $fCO_2$ observations in AABW formation
regions due to the lack of winter data and very large variability during summer. This calls for more data collection
and investigations in these regions. The apparent stability of $C_{ant}$ in AABW since 1998 could also be directly linked
to a decrease in AABW formation in the 1990s (Purkey and Johnson, 2012) or a change in the contributions of
AABWs from different sources, especially in the Prydz Bay region (Williams et al., 2016). In these scenarios, an
increased contribution of $C_T$-rich and $O_2$-poor older CDW would also explain the decoupling between $C_{ant}$ and $C_T$
(increase in $C_{Tnat}$) and decrease in $O_2$ concentrations observed in recent years. The decoupling between $C_{ant}$ and
$C_T$ is not a unique feature, as it was also reported along the SR03 section between Tasmania and Antarctica, most
probably due to advection of $C_T$-rich waters (Pardo et al., 2017). This highlights the importance of the ocean
circulation in influencing the temporal $C_T$ and $C_{ant}$ inventories changes (De Vries et al. 2017) and the need to better
separate anthropogenic and natural variability based on time-series observations.
The evaluation and understanding of decadal $C_{ant}$ changes in deep and bottom ocean waters are still challenging,
as the $C_{ant}$ concentrations remain low compared to $C_T$ measurements accuracy (at best $\pm$ 2 $\mu mol.kg^{-1}$, Bockmon
and Dickson, 2015) and uncertainties of data-based methods ($\pm$ 6 $\mu mol.kg^{-1}$). Long-term repeated and qualified
observations (at least 30 years) are needed to accurately detect and separate the anthropogenic signal from the
internal ocean variability; we thus only start to document these trends that should now help to identify
shortcomings in models regarding the carbon storage in the deep SO (e.g., Frölicher et al., 2014). As changes in
the SO (including warming, freshening, oxygenation/deoxygenation, $CO_2$ and acidification) are expected to
accelerate in the future in response to anthropogenic forcing and climate change (e.g. Hauck et al., 2015; Heuzé et
al., 2014; Ito et al., 2015, Yamamoto et al., 2015), it is important to maintain time-series observations to
complement the GO-SHIP strategy, and to occupy more regularly other sectors of the SO (Rintoul et al., 2012). In



this context, we hope to maintain our observations in the Southern Indian Ocean in the next decade, and with
ongoing synthetic products activities such as GLODAPv2 (Olsen et al., 2016; 2019), SOCAT (Bakker et al., 2016)
and more recently the SOCCOM project (Williams et al., 2018), to offer a solid database to validate ocean
biogeochemical models and coupled climate/carbon models (Russell et al. 2018), and ultimately reduce
uncertainties in future climate projections.
**Data availability**
GEOSECS, INDIGO and OISO 1998-2011 data are publicly available at the Ocean Carbon Data System (OCADS;
https://www.nodc.noaa.gov/ocads/oceans/GLODAPv2_2019). OISO original data stations are available at:
www.nodc.noaa.gov/ocads/oceans/RepeatSections/clivar_oiso.html. OISO 2012-2018 will be available in
GLODAPv2_2021.
**Author contributions**
LM, CLM, NM, JF and CM performed the sampling and carried out the measurements of the OISO data. LM
prepared the manuscript with contributions from CLM and NM.
**Competing interests**
The authors declare that they have no conflict of interest.
**Acknowledgements**
We thank the captains and crew of *R.S.V. Marion Dufresne* and the staff at the French Polar Institute (IPEV) for
their important contribution to the success of the cruises since 1998. We are also very grateful to all colleagues,
students and technicians who helped to obtain the data. We extend our gratitude to S. R. Rintoul and B. Legresy
for the discussions during the preparation of the manuscript and to M. K. Shipton for the valuable comments. The
OISO program is supported by the French institutes INSU and IPEV and the French program SOERE/Great-Gases.
Support from the European Integrated Projects CARBOOCEAN (511176) and CARBOCHANGE (264879) is also
acknowledged.

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




**Table 1. List of the cruises used in this study.**

| Cruise | Station | Location | Year | Month |
|---|---|---|---|---|
| GEOSECS | 430 | 61.0°E / 60.0°S | 1978 | February |
| INDIGO-1 | 14 | 58.9°E / 53.0°S | 1985 | March |
| INDIGO-3 | 75 | 63.2°E / 56.5°S | 1987 | January |
| OISO-01 | 11 | 63.0°E / 56.5°S | 1998 | February |
| OISO-03 | 11 | 63.0°E / 56.5°S | 1998 | December |
| OISO-05 | 11 | 63.0°E / 56.5°S | 2000 | August |
| OISO-06 | 11 | 63.0°E / 56.5°S | 2001 | January |
| OISO-08 | 11 | 63.0°E / 56.5°S | 2002 | January |
| OISO-11 | 11 | 63.0°E / 56.5°S | 2004 | January |
| OISO-18 | 11 | 63.0°E / 56.5°S | 2009 | December |
| OISO-19 | 11 | 63.0°E / 56.5°S | 2011 | January |
| OISO-21 | 11 | 63.0°E / 56.5°S | 2012 | February |
| OISO-23 | 11 | 63.0°E / 56.5°S | 2014 | January |
| OISO-26 | 11 | 63.0°E / 56.5°S | 2016 | October |
| OISO-27 | 11 | 63.0°E / 56.5°S | 2017 | January |
| OISO-28 | 11 | 63.0°E / 56.5°S | 2018 | January |





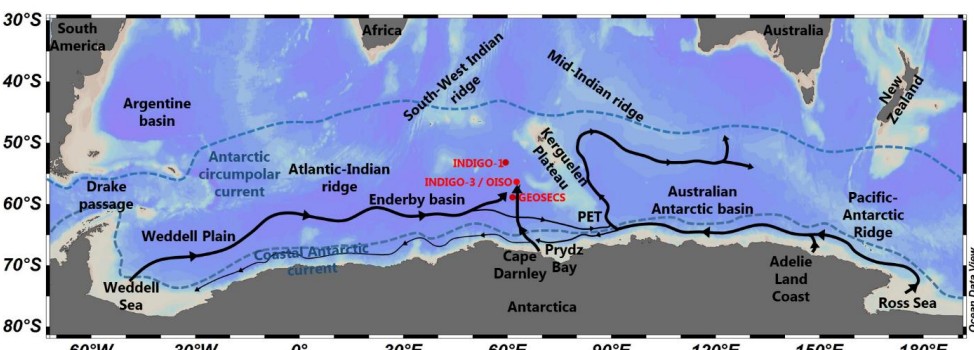

**Figure 1. The AABWs circulation from the literature (Fukamachi et al., 2010; Orsi et al., 1999) and this study, with**
**geographic indications (black text), SO currents (blue text and dash lines for the approximative positions) and stations**
**considered in this study (red text and dots). PET: Princess Elizabeth Trough. Figure produced with ODV (Schlitzer et**
**al., 2019).**




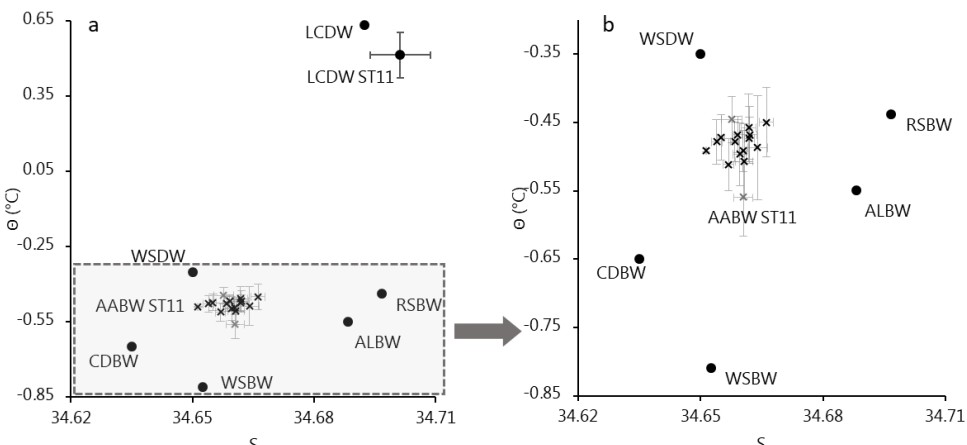


**Figure 2. (a) Full Θ-S diagram of studied water masses and (b) zoomed on bottom waters. Values are from literature for the WSBW (Fukamachi et al., 2010; van Heuven, 2013; Pardo et al., 2014; Robertson et al., 2002), the WSDW (Carmack and Foster, 1975; Fahrbach et al., 1994; van Heuven, 2013; Robertson et al., 2002), the RSBW (Fukamachi et al., 2010; Gordon et al., 2015; Johnson, 2008; Pardo et al., 2014), the ALBW (Fukamachi et al., 2010; Johnson, 2008; Pardo et al., 2014), the CDBW (Ohshima et al., 2013) and the LCDW (Lo Monaco et al., 2005a; Pardo et al., 2014; Smith and Treguer, 1994), and from the OISO-ST11 dataset for the OISO-ST11 AABW and OISO-ST11 LCDW. Error bars are calculated from the individual annual averaged values for the OISO-ST11 AABW and from all data for the OISO-ST11 LCDW. For the OISO-ST11 AABW, the grey cross are the GEOSECS (lowest Θ) and INDIGO-1 (highest Θ) values.**









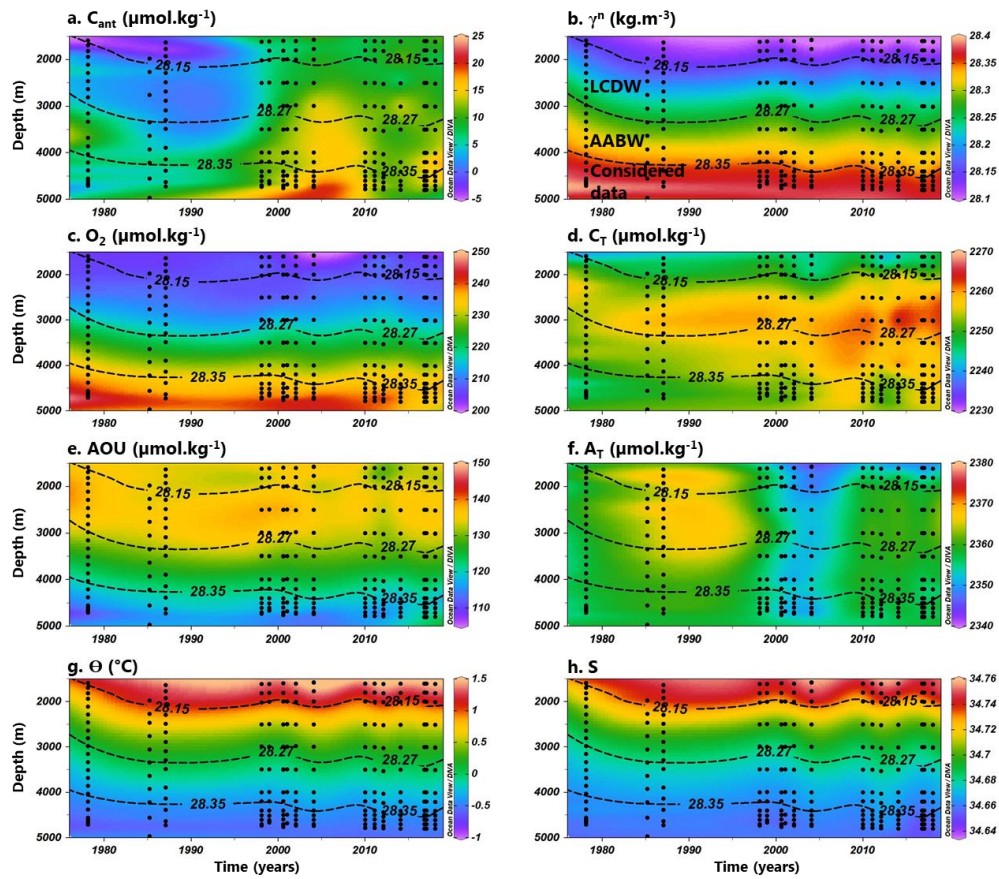

**Figure 3. Hovmöller section of the carbon related properties parameters ($C_{ant}$ via TrOCA, $O_2$, AOU, $C_T$ and $A_T$) and hydrological properties (θ, S and $γ^n$) of the dataset presented in Table 1 from 1978 to 2018 and from 1500 m to the bottom. Data points are represented by the black dots. The dash isolines represent the water masses separation by $γ^n$ detailled on the $γ^n$ plot. Figure produced with ODV (Schlitzer et al., 2019).**


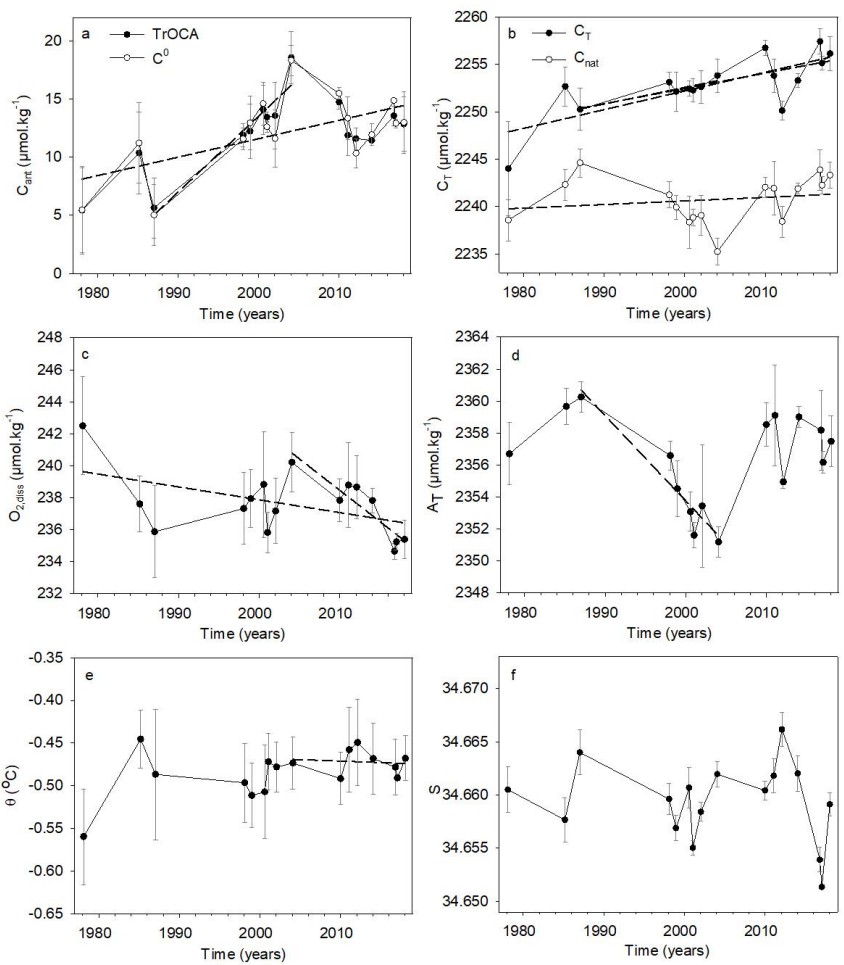

**Figure 4. Interannual variability (full lines) and significant trends (at 95%, see Table 2; dotted lines) for the 40 years of observation of the OISO-ST11 AABW properties, including (a) $C_{ant}$ by the TrOCA and the $C°$ method, (b) $C_T$ and $C_{nat}$, (c) $O_2$, (d) $A_T$, (e) $\Theta$ and (f) S.**




**Table 2: Trends (per decade) of observed and calculated properties in the AABW layer estimated over different periods**
**(in bold: significant trends at 95% confidence level).**

| Period | S | | $\Theta$ °C | | Silicate µmol.kg$^{-1}$ | | Nitrate µmol.kg$^{-1}$ | | O$_2$ µmol.kg$^{-1}$ | | AOU µmol.kg$^{-1}$ | | A$_T$ µmol.kg$^{-1}$ | | C$_T$ µmol.kg$^{-1}$ | | C$_{ant}$ TrOCA µmol.kg$^{-1}$ | |
|---|---|---|---|---|---|---|---|---|---|---|---|---|---|---|---|---|---|---|
| **1978-2018** | -0.001 | ± 0.001 | 0.01 | ± 0.01 | -1.2 | ± 0.9 | 0.2 | ± 0.2 | **-0.8** | **± 0.4** | **0.7** | **± 0.0** | -0.1 | ± 0.1 | **2.0** | **± 0.5** | **1.6** | **± 0.6** |
| **1987-2018** | -0.001 | ± 0.001 | 0.01 | ± 0.01 | -1.9 | ± 1.4 | 0.3 | ± 0.4 | -0.3 | ± 0.5 | 0.2 | ± 0.5 | 0.6 | ± 0.1 | **1.6** | **± 0.5** | 1.1 | ± 0.8 |
| **1987-2004** | -0.003 | ± 0.002 | 0.01 | ± 0.01 | **-6.5** | **± 1.8** | 0.9 | ± 0.9 | 1.7 | ± 1.0 | -1.7 | ± 1.0 | **-5.3** | **± 0.1** | **1.8** | **± 0.4** | **6.5** | **± 1.0** |
| **2004-2018** | -0.006 | ± 0.003 | **0.02** | **± 0.01** | -1.8 | ± 4.5 | -0.5 | ± 1.0 | **-3.9** | **± 0.7** | **4.0** | **± 0.8** | 3.4 | ± 0.2 | 1.7 | ± 1.9 | -3.5 | ± 1.5 |




**Table 3. Compilation of C$_{ant}$ sequestration investigations in the AABWs ($\gamma^n \geq 28.25$ kg.m$^{-3}$) using the TrOCA method.**
**The C$_{ant}$ estimation of Pardo et al. (2014) is calculated using theoretical AABW mean composition (with 3% of ALBW)**
**and the carbon data from the GLODAPv1 and CARINA databases. Sandrini et al. (2007) values has been measured at**
**the bottom in the Ross Sea and correspond to recently sunk HSSW. The mean values published by Roden et al. (2016)**
**for the AABWs present WSDW characteristics but can be a mix of CDBW and CDW.**

| Source | Location | Water masses considered | Year | C$_{ant}$ µmol.kg$^{-1}$ |
|---|---|---|---|---|
| Pardo et al. (2014) Fig. 5 | Averaged AABW composition | WSBW-RSBW-ALBW | 1994 | 12 |
| Lo Monaco et al. (2005b) Fig. 4b | WOCE line I6 (30°E; 50°-70°S) | WSBW CDBW | 1996 | 15 20 |
| Sandrini et al. (2007) Fig. 4a | Ross Sea | HSSW (previous RSBW) | 2002/2003 | Max. of 30 |
| Shadwick et al. (2014) Table 2 | Mertz polynya and Adelie depression | ALBW | 2007/2008 | 15 |
| Roden et al. (2016) Table 2 | South Indian ocean (30°-80°E; 60°-69°S) | WSDW-CDW-CDBW | 2006 | 25 |
| van Heuven et al. (2011) Fig.13 | Weddell gyre (0°E; 55°-71°S) | WSBW | 2005 | 16 |
| | | | 1978-1987 | 7 ± 3 |
| | | | 1987-1998 | 9 ± 4 |
| This study | Enderby basin (56.5°S-63°E) | WSDW-CDBW-RSBW-ALBW | 1987-2004 | 13 ± 4 |
| | | | 1998-2004 | 14 ± 2 |
| | | | 2010-2018 | 13 ± 1 |
| | | | **1978-2018** | **12 ± 3** |

