# Peer review of "Variability and stability of anthropogenic CO2 in Antarctic"

_Ocean Science, 2020_

## Referee Comment (RC1) · Anonymous Referee #1 · 2 Jun 2020

General comments: This manuscript deals with temporal variations of anthropogenic CO2 in bottom waters in the Southern Ocean. The Southern Ocean is said to take up 40% of anthropogenic CO2 absorbed by the ocean. Thus, investigations of temporal variability of anthropogenic CO2 are very important to evaluate ocean's capacity of absorbing atmospheric CO2, information of which is indispensable for the projection of global warming. In terms of oceanic observation, the Southern Ocean is one of the regions, where the number of measurements, especially for chemical and biological properties, is scare. In this point also, it is worth of being published in the journal. The manuscript is well organized, and is easy to read. The approaches used in the study are not new, but traditional ones. It is not a problem. It would be necessary

to adopt an approach, which has been demonstrated to be useful for the detection of small signals of anthropogenic CO2 variations. The authors attempt also to relate the variations to those of AABW formation, although not clearly found. As a whole, it seems that the manuscript is worthy of publication in the journal, but after a moderate revision. A few major comments are stated in the followings, and the minor ones are stated in the specific comments. In this paper, temporal variability of anthropogenic CO2 is examined using historical data collected at OISO. The data have been quality controlled by some data synthesis activities such as GLODAP. Nevertheless, I have a question on this point; the data syntheses have been done with a purpose of obtaining data consistency of a basin-scale. By contrast, the authors examine temporal variability of a local scale. In addition, data consistency is usually confirmed by data in deep layers of > 2000 m. This paper deals with data in deep layers. From these points, it is necessary to show that results obtained in the present study is not influenced by the data synthesis. Furthermore, for the recent data, quality control is made independently. Is there any possibility that the Cant stability is caused by the quality control? I recommend the authors to conduct quality-control on OISO data independently. In discussion, the authors attempt to relate variations of anthropogenic CO2 in AABW to changes in AABW formation region. It is well discussed, but information of water mass age of AABW is lacking. It is necessary to show that linkages between variations of AABW formation region and observed AABW signals at OISO are appropriate in terms of water mass age. O2 and AOU are used simultaneously. I think, it is enough for one of which, probably AOU.

Specific comments: Line 18: "from about +7 ïĄ■mol kg-1", increase from what?

Line 20: "CT", this is the first appearance in the abstract. Write it in full.

Line 23: "ïĄŚ, S", they are the first appearance in the abstract. Write them in full.

Lines 90-91: "station 430", depth?

Line 91: "405 km and 465 km", away from where?

Line 109: "the PET sector", is it usually used? I do not understand where it is.

Line 150: "AT", Probably this is the first appearance. Spell out here.

Line 160: "ïĄŚ and S", spell out here.

Lines 163-165: according the description, it seems that the figures are not accuracy but repeatability.

Line 236: "January", which year. In this paper, all the data are analyzed assuming that seasonal variations in deep waters are negligible (lines 154-156). It is not appropriate to refer to months.

Line 276: "underlying", do you mean a water mass below AABW?

---

## Referee Comment (RC2) · Anonymous Referee #2 · 9 Jun 2020

General comments The study presents results from a time series in the Indian sector of the Southern Ocean, which together with historical relevant data span a 40-year period. Using this time series, the authors evaluate the evolution of anthropogenic CO2 (Cant) in the Antarctic Bottom Waters (AABW). It is an interesting and generally well written work, and generally good figures and tables. There are some need for clarity in some parts and there is some concern of the treatment of data gaps, but most of this should be rather easily dealt with, and I recommend publication after minor revision. A detailed list of comments follows below.

My main comments are related to the definition and subsequent presentation of AABW,

[Figure]

and, the data gap between 1987 and 1998 and how this is handled and presented.

To start with the definition of AABW, this is not an issue in itself, since the denser definition has been used before, and also, since almost any definition can be accepted as long as it is clearly presented. The latter is the problem here, at least for someone not as familiar with the area and these water masses (I usually work in the high-northern latitudes). The definition and choice is clearly described in 2.3, but, then the reader is referred to Fig. 3, where AABW is noted in the layer above the focus of this study, while the data evaluated is in the layer annotated "Considered data". When then the results of the property evolution of AABW are further presented in Fig. 4, at least I got somewhat confused. Whether this is only me or not, this may call for some added clarity. I would suggest to annotate your AABW layer (hence at neutral density >28.35) as AABW (or AABW* or similar), to make this clear, and then make a distinction with the more common AABW. Nevertheless, this mostly refers to Fig. 3, and I have several concerns with this figure, as detailed below.

Hovmöller plot is a wonderful thing, and can be very illustrative. However, it can also be deceiving, especially when there are gaps in the data, and the gridding is allowed to interpolate over these gaps, which often can create features that give a false picture of actual evolution. Fig. 3 suffers from this when plotting the older data (1978–1987) together with the OISO time-series data starting from 1998. There are several peculiar features in Fig. 3, especially for Cant and AT. The fact that most of the other plotted parameters show overall stable layer properties, over the full period, may seem to reduce this concern, but I am not convinced. In addition, I'm not fully convinced about the benefit of showing depths from 1500 m, when almost all results and discussion are concerned with the layer below 4000 m. Even more so when the upper layers seems to show most of the strange features, for example the minimum in Cant in the older data (which may in part show the issue with the TrOCA method, with even negative concentrations, which are not realistic, in the most upper part of the deep waters). The interpolation of this minimum patch leads to unfortunate wordings in the results,

such as on line 236, with "a sudden increase. . . between January and December 1998" seems to refer to the low values calculated for the 1987 data and the clearly higher concentrations calculated for the OISO data. (I also don't really understand the "between Jan and Dec 1998" part, since the first OISO data were sampled in Feb 1998, and the next in Dec the same year.) Apparently there are some need for clarifications here, but also to be cautious when interpreting interpolated values over large gaps. One way to solve this is of course to exclude the older data from the Hovmöller plots. These can still be used in the comparison/evaluation, and included in Fig. 4. To continue on this figure (Fig. 3), for the bottom layer, the fact that it is stretched below the deepest samples seems to create at least the distinct maximum in mid-2000s. Perhaps this will be reduced if the maximum depth/pressure is set to the deepest sample, to exclude extrapolations below that depth.

Specific comments L18: Do the changes here (+7 and +13, respectively) refer to the whole period? Please clarify.

L23: A rather tiny remark, but the use of "pluriannual" may be grammatically correct (I'm not a native English speaker), but consider using "multiannual" (or multi-annual), which are more common (I believe). The same is used on L360.

L59: I'm expecting a reference in the end of this sentence. This may be refer to the reference in the previous line, but you may consider moving this to the end.

L95: I can't find a definition of "AAC" anywhere. Please write out and define the first time.

L96-97: Unclear sentence. Need some rephrasing/re-writing. Suggestion: ". . .Weddell Sea, where deep and bottom waters are produced. . .". L98-100: In the same sentence, there are several instances where the full water mass name is not spelled out, for example "the Ross Sea (RSBW; . . .". This may be intuitive, but I don't think the full names of some of these are written out at any place in the manuscript so would suggest to consider doing that at some place.
L100: Rephrase: In the Prytz Bay, AABW formation has also. . . This sentence is overall quite unclear, especially the last part, so please consider rewriting for clarification.

L105: The "Warm Deep Water" is not described, so not easy to follow without a previous knowledge of the area and the present water masses. Please clarify.

Section 2.4: Part of this section, and in particular from L133, deals with results of Cant from the methods not yet described. I would suggest to move this to the Result section, at least the Cant parts, or maybe part of the Discussion.

L152: Since the "P" in GLODAP refers to "Project", the "project" after should be avoided (I think). You could rephrase this into something like: . . ..not yet qualified (or included in) the most recent GLODAPv2 product.

L161: The stated accuracy for temperature and salinity seems too low. The standard CTD accuracy, for example found at the GO-SHIP home page (Hydro-manual) is 0.002 for both. Please check. L161: As far as I can see, this is the first time "AT" is mentioned, but not defined. Please add this.

L166: Same for "O2" as for AT above. Please define first time.

L170: You mean "onshore"?

L184: Clarify which "Redfield ratio". You mean the C:O ratio? Please add this.

L217: Either remove "after", so it reads ". . . and only impacted by. . .", or if more correct, add "subduction", so it reads "and after subduction only impacted by. . .".

L233: "LCBW" is here mentioned for the first time, without definition or any description anywhere in the manuscript, as far as I can see. Please add this.

L235-236: This is what was commented on in the generall comments above, with the "sudden increase". Please revise and clarify. It is more likely that there was a more gradual evolution, and none of the other parameters calls for any sudden changes. Also, the data quality and methods between the older data and the OISO data may

differ, so extra causion is taken when comparing them.

L240: The maximum in Cant in 2004 is one occasion, and followed by five (almost six) years without any data. I would be cautious to overinterpret this. However, it co-incides with a maximum in oxygen, which could indicate a ventilation event.

L256-260: The lower concentrations of AT in the years around 2000 at all depths below (at least) 1500 m (have you checked the whole water column?) seems a bit odd. Especially when this is not seen in any of the other parameters. Also, when comparing two years in the 1980s with data more than a decade later, one should be extra cautious in the interpretation, not the least when the two years/occasions in 1985/87 show the highest concentrations seen over the evaluated period. Certainly the years after 2000 show much lower concentrations, which may be a phase due to a change in different forcing, but to suggest reduced calcification from only a few years/occupations of data is very speculative, and clearly something that change a few years later.

L259-260: Is it realistic that the increase in CT is lower than the accumulation of Cant?

L261: While there is a rather clear trend in oxygen during this period – although I would be careful in talking about trends over such short periods, especially when comparing to a year with a maximum (2004) – there is no trend in Cant. Instead the latter shows some clear interannual variability. Also, the "trend" in temperature is indeed very small, and even if not significant, the change, or better, variability, in salinity is rather large. Consider these points when revising this part. Your statement on L267-268 highlights this issue.

L270-271: There is also a maximum in temperature in 1985, so this could indicate more mixing with WSDW, which are both fresher and warmer.

L275-278: This is a very long sentence. I suggest to divide it, with period after "...the underlying deep waters." Then remove "and", and start on "Since", or change the start of the sentence. For the last part of this sentence (L277-278), the suggestion of

increased contribution from the Ross Sea is not clear to me since the oxygen decrease, while the salinity goes up and down. Or are you only referring to the one occupation in 2012? (If this is the case, it seems to detailed to explain a single year taken out of a long time series.)

L280: The stated freshening of 0.01, for which period is that observed? Please clarify.

L312-313: "...(15 umol kg-1) due to mixing with older CDW."

L317: "that contain very high amounts of Cant ..."

L318-320: The last sentence of this paragraph basically repeats what have been said above. Consider to remove.

L325: Here you write out "Southern Ocean" after having used the abbreviation throughout the manuscript, even the sentence before. Consider to revise.

L340: "evaluated" should here instead be "estimated", or "calculated", or "found" (I think).

L386: Consider rewording "...vary in a very large range...". Suggestion: "show a very large variability", or maybe, "vary over a very large range".

L387-388: "(–221 mmol C m-2 d-1; Roden et al., 2016).

L416: Both these water masses (RSBW and ALBW) have higher salinity, and while oxygen show a reduced trend the salinity goes up and down, so this explanation does not hold for all years during this period.

L424: "explains most, but not all, of the observed..."

L463: GLODAPv2 version are written as "GLODAPv2.2021 (.2020 is soon to be released). You do mean 2021 and not 2020?

L851-853: Table 2 (and in general): You may want to consider if you want to keep AOU as parameter, when you mostly refer to oxygen. The trends are almost exactly the

same (but opposite of course), and gives the same message.

Technical comments L22: This is, however, modulated. . .

L35: The references should, typically, be chronologically ordered. Please check throughout the manuscript. (There are more examples of this, but I won't comment on this more.)

L71: This is, however, not the. . .

L91: ". . .(405 and 465 km, respectively)."

L107-113: Examplified with ". . .East of the Kerguelen. . .", this section has many of these "directions/locations" (east/west/. . .) spelled with a large letter, even not part of a name. I think this is not correct, and if so, please change.

L118-119: . . .28.27-bottom, respectively. . .

L172-173: Change font; the part of the sentence from "for deep samples. . ." are in a different font (maybe "Cambria").

L220: Change font for "value for".

L306: Add a comma: "2018 (Fig 3a), probably . . ."

L340: Add a ".": Pardo et al. (2017)

L347: For consistency, change "South-Western" to "South-western" (similar as on L325).

L449: Remove "." for consistency: (e.g. Frölicher et al., 2014).

L451: References in chronological order.

---

## Author Comment (AC1) · 5 Aug 2020

General comments: This manuscript deals with temporal variations of anthropogenic CO2 in bottom waters in the Southern Ocean. The Southern Ocean is said to take up 40% of anthropogenic CO2 absorbed by the ocean. Thus, investigations of temporal variability of anthropogenic CO2 are very important to evaluate ocean's capacity of absorbing atmospheric CO2, information of which is indispensable for the projection of global warming. In terms of oceanic observation, the Southern Ocean is one of the regions, where the number of measurements, especially for chemical and biological properties, is scare. In this point also, it is worth of being published in the journal. The

manuscript is well organized, and is easy to read. The approaches used in the study are not new, but traditional ones. It is not a problem. It would be necessary to adopt an approach, which has been demonstrated to be useful for the detection of small signals of anthropogenic CO2 variations. The authors attempt also to relate the variations to those of AABW formation, although not clearly found. As a whole, it seems that the manuscript is worthy of publication in the journal, but after a moderate revision. A few major comments are stated in the followings, and the minor ones are stated in the specific comments.

Response: we are thankful for the quick answer provided by the reviewer. The concerns of the reviewer have been answered here after and have been valuable help to upgrade the manuscript.

In this paper, temporal variability of anthropogenic CO2 is examined using historical data collected at OISO. The data have been quality controlled by some data synthesis activities such as GLODAP. Nevertheless, I have a question on this point; the data syntheses have been done with a purpose of obtaining data consistency of a basin-scale. By contrast, the authors examine temporal variability of a local scale. In addition, data consistency is usually confirmed by data in deep layers of > 2000 m. This paper deals with data in deep layers. From these points, it is necessary to show that results obtained in the present study is not influenced by the data synthesis. Furthermore, for the recent data, quality control is made independently. Is there any possibility that the Cant stability is caused by the quality control? I recommend the authors to conduct quality-control on OISO data independently.

Response: The reviewer is correct. For most of the ocean basins, data consistency is generally based on data in deep layers (> 1500 or 2000 m). However, because in the Southern Ocean anthropogenic CO2 is also found at depth (> 3000 m), comparison is investigated in "old" deep waters, say around 2000-3000m (LCDW) where Cant (and DIC) should be relatively stable from one year to the next (within error of measurements, 1-3  $\mu$ mol.kg-1). Following the reviewer's recommendation, we propose to add
a figure in Supplement Material (Fig. S1) showing the consistency of our dataset at the two OISO stations where samples were collected down to the bottom, the OISO-ST11 presented in the manuscript and the OISO-ST17 sampled in the Subtropical Zone (30° S-66° E). This figure shows a limited number of measurements that are out of the range of tolerance, but one has to keep in mind that interannual (or multiannual) variations may occur and this calls for great care before applying an adjustment. Since 1987 (when the cruise INDIGO3 was performed), a shift in AT is suggested at high latitudes by the comparisons of INDIGO3 data (unadjusted, following the GLODAPv1 and CARINA recommendations) with other cruises data (adjusted, following the GLO-DAPv2 recommendations). This comparison shows differences that range between -4  $\mu$ mol.kg-1 and +10  $\mu$ mol.kg-1 (Fig. S2). Most of the crossovers that suggest a positive offset for INDIGO3 data (between +6  $\mu$ mol.kg-1 and +10  $\mu$ mol.kg-1) are found south of 60°S, suggesting that AT may have decreased in deep waters at high latitudes since 1987. This is why we first decided for no adjustment in the submitted manuscript (as in the GLODAPv1 and CARINA data products, whereas the INDIGO3 data in GLO-DAPv2 were corrected by -8  $\mu$ mol.kg-1). However, at the OISO-ST11, AT data from the INDIGO3 cruise are also about 8  $\mu$ mol.kg-1 higher than the mean value in deep waters (2000-3000m), in good agreement with the other crossovers at high latitudes. In order to reduce the potential bias that could result from either over-adjusting the data (GLODAPv2 recommendation) or not adjusting the data (GLODAPv1 and CARINA recommendations), and because most of the crossovers at mid-latitudes suggest a small positive offset, we propose to apply an intermediate adjustment of -4  $\mu$ mol.kg-1 in the revised manuscript (the impact on Cant is +2  $\mu$ mol.kg-1). The uncertainty regarding this adjustment will be discussed in Supplement Material. Fig. 3 (before Fig. 4) presenting the interannual variability of LAABW properties and Table 2 presenting the calculated trends will be adjusted correspondingly. Fig. S2 will be completed by the list of the cruises presented. Figure S1 also shows that the low AT values between late 1998 and 2004 are found both in the Antarctic zone and the Subtropical zone. This is surprising, but there are no reason to believe that the data are biased since CMRs

**OSD**
were used for all OISO cruises, and the instrument and data processing were the same during the first OISO cruise in January/February 1998 (showing AT values close to the mean in Fig. S1) and the following cruises.

In discussion, the authors attempt to relate variations of anthropogenic CO2 in AABW to changes in AABW formation region. It is well discussed, but information of water mass age of AABW is lacking. It is necessary to show that linkages between variations of AABW formation region and observed AABW signals at OISO are appropriate in terms of water mass age. O2 and AOU are used simultaneously. I think, it is enough for one of which, probably AOU.

Response: we are sorry that there is no measurement related to water mass age in the available data (I.e. no CFCs measured during OISO cruises), other than O2 which is too sensitive to biological activity to be used as a water mass age tracer. We agree that the mention of both O2 and AOU is unnecessary. This is a point also noticed by Reviewer 2. Because we are most discussing the O2 concentration in the manuscript, we suggest to only present O2 in Figure 3.

Specific comments:

Line 18: "from about +7  $\mu$ mol kg-1", increase from what?

Response: We guess that what confused the two referees is the positive sign. We will delete the positive sign and rephrase as follows: 'from the average concentration of 7  $\mu$ mol.kg-1 calculated for the period 1978-1987 to the averaged concentration of 13  $\mu$ mol.kg-1 in the period 2010-2018.'

Line 20: "CT", this is the first appearance in the abstract. Write it in full.

Response: this will be added.

Line 23: "ðİIJČ, S", they are the first appearance in the abstract. Write them in full.

Response: this will be added.
Lines 90-91: "station 430", depth?

Response: the depth (4710 m) will be added.

Line 91: "405 km and 465 km", away from where?

Response: These are the distance away from the OISO-ST11 sampling site. This will be rephrased as "located near the OISO-ST11 sampling site (405 km and 465 km away from it, respectively)"

Line 109: "the PET sector", is it usually used? I do not understand where it is.

Response: A short sentence will be added to the text, as well as the references mentioned here after to clarify the use of this name. The PET, Princess Elizabeth Though, is also referred as the Balleny Though in Orsi et al. (1999), even if more currently mentioned as PET. It corresponds to the ocean section separating the Kerguelen Plateau from the Antarctic continent. Its deepest point is 3750 m, deep enough to allow AABWs to flow between the Australian Antarctic Basin and the Enderby Basin (Heywood et al., 1999). The work of Heywood et al. (1999; Fig. 1) revealed that in the northern part of the PET the AABW flow from west to east, while in the southern part the flow is from east to west.

Line 150: "AT", Probably this is the first appearance. Spell out here.

Response: this will be added.

Line 160: "ðİIJČ and S", spell out here. /

Response: this will be added.

Lines 163-165: according the description, it seems that the figures are not accuracy but repeatability.

Response: The referee is correct. The accuracy is given by the analysis of CRMs. This will be corrected.
Line 236: "January", which year. In this paper, all the data are analyzed assuming that seasonal variations in deep waters are negligible (lines 154-156). It is not appropriate to refer to months.

Response: the authors agree with the reviewer. This will be adjusted by mentioning the early and late 1998 sampling.

Line 276: "underlying", do you mean a water mass below AABW?

Response: this is a mistake, we meant overlying the AABW (referring to LCDW). This will be corrected by using 'LCDW' instead.
Figure 2. Hovmöller section of (a) Cant via TrOCA, (b) CT, (c) O2, (d) AT, (c) 0 and (f) S based on the OISO data presented in Table 1. Data points are represented by the black dots. The white isolines represent the water masses separation by ya (from the bottom: LAABW, UAABW and LCDW), Figure produced with DV (Schültzer et al., 2019).
Figure S1. Mean differences observed in deep vaters ( $\Omega_1$  minimum) between the measurements obtained during one encircules and the mean value calculated over the full period in the two sites where samples vere calculated down to the OISO ST111 in the Antarctic Zame (S42: S43: S43: E, station investigated in this study, in black) and the OISO St110 ST111 in the Antarctic Zame (S42: S43: S43: S43). The dashed lines indicate the limits for considering an adjustment (as defined in the CARINA and CLODAP syntheses). The data plotted here are adjusted as recommended in the CLODAP syntheses (see The data pointed here the commended of the CARINA and CLODAP syntheses). The data plotted here are adjusted as recommended in the GLODAP synthesis (see The data plotted here are adjusted as recommended in the GLODAP syntheses).
Fig. 2. Figure S1. Quality control of the bottom OISO measurements
Interactive

comment

Fig. 3. Figure S2. Crossovers of the INDIGO3 AT measurements

---

## Author Comment (AC2) · 5 Aug 2020

General comments

The study presents results from a time series in the Indian sector of the Southern Ocean, which together with historical relevant data span a 40-year period. Using this time series, the authors evaluate the evolution of anthropogenic CO2 (Cant) in the Antarctic Bottom Waters (AABW). It is an interesting and generally well written work, and generally good figures and tables. There are some need for clarity in some parts and there is some concern of the treatment of data gaps, but most of this should be rather easily dealt with, and I recommend publication after minor revision. A detailed

list of comments follows below.

Response: The authors are thankful for the fast answer and the positive interest given to the manuscript, as well as for the numerous valuable comments.

My main comments are related to the definition and subsequent presentation of AABW, and, the data gap between 1987 and 1998 and how this is handled and presented. To start with the definition of AABW, this is not an issue in itself, since the denser definition has been used before, and also, since almost any definition can be accepted as long as it is clearly presented. The latter is the problem here, at least for someone not as familiar with the area and these water masses (I usually work in the high-northern latitudes). The definition and choice is clearly described in 2.3, but, then the reader is referred to Fig. 3, where AABW is noted in the layer above the focus of this study, while the data evaluated is in the layer annotated "Considered data". When then the results of the property evolution of AABW are further presented in Fig. 4, at least I got somewhat confused. Whether this is only me or not, this may call for some added clarity. I would suggest to annotate your AABW layer (hence at neutral density >28.35) as AABW (or AABW* or similar), to make this clear, and then make a distinction with the more common AABW.

Response: the authors understand the concern of the reviewer. To solve this potential confusion, we suggest labelling the AABW as define in our manuscript (neutral density >28.35 kg.m-3) Lower Antarctic Bottom Water (LAABW).

Nevertheless, this mostly refers to Fig. 3, and I have several concerns with this figure, as detailed below. Hovmöller plot is a wonderful thing, and can be very illustrative. However, it can also be deceiving, especially when there are gaps in the data, and the gridding is allowed to interpolate over these gaps, which often can create features that give a false picture of actual evolution. Fig. 3 suffers from this when plotting the older data (1978–1987) together with the OISO time-series data starting from 1998. There are several peculiar features in Fig. 3, especially for Cant and AT. The fact that most of

the other plotted parameters show overall stable layer properties, over the full period, may seem to reduce this concern, but I am not convinced. In addition, I'm not fully convinced about the benefit of showing depths from 1500 m, when almost all results and discussion are concerned with the layer below 4000 m. Even more so when the upper layers seems to show most of the strange features, for example the minimum in Cant in the older data (which may in part show the issue with the TrOCA method, with even negative concentrations, which are not realistic, in the most upper part of the deep waters).

Response: The authors agree that the figure needs to be upgraded, clarified and simplified. The suggestions of the referee have been taken into account by redrawing the Fig. 3 (now Fig. 2) using only the OISO data (from 1998 to 2018). The extrapolations were very misleading indeed, so the figure is now drawn with weighted-average gridding (and limited extrapolation around the data point). The aim of this figure is to show the differences in AABW and LCDW characteristics before focusing on the variability and trends observed in the bottom layer (it also shows that the neutral density 28.35 is a better definition for a more homogeneous bottom layer that we now define as LAABW). In addition, the control quality of the data is performed in the old deep waters (well characterized in the figure by the maximum in CT). Following the recommendation from the other Referee, we propose to add a figure in Supplement Material (Fig. S1) showing the consistency of our dataset at the two OISO stations where samples were collected down to the bottom, the OISO-ST11 presented in the manuscript and the OISO-ST17 sampled in the Subtropical Zone (30° S-66° E). This figure shows a limited number of measurements that are out of the range of tolerance, but one has to keep in mind that interannual (or multiannual) variations may occur and this calls for great care before applying an adjustment. Since 1987 (when the cruise INDIGO3 was performed), a shift in AT is suggested at high latitudes by the comparisons of INDIGO3 data (unadjusted, following the GLODAPv1 and CARINA recommendations) with other cruises data (adjusted, following the GLODAPv2 recommendations). This comparison shows differences that range between -4 $\mu$mol.kg-1 and +10 $\mu$mol.kg-1 (Fig. S2).

[Figure]

Most of the crossovers that suggest a positive offset for INDIGO3 data (between +6 $\mu$mol.kg-1 and +10 $\mu$mol.kg-1) are found south of 60°S, suggesting that AT may have decreased in deep waters at high latitudes since 1987. This is why we first decided for no adjustment in the submitted manuscript (as in the GLODAPv1 and CARINA data products, whereas the INDIGO3 data in GLODAPv2 were corrected by -8 $\mu$mol.kg-1). However, at the OISO-ST11, AT data from the INDIGO3 cruise are also about 8 $\mu$mol.kg-1 higher than the mean value in deep waters (2000-3000m), in good agreement with the other crossovers at high latitudes. In order to reduce the potential bias that could result from either over-adjusting the data (GLODAPv2 recommendation) or not adjusting the data (GLODAPv1 and CARINA recommendation), and because most of the crossovers at mid-latitudes suggest a small positive offset, we propose to apply an intermediate adjustment of -4 $\mu$mol.kg-1 in the revised manuscript (the impact on Cant is +2 $\mu$mol.kg-1). The Fig. 3 (before Fig. 4) presenting the interannual variability of the LAABW properties and the Table 2 presenting the calculated trends will be adjusted correspondingly. Fig. S2 will be completed by the list of the cruises presented. The Figure S1 also shows that the low AT values between late 1998 and 2004 are found both in the Antarctic zone and the Subtropical zone. This is surprising, but there are no reason to believe that the data are biased since CMRs were used for all OISO cruises, and the instrument and data processing were the same during the first OISO cruise in January/February 1998 (showing AT values close to the mean in Fig. S1) and the following cruises.

The interpolation of this minimum patch leads to unfortunate wordings in the results, such as on line 236, with "a sudden increase. . . between January and December 1998" seems to refer to the low values calculated for the 1987 data and the clearly higher concentrations calculated for the OISO data. (I also don't really understand the "between Jan and Dec 1998" part, since the first OISO data were sampled in Feb 1998, and the next in Dec the same year.) Apparently there are some need for clarifications here, but also to be cautious when interpreting interpolated values over large gaps. One way to solve this is of course to exclude the older data from the Hovmöller plots.

These can still be used in the comparison/evaluation, and included in Fig. 4.

Response: the reviewer is right about the issue for the 1998 samplings mentioned (same as Reviewer 1). This is because the first OISO cruise started in January 1998, but the station 11 was actually sampled in the beginning of February as mentioned in Table 1. This will be corrected. We also agree that extrapolation can be misleading and we thank the Reviewer for pointing this issue. Having removed the GEOSECS and INDIGO data from the Hovmöller plots (Fig 2., before was Fig. 3), the extrapolation is no more an issue for interpreting the signal observed for the first OISO cruises, but the increase in Cant between February 1998 and December 1998 remains (from < 6 $\mu$mol.kg-1 to about 10 $\mu$mol.kg-1).

To continue on this figure (Fig. 3), for the bottom layer, the fact that it is stretched below the deepest samples seems to create at least the distinct maximum in mid-2000s. Perhaps this will be reduced if the maximum depth/pressure is set to the deepest sample, to exclude extrapolations below that depth.

Response: Having removed the INDIGO1 data from the Hovmöller plots, this no more an issue because the deepest sample is collected at the same depth for all cruises

Specific comments

L18: Do the changes here (+7 and +13, respectively) refer to the whole period? Please clarify.

Response: these are not changes, but Cant concentrations. The following rephrasing is suggested: 'from the average concentration of 7 $\mu$mol.kg-1 calculated for the period 1978-1987 to the average concentration of 13 $\mu$mol.kg-1 for the period 2010-2018.'

L23: A rather tiny remark, but the use of "pluriannual" may be grammatically correct (I'm not a native English speaker), but consider using "multiannual" (or multi-annual), which are more common (I believe). The same is used on L360.

Response: We agree that this is maybe not the best word to use. It will be replaced by

'multi-annual'.

L59: I'm expecting a reference in the end of this sentence. This may be refer to the reference in the previous line, but you may consider moving this to the end.

Response: We agree that the reference is misplaced. It will be moved to the end of the sentence.

L95: I can't find a definition of "AAC" anywhere. Please write out and define the first time.

Response: We agree that the definition of ACC is missing (Antarctic Circumpolar Current). That will be corrected.

L96-97: Unclear sentence. Need some rephrasing/re-writing. Suggestion: ". . .Weddell Sea, where deep and bottom waters are produced. . .".

Response: The sentence will be rephrased as suggested.

L98-100: In the same sentence, there are several instances where the full water mass name is not spelled out, for example "the Ross Sea (RSBW; . . .". This may be intuitive, but I don't think the full names of some of these are written out at any place in the manuscript so would suggest to consider doing that at some place.

Response: The full names will be added explicitly.

L100: Rephrase: In the Prytz Bay, AABW formation has also. . . This sentence is overall quite unclear, especially the last part, so please consider rewriting for clarification.

Response: It is indeed quite unclear. We propose the following rewriting : 'AABW formation has also been observed in the Prydz Bay (Rodehacke et al., 2007; Yabuki et al., 2006). There, three polynyas and two ice shelves have been identified as Prydz Bay Bottom Water (PBBW) production hotspots from seal tagging data (Williams et al., 2016). This PPBW flows out the Prydz Bay through the Prydz Channel and get mixed with the CDBW.'

L105: The "Warm Deep Water" is not described, so not easy to follow without a previous knowledge of the area and the present water masses. Please clarify.

Response: we agree that it may be difficult to follow. The Warm Deep Water is slightly modified Circumpolar Deep Water (by mixing with surface waters when it enters the Weddell Basin). For simplification, we suggest rewriting as follows: The exported WSDW originates from the Circumpolar Deep Water (CDW) that enters the Weddell basin and mixes with WSBW and High Salinity Surface Water (HSSW) (see Fig.2 in van Heuven et al., 2011).

Section 2.4: Part of this section, and in particular from L133, deals with results of Cant from the methods not yet described. I would suggest to move this to the Result section, at least the Cant parts, or maybe part of the Discussion.

Response: the authors agree that this section does not fit in the material and method part of the manuscript, but rather in the discussion section as suggested.

L152: Since the "P" in GLODAP refers to "Project", the "project" after should be avoided (I think). You could rephrase this into something like: . . ..not yet qualified (or included in) the most recent GLODAPv2 product.

Response: the mention of GLODAP will be rephrased as suggested.

L161: The stated accuracy for temperature and salinity seems too low. The standard CTD accuracy, for example found at the GO-SHIP home page (Hydro-manual) is 0.002 for both. Please check.

Response: the authors agree and will correct the accuracy for temperature (0.002°C) and salinity (0.005 for measurements using a salinometer).

L161: As far as I can see, this is the first time "AT" is mentioned, but not defined. Please add this. L166: Same for "O2" as for AT above. Please define first time.

Response: AT and O2 will be defined here.

L170: You mean "onshore"?

Response: the reviewer is right about this mistake.

L184: Clarify which "Redfield ratio". You mean the C:O ratio? Please add this.

Response: Initially, only $C/O_2$ and $N/O_2$ ratios were involved in the definition of the parameter 'a' (Touratier and Goyet, 2004b; Lo Monaco et al., 2005b). In the latest definition of the method Touratier et al. (2007) presents an upgraded definition of this parameter by combining the Redfield equation coefficients for $CO_2$, $O_2$, $HPO_4^{2-}$ and $H^+$ and the same rules of construction as Broecker (1974) did for tracers NO or PO. Because we want to keep the explanation simple in the manuscript, we suggest to rephrase L184 as follows : 'where a is defined in Touratier et al. (2007) as combination of the Redfield equation coefficients for $CO_2$, $O_2$, $HPO_4^{2-}$ and $H^+$. For more details about the definition and the calibration of this parameter, please refer to Touratier et al. (2007).'

L217: Either remove "after", so it reads ". . . and only impacted by. . .", or if more correct, add "subduction", so it reads "and after subduction only impacted by. . .".

Response: the word 'subduction' will be added as suggested.

L233: "LCBW" is here mentioned for the first time, without definition or any description anywhere in the manuscript, as far as I can see. Please add this.

Response: LCDW refers to the Lower Circumpolar Deep Water laying above AABW in the entire Southern Ocean. Details about this water mass will be added in Section 2.2 where it is first mentioned.

L235-236: This is what was commented on in the generall comments above, with the "sudden increase". Please revise and clarify. It is more likely that there was a more gradual evolution, and none of the other parameters calls for any sudden changes. Also, the data quality and methods between the older data and the OISO data may differ, so extra causion is taken when comparing them.

Response: we removed the older data form the Hovmoller plot, but the change in Cant in LCDW remains (from <6 $\mu$mol.kg-1 in Feb 1998 (similar as for the older data) to about 10 $\mu$mol.kg-1 for the following cruises).

L240: The maximum in Cant in 2004 is one occasion, and followed by five (almost six) years without any data. I would be cautious to over interpret this. However, it co-incides with a maximum in oxygen, which could indicate a ventilation event.

Response: we agree with the referee about being cautious with the measurements in 2004. Indeed the maximum in Cant is due to the maximum in O2 (not associated with a maximum in CT).

L256-260: The lower concentrations of AT in the years around 2000 at all depths below (at least) 1500 m (have you checked the whole water column?) seems a bit odd. Especially when this is not seen in any of the other parameters. Also, when comparing two years in the 1980s with data more than a decade later, one should be extra cautious in the interpretation, not the least when the two years/occasions in 1985/87 show the highest concentrations seen over the evaluated period. Certainly the years after 2000 show much lower concentrations, which may be a phase due to a change in different forcing, but to suggest reduced calcification from only a few years/occupations of data is very speculative, and clearly something that change a few years later.

Response: As mentioned in the general comments, the low AT values between late 1998 and 2004 are found both in the Antarctic zone and the Subtropical zone (Figure S1), but they are not observed in the surface layer (this will be added in the revised manuscript). The hypothesis about reduced calcification could explain this contrast between the surface waters and the deep ocean.

L259-260: Is it realistic that the increase in CT is lower than the accumulation of Cant?

Response: The small increase in CT over the period 1987-2004 could be caused by a reduction in CT,nat around the year 2000 (associated with the low AT values). This

said, we also have to keep in mind the uncertainty on the Cant calculations. This will be clarified in the results and in the discussion.

L261: While there is a rather clear trend in oxygen during this period – although I would be careful in talking about trends over such short periods, especially when comparing to a year with a maximum (2004) – there is no trend in Cant. Instead the latter shows some clear interannual variability. Also, the "trend" in temperature is indeed very small, and even if not significant, the change, or better, variability, in salinity is rather large. Consider these points when revising this part. Your statement on L267-268 highlights this issue.

Response: we agree with the reviewer that there is no clear trend in Cant over 2004-2018. We will change "decrease in Cant" for "no increase in Cant". The same is true for temperature and salinity.

L270-271: There is also a maximum in temperature in 1985, so this could indicate more mixing with WSDW, which are both fresher and warmer.

Response: we agree with the reviewer that more mixing with WSDW (or CDBW) could also explain the higher Cant concentrations and lower S in 1985 (the signal in temperature is not well marked due to the large error bars). This will be added in the text.

L275-278: This is a very long sentence. I suggest to divide it, with period after ". . .the underlying deep waters." Then remove "and", and start on "Since", or change the start of the sentence. For the last part of this sentence (L277-278), the suggestion of increased contribution from the Ross Sea is not clear to me since the oxygen decrease, while the salinity goes up and down. Or are you only referring to the one occupation in 2012? (If this is the case, it seems to detailed to explain a single year taken out of a long time series.)

Response: the suggestion made by the reviewer to shorten the sentence will be used. Our aim is to discuss the variability in Cant concentrations that could reflect variations

in the contribution of different types of AABWs. We suggest that the lower Cant concentrations observed in 2011, 2012 and 2013 may be due to an increased contribution of older types of AABW. We agree that pointing to RSBW as a possible candidate because salinity was higher in 2012 is too speculative. This will be removed.

L280: The stated freshening of 0.01, for which period is that observed? Please clarify.

Response: The sentence will be corrected as follows: 'The freshening in S of -0.006 decade-1 between 2004 and 2018 that we observed on the Western side of the Kerguelen Plateau was also observed on the Eastern side of the Plateau by Menezes et al. (2017) over a similar period.'

L312-313: ". . .(15 umol kg-1) due to mixing with older CDW."

Response: the sentence will be corrected.

L317: "that contain very high amounts of Cant . . ."

Response: the sentence will be correct as suggested.

L318-320: The last sentence of this paragraph basically repeats what have been said above. Consider to remove.

Response: the authors agree with the reviewer and will remove this sentence.

L325: Here you write out "Southern Ocean" after having used the abbreviation throughout the manuscript, even the sentence before. Consider to revise.

Response: "Southern Ocean" will be changed to SO.

L340: "evaluated" should here instead be "estimated", or "calculated", or "found" (I think).

Response: "evaluated" will be replaced by "calculated".

L386: Consider rewording ". . .vary in a very large range. . .". Suggestion: "show a very large variability", or maybe, "vary over a very large range".

Response: the rewording 'vary over a very large range' will be used.

L387-388: "(–221 mmol C m-2 d-1; Roden et al., 2016).

Response: we will correct this according to the reviewer suggestion.

L416: Both these water masses (RSBW and ALBW) have higher salinity, and while oxygen show a reduced trend the salinity goes up and down, so this explanation does not hold for all years during this period.

Response: we understand the concern of the reviewer. The mention of the WSDW will be added, as for the response of the comment L275-278.

L424: "explains most, but not all, of the observed. . ."

Response: the sentence will be corrected.

L463: GLODAPv2 version are written as "GLODAPv2.2021 (.2020 is soon to be released). You do mean 2021 and not 2020?

Response: the data will not be included in GLODAP in the 2020 version, but in the following one.

L851-853: Table 2 (and in general): You may want to consider if you want to keep AOU as parameter, when you mostly refer to oxygen. The trends are almost exactly thesame (but opposite of course), and gives the same message.

Response: we agree with the reviewer. AOU will be removed from Table 2 and from Figure 2 (and from the corresponding parts in the text).

Technical comments

L22: This is, however, modulated. . .

Response: the comas will be added.

L35: The references should, typically, be chronologically ordered. Please check
throughout the manuscript. (There are more examples of this, but I won't comment on this more.)

Response: we agree with the referee. We will check for other occurrences.

L71: This is, however, not the. . .

Response: the comas will be added.

L91: ". . .(405 and 465 km, respectively)."

Response: the coma will be added.

L107-113: Examplified with ". . .East of the Kerguelen. . .", this section has many of these "directions/locations" (east/west/. . .) spelled with a large letter, even not part of a name. I think this is not correct, and if so, please change.

Response: this will be corrected.

L118-119: . . .28.27-bottom, respectively. . .

Response: the coma will be added.

L172-173: Change font; the part of the sentence from "for deep samples. . ." are in a different font (maybe "Cambria").

Response: the font will be changed.

L220: Change font for "value for".

Response: the font will be changed.

L306: Add a comma: "2018 (Fig 3a), probably . . ."

Response: the coma will be added.

L340: Add a ".": Pardo et al. (2017)

Response: the dot will be added.

L347: For consistency, change "South-Western" to "South-western" (similar as on L325).

Response: this will be corrected.

L449: Remove "." for consistency: (e.g. Frölicher et al., 2014).

Response: the coma will be deleted.

L451: References in chronological order.

Response: we agree with the referee. This will be corrected.

―――――――――――――――――

[Figure]

Figure 2. Hovmöller section of (a) Cant via TrOCA, (b) CT, (c) O2, (d) AT, (e) θ and (f) S based on the OISO data presented in Table 1. Data points are represented by the black dots. The white isolines represent the water masses separation by γn (from the bottom: LAABW, UAABW and LCDW). Figure produced with ODV (Schlitzer et al., 2019).

**Fig. 1.** Figure 2. Hovmoller section

Figure S1. Mean differences observed in deep waters (O₂ minimum) between the measurements obtained during one cruise and the mean value calculated over the full period at the two sites where samples were collected down to the bottom: the OISO-ST11 in the Antarctic Zone (56.5° S-63° E, station investigated in this study, in black) and the OISO station 17 in the Subtropical Zone (30° S-66° E, in gray). The dashed lines indicate the limits for considering an adjustment (as defined in the CARINA and GLODAP syntheses). The data plotted here are adjusted as recommended in the GLODAPv2 synthesis, except for $A_T$ in 1987 (INDIGO3 cruise) that was adjusted by -4 µmol.kg⁻¹.

**Fig. 2.** Figure S1. Quality control of the bottom OISO measurements

Figure S2. Difference in $A_T$ found in deep waters at crossovers with the INDIGO3 cruise (INDIGO3 - the other cruise). Data are adjusted as recommended in the GLODAPv2 synthesis except for the INDIGO3 cruise for which $A_T$ was not adjusted.

**Fig. 3.** Figure S2. Crossovers of the INDIGO3 AT measurements

---

## Author Response (AR1)

**1 Comments Referee 1**

General comments: This manuscript deals with temporal variations of anthropogenic CO2 in bottom 2 waters in the Southern Ocean. The Southern Ocean is said to take up 40% of anthropogenic CO2 3 4 absorbed by the ocean. Thus, investigations of temporal variability of anthropogenic CO2 are very 5 important to evaluate ocean's capacity of absorbing atmospheric CO2, information of which is 6 indispensable for the projection of global warming. In terms of oceanic observation, the Southern 7 Ocean is one of the regions, where the number of measurements, especially for chemical and biological properties, is scare. In this point also, it is worth of being published in the journal. The manuscript is 8 9 well organized, and is easy to read. The approaches used in the study are not new, but traditional ones. 10 It is not a problem. It would be necessary to adopt an approach, which has been demonstrated to be 11 useful for the detection of small signals of anthropogenic CO2 variations. The authors attempt also to 12 relate the variations to those of AABW formation, although not clearly found. As a whole, it seems that the manuscript is worthy of publication in the journal, but after a moderate revision. A few major 13 14 comments are stated in the followings, and the minor ones are stated in the specific comments.

**15 Response: we are thankful for the quick answer provided by the reviewer. The concerns of the 16 reviewer have been answered here after and have been valuable help to upgrade the manuscript.**

In this paper, temporal variability of anthropogenic CO2 is examined using historical data collected at 18 OISO. The data have been quality controlled by some data synthesis activities such as GLODAP. 19 Nevertheless, I have a question on this point; the data syntheses have been done with a purpose of obtaining data consistency of a basin-scale. By contrast, the authors examine temporal variability of a 20 21 local scale. In addition, data consistency is usually confirmed by data in deep layers of > 2000 m. This 22 paper deals with data in deep layers. From these points, it is necessary to show that results obtained 23 in the present study is not influenced by the data synthesis. Furthermore, for the recent data, quality control is made independently. Is there any possibility that the Cant stability is caused by the quality 24 25 control? I recommend the authors to conduct quality-control on OISO data independently.

Response: The reviewer is correct. For most of the ocean basins, data consistency is generally based 26 27 on data in deep layers (> 1500 or 2000 m). However, because in the Southern Ocean anthropogenic 28 CO2 is also found at depth (> 3000 m), comparison is investigated in "old" deep waters, say around 29 2000-3000m (LCDW) where Cant (and DIC) should be relatively stable from one year to the next (within error of measurements, 1-3 µmol.kg-1). Following the reviewer's recommendation, we 30 31 propose to add a figure in Supplement Material (Fig. S1) showing the consistency of our dataset at 32 the two OISO stations where samples were collected down to the bottom, the OISO-ST11 presented 33 in the manuscript and the OISO-ST17 sampled in the Subtropical Zone (30° S-66° E). This figure shows a limited number of measurements that are out of the range of tolerance, but one has to keep in 34 35 mind that interannual (or multiannual) variations may occur and this calls for great care before 36 applying an adjustment.

Since 1987 (when the cruise INDIGO3 was performed), a shift in  $A_T$  is suggested at high latitudes by the comparisons of INDIGO3 data (unadjusted, following the GLODAPv1 and CARINA 38 39 recommendations) with other cruises data (adjusted, following the GLODAPv2 recommendations). 40 This comparison shows differences that range between -4 µmol.kg-1 and +10 µmol.kg-1 (Fig. S2). Most of the crossovers that suggest a positive offset for INDIGO3 data (between +6 µmol.kg-1 and +10 41 42  $\mu$ mol.kg-1) are found south of 60°S, suggesting that AT may have decreased in deep waters at high 43 latitudes since 1987. This is why we first decided for no adjustment in the submitted manuscript (as 44 in the GLODAPv1 and CARINA data products, whereas the INDIGO3 data in GLODAPv2 were corrected by -8 μmol.kg-1). However, at the OISO-ST11, AT data from the INDIGO3 cruise are also 45 46 about 8 µmol.kg-1 higher than the mean value in deep waters (2000-3000m), in good agreement with 47 the other crossovers at high latitudes. In order to reduce the potential bias that could result from 48 either over-adjusting the data (GLODAPv2 recommendation) or not adjusting the data (GLODAPv1 49 and CARINA recommendations), and because most of the crossovers at mid-latitudes suggest a small positive offset, we propose to apply an intermediate adjustment of -4 µmol.kg-1 in the revised 50 manuscript (the impact on  $C_{ant}$  is +2 µmol.kg-1). The uncertainty regarding this adjustment will be 51 52 discussed in Supplement Material. Fig. 3 (before Fig. 4) presenting the interannual variability of 53 LAABW properties and Table 2 presenting the calculated trends will be adjusted correspondingly.

Figure S1 also shows that the low  $A_T$  values between late 1998 and 2004 are found both in the Antarctic zone and the Subtropical zone. This is surprising, but there are no reason to believe that the data are biased since CMRs were used for all OISO cruises, and the instrument and data processing were the same during the first OISO cruise in January/February 1998 (showing  $A_T$  values close to the mean in Fig. S1) and the following cruises.

In discussion, the authors attempt to relate variations of anthropogenic CO2 in AABW to changes in AABW formation region. It is well discussed, but information of water mass age of AABW is lacking. It is necessary to show that linkages between variations of AABW formation region and observed AABW signals at OISO are appropriate in terms of water mass age. O2 and AOU are used simultaneously. I think, it is enough for one of which, probably AOU.

Response: we are sorry that there is no measurement related to water mass age in the available data (I.e. no CFCs measured during OISO cruises), other than  $O_2$  which is too sensitive to biological activity to be used as a water mass age tracer. We agree that the mention of both  $O_2$  and AOU is unnecessary. This is a point also noticed by Reviewer 2. Because we are most discussing the  $O_2$

- 68 concentration in the manuscript, we suggest to only present O2 in Figure 3.
- 69 Specific comments:
- 70 Line 18 (here around 460): "from about +7 μmol kg-1", increase from what?

71Response: We guess that what confused the two referees is the positive sign. We will delete the72positive sign and rephrase as follows: 'from the average concentration of 7  $\mu$ mol.kg-1 calculated for73the period 1978-1987 to the averaged concentration of 13  $\mu$ mol.kg-1 in the period 2010-2018.'

- Line 20 (here around 463): "CT", this is the first appearance in the abstract. Write it in full.
- 76 **Response: this will be added.**
- 77

Line 23 (here around 467): " $\theta$ , S", they are the first appearance in the abstract. Write them in full.

- 79 Response: this will be added.
- 81 Lines 90-91 (here around 535-536): "station 430", depth?
- 82 Response: the depth (4710 m) will be added.
- Line 91 (here around 535): "405 km and 465 km", away from where?
- Response: These are the distance away from the OISO-ST11 sampling site. This will be rephrased as
   "located near the OISO-ST11 sampling site (405 km and 465 km away from it, respectively)"
- Line 109 (here around 556): "the PET sector", is it usually used? I do not understand where it is.

Response: A short sentence will be added to the text, as well as the references mentioned here after to clarify the use of this name. The PET, Princess Elizabeth Though, is also referred as the Balleny Though in Orsi et al. (1999), even if more currently mentioned as PET. It corresponds to the ocean section separating the Kerguelen Plateau from the Antarctic continent. Its deepest point is 3750 m, deep enough to allow AABWs to flow between the Australian Antarctic Basin and the Enderby Basin (Heywood et al., 1999). The work of Heywood et al. (1999; Fig. 1) revealed that in the northern part of the PET the AABW flow from west to east, while in the southern part the flow is from east to west. Line 150 (here around 612): "AT", Probably this is the first appearance. Spell out here. Response: this will be added. Line 160 (here around 628): " $\theta$  and S", spell out here. ' Response: this will be added. Lines 163-165 (here around 633-635): according the description, it seems that the figures are not accuracy but repeatability. Response: The referee is correct. The accuracy is given by the analysis of CRMs. This will be corrected. Line 236 (here around 714): "January", which year. In this paper, all the data are analyzed assuming that seasonal variations in deep waters are negligible (lines 154-156). It is not appropriate to refer to months. Response: the authors agree with the reviewer. This will be adjusted by mentioning the early and late 1998 sampling. Line 276 (here around 762): "underlying", do you mean a water mass below AABW? Response: this is a mistake, we meant overlying the AABW (referring to LCDW). This will be corrected by using 'LCDW' instead.

**126 Comments Referee 2**

**127 General comments**

The study presents results from a time series in the Indian sector of the Southern Ocean, which together with historical relevant data span a 40-year period. Using this time series, the authors evaluate the evolution of anthropogenic CO2 (Cant) in the Antarctic Bottom Waters (AABW). It is an interesting and generally well written work, and generally good figures and tables. There are some need for clarity in some parts and there is some concern of the treatment of data gaps, but most of this should be rather easily dealt with, and I recommend publication after minor revision. A detailed list of comments follows below.

**The authors are thankful for the fast answer and the positive interest given to the manuscript, aswell as for the numerous valuable comments.**

My main comments are related to the definition and subsequent presentation of AABW, and, the data 138 gap between 1987 and 1998 and how this is handled and presented. To start with the definition of 139 AABW, this is not an issue in itself, since the denser definition has been used before, and also, since 140 almost any definition can be accepted as long as it is clearly presented. The latter is the problem here, 141 at least for someone not as familiar with the area and these water masses (I usually work in the high-142 northern latitudes). The definition and choice is clearly described in 2.3, but, then the reader is referred to Fig. 3, where AABW is noted in the layer above the focus of this study, while the data evaluated is 143 144 in the layer annotated "Considered data". When then the results of the property evolution of AABW 145 are further presented in Fig. 4, at least I got somewhat confused. Whether this is only me or not, this 146 may call for some added clarity. I would suggest to annotate your AABW layer (hence at neutral density 147 >28.35) as AABW (or AABW\* or similar), to make this clear, and then make a distinction with the more 148 common AABW.

**Response: the authors understand the concern of the reviewer. To solve this potential confusion, we suggest labelling the AABW as define in our manuscript (neutral density >28.35 kg.m-3) Lower Antarctic Bottom Water (LAABW).**

Nevertheless, this mostly refers to Fig. 3, and I have several concerns with this figure, as detailed below. 153 Hovmöller plot is a wonderful thing, and can be very illustrative. However, it can also be deceiving, 154 especially when there are gaps in the data, and the gridding is allowed to interpolate over these gaps, which often can create features that give a false picture of actual evolution. Fig. 3 suffers from this 155 156 when plotting the older data (1978–1987) together with the OISO time-series data starting from 1998. 157 There are several peculiar features in Fig. 3, especially for Cant and AT. The fact that most of the other plotted parameters show overall stable layer properties, over the full period, may seem to reduce this 158 159 concern, but I am not convinced. In addition, I'm not fully convinced about the benefit of showing 160 depths from 1500 m, when almost all results and discussion are concerned with the layer below 4000 161 m. Even more so when the upper layers seems to show most of the strange features, for example the 162 minimum in Cant in the older data (which may in part show the issue with the TrOCA method, with 163 even negative concentrations, which are not realistic, in the most upper part of the deep waters).

Response: The authors agree that the figure needs to be upgraded, clarified and simplified. The suggestions of the referee have been taken into account by redrawing the Fig. 3 (now Fig. 2) using only the OISO data (from 1998 to 2018). The extrapolations were very misleading indeed, so the figure is now drawn with weighted-average gridding (and limited extrapolation around the data point). The aim of this figure is to show the differences in AABW and LCDW characteristics before focusing on the variability and trends observed in the bottom layer (it also shows that the neutral 170 density 28.35 is a better definition for a more homogeneous bottom layer that we now define as 171 LAABW). In addition, the control quality of the data is performed in the old deep waters (well 172 characterized in the figure by the maximum in  $C_T$ ). Following the recommendation from the other 173 Referee, we propose to add a figure in Supplement Material (Fig. S1) showing the consistency of our 174 dataset at the two OISO stations where samples were collected down to the bottom, the OISO-ST11 175 presented in the manuscript and the OISO-ST17 sampled in the Subtropical Zone (30° S-66° E). This 176 figure shows a limited number of measurements that are out of the range of tolerance, but one has 177 to keep in mind that interannual (or multiannual) variations may occur and this calls for great care 178 before applying an adjustment.

Since 1987 (when the cruise INDIGO3 was performed), a shift in AT is suggested at high latitudes by 180 the comparisons of INDIGO3 data (unadjusted, following the GLODAPv1 and CARINA recommendations) with other cruises data (adjusted, following the GLODAPv2 recommendations). 181 182 This comparison shows differences that range between -4 µmol.kg-1 and +10 µmol.kg-1 (Fig. S2). Most 183 of the crossovers that suggest a positive offset for INDIGO3 data (between +6  $\mu$ mol.kg-1 and +10 184  $\mu$ mol.kg-1) are found south of 60°S, suggesting that AT may have decreased in deep waters at high 185 latitudes since 1987. This is why we first decided for no adjustment in the submitted manuscript (as in the GLODAPv1 and CARINA data products, whereas the INDIGO3 data in GLODAPv2 were 186 corrected by -8 μmol.kg-1). However, at the OISO-ST11, AT data from the INDIGO3 cruise are also 187 about 8 µmol.kg-1 higher than the mean value in deep waters (2000-3000m), in good agreement with 188 189 the other crossovers at high latitudes. In order to reduce the potential bias that could result from either over-adjusting the data (GLODAPv2 recommendation) or not adjusting the data (GLODAPv1 190 191 and CARINA recommendation), and because most of the crossovers at mid-latitudes suggest a small 192 positive offset, we propose to apply an intermediate adjustment of -4 µmol.kg-1 in the revised manuscript (the impact on Cant is +2 µmol.kg-1). The Fig. 3 (before Fig. 4) presenting the interannual 193 194 variability of the LAABW properties and the Table 2 presenting the calculated trends will be adjusted correspondingly. Fig. S2 will be completed by the list of the cruises presented. 195

196The Figure S1 also shows that the low  $A_T$  values between late 1998 and 2004 are found both in the197Antarctic zone and the Subtropical zone. This is surprising, but there are no reason to believe that198the data are biased since CMRs were used for all OISO cruises, and the instrument and data199processing were the same during the first OISO cruise in January/February 1998 (showing  $A_T$  values200close to the mean in Fig. S1) and the following cruises.

The interpolation of this minimum patch leads to unfortunate wordings in the results, such as on line 201 236, with "a sudden increase. . . between January and December 1998" seems to refer to the low 202 203 values calculated for the 1987 data and the clearly higher concentrations calculated for the OISO data. 204 (I also don't really understand the "between Jan and Dec 1998" part, since the first OISO data were sampled in Feb 1998, and the next in Dec the same year.) Apparently there are some need for 205 206 clarifications here, but also to be cautious when interpreting interpolated values over large gaps. One 207 way to solve this is of course to exclude the older data from the Hovmöller plots. These can still be 208 used in the comparison/evaluation, and included in Fig. 4.

Response: the reviewer is right about the issue for the 1998 samplings mentioned (same as Reviewer 1). This is because the first OISO cruise started in January 1998, but the station 11 was actually sampled in the beginning of February as mentioned in Table 1. This will be corrected. We also agree that extrapolation can be misleading and we thank the Reviewer for pointing this issue. Having removed the GEOSECS and INDIGO data from the Hovmöller plots (Fig 2., before was Fig. 3), the extrapolation is no more an issue for interpreting the signal observed for the first OISO cruises, but

**samples seems to create at least the distinct maximum in mid-2000s. Perhaps this will be reduced if 219 220 the maximum depth/pressure is set to the deepest sample, to exclude extrapolations below that 221 depth. 222 223 Response: Having removed the INDIGO1 data from the Hovmöller plots, this no more an issue 224 because the deepest sample is collected at the same depth for all cruises 225 226 227 Specific comments 228 L18 (here around 460): Do the changes here (+7 and +13, respectively) refer to the whole period? 229 Please clarify. 230 Response: these are not changes, but Cant concentrations. The following rephrasing is suggested: 231 'from the average concentration of 7 $\mu$ mol.kg-1 calculated for the period 1978-1987 to the average 232 concentration of 13 µmol.kg-1 for the period 2010-2018.' 233 L23 (here around 467): A rather tiny remark, but the use of "pluriannual" may be grammatically correct 234 235 (I'm not a native English speaker), but consider using "multiannual" (or multi-annual), which are more 236 common (I believe). The same is used on L360. 237 Response: We agree that this is maybe not the best word to use. It will be replaced by 'multi-annual'. 238 L59 (here around 502): I'm expecting a reference in the end of this sentence. This may be refer to the 239 reference in the previous line, but you may consider moving this to the end. 240 Response: We agree that the reference is misplaced. It will be moved to the end of the sentence. L95 (here around 541): I can't find a definition of "AAC" anywhere. Please write out and define the first 241 242 time. 243 Response: We agree that the definition of ACC is missing (Antarctic Circumpolar Current). That will 244 be corrected. 245 L96-97 (here around 542-543): Unclear sentence. Need some rephrasing/re-writing. Suggestion: "... .Weddell Sea, where deep and bottom waters are produced. .. ". 246 247 Response: The sentence will be rephrased as suggested. 248 L98-100 (here around 544-546): In the same sentence, there are several instances where the full water 249 mass name is not spelled out, for example "the Ross Sea (RSBW; ... ". This may be intuitive, but I don't 250 think the full names of some of these are written out at any place in the manuscript so would suggest to consider doing that at some place. 251 Response: The full names will be added explicitly. 252 253 L100 (here around 546): Rephrase: In the Prytz Bay, AABW formation has also. . . This sentence is**

the increase in Cant between February 1998 and December 1998 remains (from < 6  $\mu$ mol.kg-1 to

To continue on this figure (Fig. 3), for the bottom layer, the fact that it is stretched below the deepest

- 254 overall quite unclear, especially the last part, so please consider rewriting for clarification.
- Response: It is indeed quite unclear. We propose the following rewriting : 'AABW formation has also been observed in the Prydz Bay (Rodehacke et al., 2007; Yabuki et al., 2006). There, three polynyas

218 about 10 µmol.kg-1).

and two ice shelves have been identified as Prydz Bay Bottom Water (PBBW) production hotspots from seal tagging data (Williams et al., 2016). This PPBW flows out the Prydz Bay through the Prydz 258 259 Channel and get mixed with the CDBW.'

L105 (here around 552): The "Warm Deep Water" is not described, so not easy to follow without a 261 262 previous knowledge of the area and the present water masses. Please clarify.

Response: we agree that it may be difficult to follow. The Warm Deep Water is slightly modified 263 Circumpolar Deep Water (by mixing with surface waters when it enters the Weddell Basin). For 264 simplification, we suggest rewriting as follows: The exported WSDW originates from the 265 266 Circumpolar Deep Water (CDW) that enters the Weddell basin and mixes with WSBW and High 267 Salinity Surface Water (HSSW) (see Fig.2 in van Heuven et al., 2011).

Section 2.4: Part of this section, and in particular from L133, deals with results of Cant from the 270 methods not yet described. I would suggest to move this to the Result section, at least the Cant parts, 271 or maybe part of the Discussion.

Response: the authors agree that this section does not fit in the material and method part of the 273 manuscript, but rather in the discussion section as suggested. 274

L152 (here around 614): Since the "P" in GLODAP refers to "Project", the "project" after should be 275 276 avoided (I think). You could rephrase this into something like: . . . . not yet qualified (or included in) the most recent GLODAPv2 product. 277

**278 Response: the mention of GLODAP will be rephrased as suggested.**

L161 (here around 629): The stated accuracy for temperature and salinity seems too low. The standard 281 CTD accuracy, for example found at the GO-SHIP home page (Hydro-manual) is 0.002 for both. Please 282 check.

**283 Response: the authors agree and will correct the accuracy for temperature (0.002°C) and salinity 284 (0.005 for measurements using a salinometer).**

- L161 (here around 629): As far as I can see, this is the first time "AT" is mentioned, but not defined. 286 287 Please add this.
- 288 L166 (here around 636): Same for "O2" as for AT above. Please define first time.

**289 Response: AT and O2 will be defined here.**

L170 (here around 641): You mean "onshore"?

**292 Response: the reviewer is right about this mistake.**

294 L184 (here around 656): Clarify which "Redfield ratio". You mean the C:O ratio? Please add this.

Response: Initially, only C/O2 and N/O2 ratios were involved in the definition of the parameter 'a' 296 (Touratier and Goyet, 2004b; Lo Monaco et al., 2005b). In the latest definition of the method

Touratier et al. (2007) presents an upgraded definition of this parameter by combining the Redfield 297

equation coefficients for CO2, O2, HPO42- and H+ and the same rules of construction as Broecker (1974) 298

did for tracers NO or PO. Because we want to keep the explanation simple in the manuscript, we suggest to rephrase L184 as follows : 'where a is defined in Touratier et al. (2007) as combination of

**the Redfield equation coefficients for $CO_2$ , $O_2$ , $HPO_4^{2-}$ and $H^+$ . For more details about the definition and the calibration of this parameter, please refer to Touratier et al. (2007).'**

L217 (here around 691): Either remove "after", so it reads "... and only impacted by...", or if more correct, add "subduction", so it reads "and after subduction only impacted by...".

**305 **Response: the word 'subduction' will be added as suggested.**

L233: "LCBW" is here mentioned for the first time, without definition or any description anywhere in
 the manuscript, as far as I can see. Please add this.

**Response: LCDW refers to the Lower Circumpolar Deep Water laying above AABW in the entire Southern Ocean. Details about this water mass will be added in Section 2.2 where it is first mentioned.**

- L235-236: This is what was commented on in the generall comments above, with the "sudden increase". Please revise and clarify. It is more likely that there was a more gradual evolution, and none of the other parameters calls for any sudden changes. Also, the data quality and methods between the older data and the OISO data may differ, so extra causion is taken when comparing them.
- Response: we removed the older data form the Hovmoller plot, but the change in Cant in LCDW
   remains (from <6 μmol.kg-1 in Feb 1998 (similar as for the older data) to about 10 μmol.kg-1 for the
   following cruises).
- 1321 L240 (here around 719): The maximum in Cant in 2004 is one occasion, and followed by five (almost 1322 six) years without any data. I would be cautious to over interpret this. However, it co-incides with a 1323 maximum in oxygen, which could indicate a ventilation event.

**Response: we agree with the referee about being cautious with the measurements in 2004. Indeed the maximum in $C_{ant}$ is due to the maximum in $O_2$ (not associated with a maximum in $C_T$ ).**

L256-260 (here around 739-743): The lower concentrations of AT in the years around 2000 at all depths 328 below (at least) 1500 m (have you checked the whole water column?) seems a bit odd. Especially when this is not seen in any of the other parameters. Also, when comparing two years in the 1980s with data 329 330 more than a decade later, one should be extra cautious in the interpretation, not the least when the 331 two years/occasions in 1985/87 show the highest concentrations seen over the evaluated period. 332 Certainly the years after 2000 show much lower concentrations, which may be a phase due to a change 333 in different forcing, but to suggest reduced calcification from only a few years/occupations of data is 334 very speculative, and clearly something that change a few years later.

Response: As mentioned in the general comments, the low AT values between late 1998 and 2004 are found both in the Antarctic zone and the Subtropical zone (Figure S1), but they are not observed in the surface layer (this will be added in the revised manuscript). The hypothesis about reduced calcification could explain this contrast between the surface waters and the deep ocean.

L259-260 (here around 742-743): Is it realistic that the increase in CT is lower than the accumulationof Cant?

341Response: The small increase in CT over the period 1987-2004 could be caused by a reduction in CT,nat342around the year 2000 (associated with the low AT values). This said, we also have to keep in mind the uncertainty on the Cant calculations. This will be clarified in the results and in the discussion.

L261 (here around 744): While there is a rather clear trend in oxygen during this period – although I would be careful in talking about trends over such short periods, especially when comparing to a year with a maximum (2004) – there is no trend in Cant. Instead the latter shows some clear interannual variability. Also, the "trend" in temperature is indeed very small, and even if not significant, the change, or better, variability, in salinity is rather large. Consider these points when revising this part. Your statement on L267-268 highlights this issue.

- Response: we agree with the reviewer that there is no clear trend in Cant over 2004-2018. We will change "decrease in Cant" for "no increase in Cant". The same is true for temperature and salinity.
- L270-271 (here around 755): There is also a maximum in temperature in 1985, so this could indicate more mixing with WSDW, which are both fresher and warmer.

**Response: we agree with the reviewer that more mixing with WSDW (or CDBW) could also explain the higher Cant concentrations and lower S in 1985 (the signal in temperature is not well marked due to the large error bars). This will be added in the text.**

- L275-278 (here around 761-764): This is a very long sentence. I suggest to divide it, with period after "...the underlying deep waters." Then remove "and", and start on "Since", or change the start of the sentence. For the last part of this sentence (L277-278), the suggestion of increased contribution from the Ross Sea is not clear to me since the oxygen decrease, while the salinity goes up and down. Or are you only referring to the one occupation in 2012? (If this is the case, it seems to detailed to explain a single year taken out of a long time series.)
- Response: the suggestion made by the reviewer to shorten the sentence will be used. Our aim is to discuss the variability in Cant concentrations that could reflect variations in the contribution of different types of AABWs. We suggest that the lower Cant concentrations observed in 2011, 2012 and 2013 may be due to an increased contribution of older types of AABW. We agree that pointing to RSBW as a possible candidate because salinity was higher in 2012 is too speculative. This will be removed.
- 371 L280 (here around 767): The stated freshening of 0.01, for which period is that observed? Please clarify.
- Response: The sentence will be corrected as follows: 'The freshening in S of -0.006 decade-1 between
and 2018 that we observed on the Western side of the Kerguelen Plateau was also observed
  on the Eastern side of the Plateau by Menezes et al. (2017) over a similar period.'
- 375 L312-313 (here around 823): "... (15 umol kg-1) due to mixing with older CDW."
- 376 **Response: the sentence will be corrected.**
- 377 L317 (here around 827): "that contain very high amounts of Cant . . ."
- 378 Response: the sentence will be correct as suggested.
- L318-320 (here around 829-830): The last sentence of this paragraph basically repeats what have beensaid above. Consider to remove.
- 381 Response: the authors agree with the reviewer and will remove this sentence.
- L325 (here around 835): Here you write out "Southern Ocean" after having used the abbreviation
   throughout the manuscript, even the sentence before. Consider to revise.
- 384 **Response: "Southern Ocean" will be changed to SO.**

L340 (here around 845): "evaluated" should here instead be "estimated", or "calculated", or "found"
(I think).

**387 Response: "evaluated" will be replaced by "calculated".**

L386 (here around 898): Consider rewording ". . .vary in a very large range. . .". Suggestion: "show a
 very large variability", or maybe, "vary over a very large range".

**390 Response: the rewording 'vary over a very large range' will be used.**

L387-388 (here around 901): "(-221 mmol C m-2 d-1; Roden et al., 2016).

**392 Response: we will correct this according to the reviewer suggestion.**

- 393 L416 (here around 928): Both these water masses (RSBW and ALBW) have higher salinity, and while
- oxygen show a reduced trend the salinity goes up and down, so this explanation does not hold for allyears during this period.

**Response: we understand the concern of the reviewer. The mention of the WSDW will be added, as for the response of the comment L275-278.**

L424 (here around 937): "explains most, but not all, of the observed. . ."

**399 Response: the sentence will be corrected.**

L463 (here around 977): GLODAPv2 version are written as "GLODAPv2.2021 (.2020 is soon to be 401 released). You do mean 2021 and not 2020?

**402 Response: the data will not be included in GLODAP in the 2020 version, but in the following one.**

L851-853 (here around 1353-1354): Table 2 (and in general): You may want to consider if you want to keep AOU as parameter, when you mostly refer to oxygen. The trends are almost exactly thesame (but opposite of course), and gives the same message.

**Response: we agree with the reviewer. AOU will be removed from Table 2 and from Figure 2 (and from the corresponding parts in the text).**

Technical comments

L22: This is, however, modulated. . .

**412 Response: the comas will be added.**

L35: The references should, typically, be chronologically ordered. Please check throughout the 414 manuscript. (There are more examples of this, but I won't comment on this more.)

**415 Response: we agree with the referee. We will check for other occurrences.**

L71: This is, however, not the...

- 417 Response: the comas will be added.
- 418 L91: "... (405 and 465 km, respectively)."
- 419 **Response: the coma will be added.**

- 420 L107-113: Examplified with ". . .East of the Kerguelen. . .", this section has many of these 421 "directions/locations" (east/west/. . .) spelled with a large letter, even not part of a name. I think this
- 422 is not correct, and if so, please change.
- 423 **Response: this will be corrected.**
- 424 L118-119: . . . 28.27-bottom, respectively. . .
- 425 **Response: the coma will be added.**
- 426 L172-173: Change font; the part of the sentence from "for deep samples. . ." are in a different font 427 (maybe "Cambria").
- 428 **Response: the font will be changed.**
- 429 L220: Change font for "value for".
- 430 Response: the font will be changed.
- 431 L306: Add a comma: "2018 (Fig 3a), probably . . ."
- 432 **Response: the coma will be added.**
- 433 L340: Add a ".": Pardo et al. (2017)
- 434 **Response: the dot will be added.**
- 435 L347: For consistency, change "South-Western" to "South-western" (similar as on L325).
- 436 **Response: this will be corrected.**
- 437 L449: Remove "." for consistency: (e.g. Frölicher et al., 2014).
- 438 Response: the coma will be deleted.
- 439 L451: References in chronological order.
- 440 Response: we agree with the referee. This will be corrected.

- - 11

**Variability and stability of anthropogenic CO2 in Antarctic Bottom Waters observed in the Indian sector of the Southern Ocean, 1978-2018**

Léo Mahieu1, Claire Lo Monaco2, Nicolas Metzl2, Jonathan Fin2, Claude Mignon2

1Ocean Sciences, School of Environmental Sciences, University of 4 Brownlow Street, Liverpool L69 3GP, UK

2LOCEAN-IPSL, CNRS, Sorbonne Université, CNRS/IRD/MNHN Paris, France

Correspondence to: Léo Mahieu (Leo.Mahieu@live.fr); Claire Lo Monaco (claire.lomonaco@locean.upmc.fr)

**454 Abstract**

Antarctic bottom waters (AABWs) are known as a long term sink for anthropogenic CO2 (Cant) but is hardly 456 quantified because of the scarcity of the observations, specifically at an interannual scale. We present in this 457 manuscript an original dataset combining 40 years of carbonate system observations in the Indian sector of the Southern Ocean (Enderby Basin) to evaluate and interpret the interannual variability of Cant in the AABW. This 458 459 investigation is based on regular observations collected at the same location (63° E/56.5° S) in the frame of the French observatory OISO from 1998 to 2018 extended by GEOSECS and INDIGO observations (1978, 1985 and 460 461 1987). At this location the main sources of AABW sampled is the fresh and younger Cape Darnley bottom waterBottom 462 463 Water (CDBW) and the Weddell Sea deep waterDeep Water (WSDW). Our calculations reveal that Cant 464 concentrations increased significantly in the AABW, from about +the average concentration of 7 µmol.kg-1 465 incalculated for the period 1978-1987 to +the average concentration of 13 µmol.kg-1 infor the period 2010-2018. This is comparable to previous estimates in other SO basins, with the exception of bottom waters close to their 466 467 formation sites where  $C_{ant}$  concentrations are about twice as large. Our analysis shows that the total carbon ( $C_T$ ) 468 and Cant increasing rates in the AABW are about the same over the period 1978-2018, and we conclude that the 469 long-term change in  $C_T$  is mainly due to the uptake of anthropogenic  $CO_2C_{ant}$  in the different formation regions.

variations in hydrological ( $\Theta$ , potential temperature ( $\Theta$ ), salinity (S))) and biogeochemical ( $C_T$ , total alkalinity ( $A_{T,T}$ )

This is, however, modulated by significant interannual to pluriannualmulti-annual variability associated with

), dissolved oxygen (O2)) properties. A surprising result is the apparent stability of  $C_{ant}$  concentrations in recent 473 years despite the increase in  $C_T$  and the gradual acceleration of atmospheric CO2.

The Cant sequestration by AABWs is more variable than expected and depends on a complex combination of
physical, chemical and biological processes at the formation sites and during the transit of the different AABWs.
The interannual variability at play in AABWAABWs needs to be carefully considered on the extrapolated
estimation of Cant sequestration based on sparse observations over several years.

**479 1 Introduction**

 $\frac{\text{Carbon dioxide (CO2)}{\text{Corbon dioxide (CO2)}} atmospheric concentration has been increasing since the start of the industrialization$ (Keeling and Whorf, 2000). This increase leads to an ocean uptake of about a quarter of Cant emissions (Le Quéréet al., 2018; Gruber et al., 2019a). It is widely acknowledged that the Southern Ocean (SO) is responsible for 40

[revised manuscript text omitted]

|--------------------------------------|

Circumpolar Deep Water (LCDW) by mixing with High Salinity Surface Water (HSSW) when the WSBW-LCDW 561 enters the Weddell basin (see Fig. 2 in van Heuven et al., 2011). The WSDW in the ACC and Weddell Gyre mixes 562 with the Circumpolar Deep Water (CDW).LCDW during its transit from the Weddell basin. A part of the WSDW 563 deflecting southward with the ACC in the Enderby Basin reaches the north-western part of the Princess Elizabeth 564 Trough (PET) region, East of (area separating the Kerguelen Plateau, from the Antarctic continent), where it mixes 565 with other types of AABWs (Heywood et al., 1999; Orsi et al., 1999). The PET deepest point is 3750 m, 566 deep enough to allow AABWs to flow between the Australian Antarctic Basin and the Enderby Basin (Heywood 567 et al., 1999). 568 InAt the east of the PET-sector, the CAC transports a mixture of RSBW and ALBW and accelerates

Northwardnorthward along the Easterneastern side of the Kerguelen Plateau (Mantyla and Reid, 1995; Fukamachi et al., 2010). Part of the ALBW-RSBW mixture also reaches the Westernwestern side of the Kerguelen Plateau (by the southern part of the PET (Heywood et al., 1999; Orsi et al., 1999; Van Wijk and Rintoul, 2014) and mixmixes with the CDBW. The mixture of CDBW and ALBW-RSBW either flows Westwardwestward with the CAC and dilutes with the CDWLCDW (Meijers et al., 2010) or flow Northwardflows northward (Ohshima et al., 2013) and mixmixes with the-older WSDW before reaching the location of our time-series station in the eastern Enderby Basin.

Figure 1.

**577 2.3 AABW definition**

Nowadays, the The distinction of water masses is usually performed according to neutral density ( $\gamma^n$ ) layers. In the 579 SO, CDWLCDW and AABW properties are generally well defined in the range 28.15-28.27 kg.m-3 and 28.27-580 bottom, respectively (Orsi et al., 1999; Murata et al 2019). However, to interpret the long-term variability of the 581 properties in the AABWsAABW core at our location, we prefer to adjust the AABW layer in definition to a narrow 582 band, and select the samples for(more homogeneous) layer that we call Lower Antarctic Bottom Water (LAABW), 583 characterised by  $\gamma^n > 28.35$  kg.m-3 (range starting atroughly ranging from 4200m to 4600m depending on the 584 year4800m, see Fig. 3).  $\gamma^{\text{B}} > 28.35 \text{ kg.m}^3 \text{This definition}$  corresponds to the AABW characteristics observed at 585 higher latitudes in the Indian SO sector (Roden et al., 2016). The layer above the LAABW is hereafter called 586 Upper Antarctic Bottom Water (UAABW).

**587 2.4 AABW composition at OISO-ST11**

At each formation site, AABWs experienced significant temporal property changes, mostly recognized at decadal 589 seale (e.g. freshening in the South Indian Ocean, Menezes et al., 2017) with potential impact on carbon uptake and 590 Cant concentrations during AABW formation (Shadwick et al., 2013). The O-S diagram constructed from yearly 591 averaged data in bottom waters (Fig. 2) shows that the AABW at OISO-ST11 is a complex mixture of WSDW, 592 CDBW, RSBW and ALBW. The coldest type of AABW was observed at the GEOSECS station at 60° S (-0.56 593 °C), probably because it experienced less mixing with CDW compared to the warmer type of AABW observed at 594 the INDIGO-1 station at 53° S (-0.44 °C). For the other eruises and years, O in AABW ranges from -0.51 to -0.45 595 °C with no clear indication on the specific AABW origin. The S range observed in the bottom waters at OISO-596 ST11 (34.65 34.67), illustrates either changes in mixing with various AABW sources or temporal variations at the 597 formation site. Given the knowledge of deep and bottom waters circulation and characteristics (Fig.s 1 and 2) and

| 598 | the significant C ant concentrations that we estimated at depth (Fig. 3), the main contribution at our location is likely |  |
|-----|--------------------------------------------------------------------------------------------------------------------------------------|--|
| 599 | the younger and colder CDBW for which relatively high Cant concentrations have been recently documented                              |  |
| 600 | (Roden et al., 2016). From its formation region, the CDBW can either flow westward with the CAC or flow                              |  |
| 601 | northward in the Enderby Basin (Ohshima et al., 2013, Fig. 1)In the CAC branch, the CDBW mixes with the                              |  |
| 602 | CDW along the Antarctic shelf and the continental slope between 80°E and 30°E (Meijers et al., 2010; Roden et                        |  |
| 603 | al., 2016). On the western side of the Kerguelen Plateau, CDBW also mixes with RSBW and ALBW (Orsi et al.,                           |  |
| 604 | 1999; Van Wijk and Rintoul, 2014). In this context, the Cant concentrations observed in the bottom layer at OISO-                    |  |
| 605 | ST11 are probably not linked to a single AABW source, but are likely a complex interplay of AABW from different                      |  |
| 606 | sources with different biogeochemical properties.                                                                                    |  |
| 607 | Einer 2                                                                                                                              |  |

Figure 2.

**608 3 Material and methods**

**609 3.1 Validation of the data**

For 1998-2004, the OISO data were quality controlled in CARINA (Lo Monaco et al., 2010) and for 2005 and 2009-2011 in GLODAPv2 (Key et al., 2015; Olsen et al., 2016, 2019). The 3 additional stationsdatasets from 611 612 GEOSECS. INDIGO-1 and INDIGO-3 were first qualified in GLODAP v1GLODAPv1 (Key et al., 2004) and 613 previously used for the first Cant estimates in the Indian Ocean (Sabine et al., 1999). The dataadjustments 614 recommended for INDIGO III (1987)these historical datasets have been revisited in GLODAP-v2 but the 615 correctionCARINA and GLODAPv2. In this paper we used the revised adjustments applied on  $A_{\rm T}$  values leads to 616 a suspicious offset and we decided to use the GLODAPv2 data product, with one exception for the total alkalinity 617 (AT) data from INDIGO-3 for which we applied an intermediate adjustment proposed in GLODAP-v1 and between 618 the recommendation from GLODAPv1 (confirmed in CARINA-) for no adjustment and the adjustment by -8 619 µmol.kg-1 applied to the GLODAPv2 data product (justification in Supp. Mat.).

For the recent OISO cruises conducted in 2012-2018 not yet qualified included in the GLODAP project most recent

GLODAPv2 product, we have proceeded to a data quality control mainly based on repeated observations in deep waters (CDW)-where Cant concentrations are low and subject to very small changes from year to year. At this location, the seasonal variations of all properties are only observed in the mixed layer, about 50 m in austral summer and 150 m in winter (Metzl et al., 2006). Therefore, for deep water analysis, we used the observations available for all seasons (Table 1). (see Supp. Mat.).

**626 3.2 Biogeochemical measurements**

Measurement methods during OISO cruises were previously described (Jabaud-Jan et al., 2004; Metzl et al., 2006). 628 In short, measurements were obtained using Conductivity-Temperature-Depth (CTD) casts fixed on a 24 bottles 629 rosette equipped with 12 L General Oceanics Niskin bottles. Potential temperature (O-(in °C) and salinity (S (no 630 unit) measurements have an accuracy of 0.01 of their respective units. Cr002 °C and Ar0.005 respectively. Ar and 631 CT were sampled in 500 mL glass bottles and poisoned with 100 µL of HgCl2mercuric chloride saturated solution 632 to halt biological activity. Discrete CT and AT samples were analyzed onboard by potentiometric titration derived 633 from the method developed by Edmond (1970) using a closed cell. The accuracyrepeatability for CT and AT varies 634 from 1 to 3.5 µmol.kg-1 (depending on the cruise) and is determined by sample duplicates (in surface, at 1000 m

Mis en forme : Emphase pâle and in bottom waters. All). The accuracy of CT and AT measurements were calibrated with was ensured by daily 636 analyses of Certified Reference Materials (CRMs) provided by A.G. Dickson laboratory (Scripps Institute of 637 Oceanography). O2Dissolved oxygen (O2) concentration was determined by an oxygen sensor fixed on the rosette. 638 These values were adjusted using measurements obtained by Winkler titrations using a potentiometric titration 639 system (at least 12 measurements for each profile). Thiosulphate The thiosulphate solution used infor the Winkler 640 titration was calibrated using iodate standard solution to provide(provided by Ocean Scientific International 641 Limited) to ensure the standard O2 accuracy of 2 µmol.kg-1. Nitrate (NO3) and Silicate (Si) concentrations were 642 measured onboard or offshoreonshore with an automatic colorimetric Technicon analyser following the methods 643 described by Tréguer and Le Corre (1975) until 2008, and the revised protocol described by Aminot and Kérouel 644 (2007) since 2009. Based on replicate measurements for deep samples we estimate an error of about 0.3 % for 645 both nutrients. NO3 concentrationsdata are not available for all the cruises used in this analysis. The mean NO3 concentrations in the AABWLAABW at OISO-ST11 is  $32.8 \pm 1.2 \mu mol.kg^{-1}$  while the average value derived from 646 647 the GLODAP-v2 database in bottom waters south of 50°S in the South Indian Ocean is  $32.4 \pm 0.6 \,\mu$ mol.kg-1. The 648 lack of NO3 data for few cruises has been palliated by considering a standardclimatological value of 33 µmol.kg-1 649 with a limited impact on Cant determined by the C° method (from 0.1 µmol.kg-1 to 1.7(<2 µmol.kg-1 on the mean 650 annual valuesestimates based on the differences observed between NO3 measurements and the climatological 651 value).

3.3 Cant calculation using the TrOCA method

The TrOCA method was first presented by Touratier and Goyet (2004a, 2004bb) and revised by Touratier et al.

(2007). Following the concept of the quasi-conservative tracer NO (Broecker, 1974), TrOCA is a tracer defined as a combination of  $O_2$ ,  $C_T$  and  $A_T$ , following:

$TrOCA = O_2 + a\left(C_T - \frac{1}{2}A_T\right),$

where a is the Redfield ratio.

(1)

(2)

defined in Touratier et al. (2007) as combination of the Redfield equation coefficients for  $CO_2$ ,  $O_2$ ,  $HPO_4^{2-}$  and  $H^+$ . For more details about the definition and the calibration of this parameter, please refer to Touratier et al. (2007). The temporal change in TrOCA is independent of biological processes and can be attributed to anthropogenic carbon (Touratier and Goyet, 2004a). Therefore,  $C_{ant}$  can be directly calculated from the difference between TrOCA and its pre-industrial value TrOCA°:

 $663 \qquad C_{ant} = \frac{TrOCA - TrOCA^0}{a},$

where TrOCA $\circ$  is evaluated as a function of  $\theta$  and AT (Eq. 3):

$$\quad TrOCA^0 = e^{\left[b - (c) \cdot \theta - \frac{d}{A_T^2}\right]},\tag{3}$$

In these expressions, coefficients a, b, c and d were adjusted by Touratier et al. (2007) from free anthropogenic 667  $CO_2$  deep waters using the tracers  $\Delta^{14}C$  and CFC-11 from the GLODAP V1GLODAPv1 database (Key et al.,

2004). The final expression used to calculate Cant is:

$$C_{ant} = \frac{O_2 + 1,279 \left( C_T - \frac{1}{2} A_T \right) - e^{\left[ 7,511 - \left( 1,087,10^{-2} \right) \cdot \theta - \frac{7,8110^5}{A_T 2} \right]}}{1,279},$$
(4)

| 671 | The consideration of the errors on the different parameters involved in the TrOCA method results in an uncertainty                                                                                                                                                                                                                                                                                                                                                                                                                                                                                                                                                                                                                                                                                                                                                                                                                                                                                                                                                                                                                                                                                                                                                                                                                                                                                                                                                                                                                                                                                                                                                                                                                                                                                                                                                                                                                                                                                                                                                                                                                                                                                                                                                                                                                                                                                                                                                                                                                                                                                                                                                     |                                         |
|-----|------------------------------------------------------------------------------------------------------------------------------------------------------------------------------------------------------------------------------------------------------------------------------------------------------------------------------------------------------------------------------------------------------------------------------------------------------------------------------------------------------------------------------------------------------------------------------------------------------------------------------------------------------------------------------------------------------------------------------------------------------------------------------------------------------------------------------------------------------------------------------------------------------------------------------------------------------------------------------------------------------------------------------------------------------------------------------------------------------------------------------------------------------------------------------------------------------------------------------------------------------------------------------------------------------------------------------------------------------------------------------------------------------------------------------------------------------------------------------------------------------------------------------------------------------------------------------------------------------------------------------------------------------------------------------------------------------------------------------------------------------------------------------------------------------------------------------------------------------------------------------------------------------------------------------------------------------------------------------------------------------------------------------------------------------------------------------------------------------------------------------------------------------------------------------------------------------------------------------------------------------------------------------------------------------------------------------------------------------------------------------------------------------------------------------------------------------------------------------------------------------------------------------------------------------------------------------------------------------------------------------------------------------------------------|-----------------------------------------|
| 672 | of $\pm 6.25 \ \mu mol.kg^{-1}$ (mostly due to the parameter a, leading to $\pm 3.31 \ \mu mol.kg^{-1}$ ). As this error is relatively large                                                                                                                                                                                                                                                                                                                                                                                                                                                                                                                                                                                                                                                                                                                                                                                                                                                                                                                                                                                                                                                                                                                                                                                                                                                                                                                                                                                                                                                                                                                                                                                                                                                                                                                                                                                                                                                                                                                                                                                                                                                                                                                                                                                                                                                                                                                                                                                                                                                                                                                           |                                         |
| 673 | compared to the expected Cant concentrations in deep and bottom SO waters (Pardo et al., 2014) we will compare                                                                                                                                                                                                                                                                                                                                                                                                                                                                                                                                                                                                                                                                                                                                                                                                                                                                                                                                                                                                                                                                                                                                                                                                                                                                                                                                                                                                                                                                                                                                                                                                                                                                                                                                                                                                                                                                                                                                                                                                                                                                                                                                                                                                                                                                                                                                                                                                                                                                                                                                                         |                                         |
| 674 | the TrOCA results using another indirect method to interpret Cant changes over 40 years.                                                                                                                                                                                                                                                                                                                                                                                                                                                                                                                                                                                                                                                                                                                                                                                                                                                                                                                                                                                                                                                                                                                                                                                                                                                                                                                                                                                                                                                                                                                                                                                                                                                                                                                                                                                                                                                                                                                                                                                                                                                                                                                                                                                                                                                                                                                                                                                                                                                                                                                                                                               | Mis en forme : Indice                   |
|     |                                                                                                                                                                                                                                                                                                                                                                                                                                                                                                                                                                                                                                                                                                                                                                                                                                                                                                                                                                                                                                                                                                                                                                                                                                                                                                                                                                                                                                                                                                                                                                                                                                                                                                                                                                                                                                                                                                                                                                                                                                                                                                                                                                                                                                                                                                                                                                                                                                                                                                                                                                                                                                                                        |                                         |
| 675 | 3.4 $C_{ant}$ calculation using the preformed inorganic carbon ( $(C^{v})$ ) method                                                                                                                                                                                                                                                                                                                                                                                                                                                                                                                                                                                                                                                                                                                                                                                                                                                                                                                                                                                                                                                                                                                                                                                                                                                                                                                                                                                                                                                                                                                                                                                                                                                                                                                                                                                                                                                                                                                                                                                                                                                                                                                                                                                                                                                                                                                                                                                                                                                                                                                                                                                    |                                         |
| 676 | To support the $C_{ant}$ trend determined with the TrOCA method, $C_{ant}$ was also estimated using a back-calculation                                                                                                                                                                                                                                                                                                                                                                                                                                                                                                                                                                                                                                                                                                                                                                                                                                                                                                                                                                                                                                                                                                                                                                                                                                                                                                                                                                                                                                                                                                                                                                                                                                                                                                                                                                                                                                                                                                                                                                                                                                                                                                                                                                                                                                                                                                                                                                                                                                                                                                                                                 |                                         |
| 677 | approach noted $\underline{\mathbb{C}^{0}}_{c}$ (Brewer, 1978; Chen and Millero, 1979), previously adapted for $C_{ant}$ estimates along the                                                                                                                                                                                                                                                                                                                                                                                                                                                                                                                                                                                                                                                                                                                                                                                                                                                                                                                                                                                                                                                                                                                                                                                                                                                                                                                                                                                                                                                                                                                                                                                                                                                                                                                                                                                                                                                                                                                                                                                                                                                                                                                                                                                                                                                                                                                                                                                                                                                                                                                           |                                         |
| 678 | WOCE-I6 section between South Africa and Antarctica (Lo Monaco et al., 2005a). This method consists in the                                                                                                                                                                                                                                                                                                                                                                                                                                                                                                                                                                                                                                                                                                                                                                                                                                                                                                                                                                                                                                                                                                                                                                                                                                                                                                                                                                                                                                                                                                                                                                                                                                                                                                                                                                                                                                                                                                                                                                                                                                                                                                                                                                                                                                                                                                                                                                                                                                                                                                                                                             |                                         |
| 679 | correction of the measured $C_T$ for the biological contribution ( $C_{bio}$ ) and the preindustrial preformed $C_T$ ( $C_{0,PT}C_{PI}^0$ ):                                                                                                                                                                                                                                                                                                                                                                                                                                                                                                                                                                                                                                                                                                                                                                                                                                                                                                                                                                                                                                                                                                                                                                                                                                                                                                                                                                                                                                                                                                                                                                                                                                                                                                                                                                                                                                                                                                                                                                                                                                                                                                                                                                                                                                                                                                                                                                                                                                                                                                                           |                                         |
| 680 | $C_{ant} = C_T - C_{bio} - \frac{C_{0,\mu T}}{C_{0,\mu T}}, C_{PI}^0,$                                                                                                                                                                                                                                                                                                                                                                                                                                                                                                                                                                                                                                                                                                                                                                                                                                                                                                                                                                                                                                                                                                                                                                                                                                                                                                                                                                                                                                                                                                                                                                                                                                                                                                                                                                                                                                                                                                                                                                                                                                                                                                                                                                                                                                                                                                                                                                                                                                                                                                                                                                                                 |                                         |
| 681 | (5)                                                                                                                                                                                                                                                                                                                                                                                                                                                                                                                                                                                                                                                                                                                                                                                                                                                                                                                                                                                                                                                                                                                                                                                                                                                                                                                                                                                                                                                                                                                                                                                                                                                                                                                                                                                                                                                                                                                                                                                                                                                                                                                                                                                                                                                                                                                                                                                                                                                                                                                                                                                                                                                                    |                                         |
| 682 | $C_{\text{bio}}$ (Eq. 6) depends on carbonate dissolution and organic matter remineralization, taking account of the corrected                                                                                                                                                                                                                                                                                                                                                                                                                                                                                                                                                                                                                                                                                                                                                                                                                                                                                                                                                                                                                                                                                                                                                                                                                                                                                                                                                                                                                                                                                                                                                                                                                                                                                                                                                                                                                                                                                                                                                                                                                                                                                                                                                                                                                                                                                                                                                                                                                                                                                                                                         |                                         |
| 683 | RedfieldC/O 2 ratio from Kortzinger et al. (2001):                                                                                                                                                                                                                                                                                                                                                                                                                                                                                                                                                                                                                                                                                                                                                                                                                                                                                                                                                                                                                                                                                                                                                                                                                                                                                                                                                                                                                                                                                                                                                                                                                                                                                                                                                                                                                                                                                                                                                                                                                                                                                                                                                                                                                                                                                                                                                                                                                                                                                                                                                                                                          |                                         |
| 684 | $C_{bio} = 0.5\Delta A_T - (C/O_2 + 0.5N/O_2)\Delta O_2 , \qquad (6)$                                                                                                                                                                                                                                                                                                                                                                                                                                                                                                                                                                                                                                                                                                                                                                                                                                                                                                                                                                                                                                                                                                                                                                                                                                                                                                                                                                                                                                                                                                                                                                                                                                                                                                                                                                                                                                                                                                                                                                                                                                                                                                                                                                                                                                                                                                                                                                                                                                                                                                                                                                                                  |                                         |
| 685 | Where $C/O_2 = 106/138$ and $N/O_2 = 16/138$ . $\Delta A_T$ and $\Delta O_2$ are the difference between the measured values (A T and A T and A O are the difference between the measured values (A T and A T and A O are the difference between the measured values (A T and A O are the difference between the measured values (A T and A O are the difference between the measured values (A T and A O are the difference between the measured values (A T and A O are the difference between the measured values (A T and A O are the difference between the measured values (A T and A O are the difference between the measured values (A T and A O are the difference between the measured values (A T and A O are the difference between the measured values (A T and A O are the difference between the measured values (A T and A O are the difference between the measured values (A T and A O are the difference between the measured values (A T and A O are the difference between the measured values (A T and A O are the difference between the measured values (A T and A O are the difference between the measured values (A T and A O are the difference between the measured values (A T and A O are the difference between the measured values (A T and A O are the difference between the measured values (A T and A O are the difference between the measured values (A T and A O are the difference between the measured values (A T and A O are the difference between the measured values (A T and A O are the difference between the measured values (A T are the difference between the difference between the difference between the measured values (A T and A O are the difference between th |                                         |
| 686 | $O_2$ ) and the preindustrial preformed values ( $A_T^0$ and $O_2^0$ ). $A_T^0$ (Eq. 7) has been computed by Lo Monaco et al.                                                                                                                                                                                                                                                                                                                                                                                                                                                                                                                                                                                                                                                                                                                                                                                                                                                                                                                                                                                                                                                                                                                                                                                                                                                                                                                                                                                                                                                                                                                                                                                                                                                                                                                                                                                                                                                                                                                                                                                                                                                                                                                                                                                                                                                                                                                                                                                                                                                                                                                                          |                                         |
| 687 | (2005a) as a function of $\Theta$ , S and the conservative tracer PO:                                                                                                                                                                                                                                                                                                                                                                                                                                                                                                                                                                                                                                                                                                                                                                                                                                                                                                                                                                                                                                                                                                                                                                                                                                                                                                                                                                                                                                                                                                                                                                                                                                                                                                                                                                                                                                                                                                                                                                                                                                                                                                                                                                                                                                                                                                                                                                                                                                                                                                                                                                                                  |                                         |
| 688 | $A_T^0 = 0.0685PO + 59.79S - 1.45\Theta + 217.1, $ (7)                                                                                                                                                                                                                                                                                                                                                                                                                                                                                                                                                                                                                                                                                                                                                                                                                                                                                                                                                                                                                                                                                                                                                                                                                                                                                                                                                                                                                                                                                                                                                                                                                                                                                                                                                                                                                                                                                                                                                                                                                                                                                                                                                                                                                                                                                                                                                                                                                                                                                                                                                                                                      |                                         |
| 689 | PO (Eq. 8) has been defined by Broecker (1974) and depends on the equilibrium of $O_2$ with phosphate (PO 4 ). When                                                                                                                                                                                                                                                                                                                                                                                                                                                                                                                                                                                                                                                                                                                                                                                                                                                                                                                                                                                                                                                                                                                                                                                                                                                                                                                                                                                                                                                                                                                                                                                                                                                                                                                                                                                                                                                                                                                                                                                                                                                                                                                                                                                                                                                                                                                                                                                                                                                                                                                                         |                                         |
| 690 | $PO_4$ data are not available, nitrate (NO 3 ) can be used instead (as follows (the N/P ratio of 16 is from Anderson and                                                                                                                                                                                                                                                                                                                                                                                                                                                                                                                                                                                                                                                                                                                                                                                                                                                                                                                                                                                                                                                                                                                                                                                                                                                                                                                                                                                                                                                                                                                                                                                                                                                                                                                                                                                                                                                                                                                                                                                                                                                                                                                                                                                                                                                                                                                                                                                                                                                                                                                                    |                                         |
| 691 | Sarmiento, 1994):                                                                                                                                                                                                                                                                                                                                                                                                                                                                                                                                                                                                                                                                                                                                                                                                                                                                                                                                                                                                                                                                                                                                                                                                                                                                                                                                                                                                                                                                                                                                                                                                                                                                                                                                                                                                                                                                                                                                                                                                                                                                                                                                                                                                                                                                                                                                                                                                                                                                                                                                                                                                                                                      |                                         |
| 692 | $PO = O_2 + 170PO_4 = O_2 + (170/16N16)NO_3, -$                                                                                                                                                                                                                                                                                                                                                                                                                                                                                                                                                                                                                                                                                                                                                                                                                                                                                                                                                                                                                                                                                                                                                                                                                                                                                                                                                                                                                                                                                                                                                                                                                                                                                                                                                                                                                                                                                                                                                                                                                                                                                                                                                                                                                                                                                                                                                                                                                                                                                                                                                                                                                        |                                         |
| 693 | (8)                                                                                                                                                                                                                                                                                                                                                                                                                                                                                                                                                                                                                                                                                                                                                                                                                                                                                                                                                                                                                                                                                                                                                                                                                                                                                                                                                                                                                                                                                                                                                                                                                                                                                                                                                                                                                                                                                                                                                                                                                                                                                                                                                                                                                                                                                                                                                                                                                                                                                                                                                                                                                                                                    |                                         |
| 694 | To determine $O_2^0$ , it is assumed that the surface water is in full equilibrium with the atmosphere ( $O_2^0=O_{2,sat}$ ; Benson                                                                                                                                                                                                                                                                                                                                                                                                                                                                                                                                                                                                                                                                                                                                                                                                                                                                                                                                                                                                                                                                                                                                                                                                                                                                                                                                                                                                                                                                                                                                                                                                                                                                                                                                                                                                                                                                                                                                                                                                                                                                                                                                                                                                                                                                                                                                                                                                                                                                                                                                    |                                         |
| 695 | and Krause, 1980) and that after subduction <math>O_2</math> in a given water mass is only impacted by the biological activity                                                                                                                                                                                                                                                                                                                                                                                                                                                                                                                                                                                                                                                                                                                                                                                                                                                                                                                                                                                                                                                                                                                                                                                                                                                                                                                                                                                                                                                                                                                                                                                                                                                                                                                                                                                                                                                                                                                                                                                                                                                                                                                                                                                                                                                                                                                                                                                                                                                                                                                                  |                                         |
| 696 | (Weiss, 1970). The A correction of $\Theta_{2,aat} O_2^0$ has been proposed by Lo Monaco et al. (2005a) to take account of the                                                                                                                                                                                                                                                                                                                                                                                                                                                                                                                                                                                                                                                                                                                                                                                                                                                                                                                                                                                                                                                                                                                                                                                                                                                                                                                                                                                                                                                                                                                                                                                                                                                                                                                                                                                                                                                                                                                                                                                                                                                                                                                                                                                                                                                                                                                                                                                                                                                                                                                                  |                                         |
| 697 | undersaturation of $O_2$ due to sea-ice cover. $\Delta O_2 at high latitudes. O_2^0$ is, therefore, corrected by assuming a mean                                                                                                                                                                                                                                                                                                                                                                                                                                                                                                                                                                                                                                                                                                                                                                                                                                                                                                                                                                                                                                                                                                                                                                                                                                                                                                                                                                                                                                                                                                                                                                                                                                                                                                                                                                                                                                                                                                                                                                                                                                                                                                                                                                                                                                                                                                                                                                                                                                                                                                                                       |                                         |
| 698 | mixing ratio of the ice-covered surface waters k=50 % (Lo Monaco et al., 2005a), and a mean value for $O_{\rm 2}$                                                                                                                                                                                                                                                                                                                                                                                                                                                                                                                                                                                                                                                                                                                                                                                                                                                                                                                                                                                                                                                                                                                                                                                                                                                                                                                                                                                                                                                                                                                                                                                                                                                                                                                                                                                                                                                                                                                                                                                                                                                                                                                                                                                                                                                                                                                                                                                                                                                                                                                                                      | Mis en forme : Police : Times New Roman |
| 699 | undersaturation in ice-covered surface waters $\alpha = \pm 12 \%$ (Anderson et al., 1991) according to Eq. 9:                                                                                                                                                                                                                                                                                                                                                                                                                                                                                                                                                                                                                                                                                                                                                                                                                                                                                                                                                                                                                                                                                                                                                                                                                                                                                                                                                                                                                                                                                                                                                                                                                                                                                                                                                                                                                                                                                                                                                                                                                                                                                                                                                                                                                                                                                                                                                                                                                                                                                                                                                         |                                         |
| 700 | $\Delta O_2 = (1 - \alpha k) O_{2,sat} - O_2 = A O U , $ (9)                                                                                                                                                                                                                                                                                                                                                                                                                                                                                                                                                                                                                                                                                                                                                                                                                                                                                                                                                                                                                                                                                                                                                                                                                                                                                                                                                                                                                                                                                                                                                                                                                                                                                                                                                                                                                                                                                                                                                                                                                                                                                                                                                                                                                                                                                                                                                                                                                                                                                                                                                                                                           |                                         |
| 701 | $C_{0,pq}C_{p1}^{0}$ in equation 5 is a function of the current preformed $C_T$ ( $C_{0,obs}C_{obs}^{0}$ ) and a reference water term (Eq. 10):                                                                                                                                                                                                                                                                                                                                                                                                                                                                                                                                                                                                                                                                                                                                                                                                                                                                                                                                                                                                                                                                                                                                                                                                                                                                                                                                                                                                                                                                                                                                                                                                                                                                                                                                                                                                                                                                                                                                                                                                                                                                                                                                                                                                                                                                                                                                                                                                                                                                                                                        |                                         |
| 702 | $C_{0,PT} = C_{0,obs} C_{PI}^0 = C_{obs}^0 + \left[ C_T - C_{bio} - C_{0,obs} \right]_{PET} [C_T - C_{bio} - C_{obs}^0]_{REF} ,$                                                                                                                                                                                                                                                                                                                                                                                                                                                                                                                                                                                                                                                                                                                                                                                                                                                                                                                                                                                                                                                                                                                                                                                                                                                                                                                                                                                                                                                                                                                                                                                                                                                                                                                                                                                                                                                                                                                                                                                                                                                                                                                                                                                                                                                                                                                                                                                                                                                                                                                                       |                                         |
| 703 | (10)                                                                                                                                                                                                                                                                                                                                                                                                                                                                                                                                                                                                                                                                                                                                                                                                                                                                                                                                                                                                                                                                                                                                                                                                                                                                                                                                                                                                                                                                                                                                                                                                                                                                                                                                                                                                                                                                                                                                                                                                                                                                                                                                                                                                                                                                                                                                                                                                                                                                                                                                                                                                                                                                   |                                         |
| 704 | $\underline{C_{0.obs}}$ has been computed similarly as $A_T^0$ (Eq. 11):                                                                                                                                                                                                                                                                                                                                                                                                                                                                                                                                                                                                                                                                                                                                                                                                                                                                                                                                                                                                                                                                                                                                                                                                                                                                                                                                                                                                                                                                                                                                                                                                                                                                                                                                                                                                                                                                                                                                                                                                                                                                                                                                                                                                                                                                                                                                                                                                                                                                                                                                                                                               |                                         |
| 705 | $C_{obs}^{0} = -0.0439PO + 42.79S - 12.02O + 739.8, $ (11)                                                                                                                                                                                                                                                                                                                                                                                                                                                                                                                                                                                                                                                                                                                                                                                                                                                                                                                                                                                                                                                                                                                                                                                                                                                                                                                                                                                                                                                                                                                                                                                                                                                                                                                                                                                                                                                                                                                                                                                                                                                                                                                                                                                                                                                                                                                                                                                                                                                                                                                                                                                                             |                                         |
| 706 | Where the reference water term is a constant for a given time of observation, corresponding to the time when $C_{obs}^0$                                                                                                                                                                                                                                                                                                                                                                                                                                                                                                                                                                                                                                                                                                                                                                                                                                                                                                                                                                                                                                                                                                                                                                                                                                                                                                                                                                                                                                                                                                                                                                                                                                                                                                                                                                                                                                                                                                                                                                                                                                                                                                                                                                                                                                                                                                                                                                                                                                                                                                                                               |                                         |
| 707 | is parameterized. In this paper, we used the parameterization given by Lo Monaco et al., (2005a) and their                                                                                                                                                                                                                                                                                                                                                                                                                                                                                                                                                                                                                                                                                                                                                                                                                                                                                                                                                                                                                                                                                                                                                                                                                                                                                                                                                                                                                                                                                                                                                                                                                                                                                                                                                                                                                                                                                                                                                                                                                                                                                                                                                                                                                                                                                                                                                                                                                                                                                                                                                             |                                         |
| 708 | estimated value for the reference term of 51 µmol.kg -1 . This number has been computed using an optimum                                                                                                                                                                                                                                                                                                                                                                                                                                                                                                                                                                                                                                                                                                                                                                                                                                                                                                                                                                                                                                                                                                                                                                                                                                                                                                                                                                                                                                                                                                                                                                                                                                                                                                                                                                                                                                                                                                                                                                                                                                                                                                                                                                                                                                                                                                                                                                                                                                                                                                                                                    |                                         |
| 709 | multiparametric (OMP) model and defined as 51 µmol.kg -1 fromto estimate the mixing ratio of the North Atlantic                                                                                                                                                                                                                                                                                                                                                                                                                                                                                                                                                                                                                                                                                                                                                                                                                                                                                                                                                                                                                                                                                                                                                                                                                                                                                                                                                                                                                                                                                                                                                                                                                                                                                                                                                                                                                                                                                                                                                                                                                                                                                                                                                                                                                                                                                                                                                                                                                                                                                                                                             |                                         |
| I   |                                                                                                                                                                                                                                                                                                                                                                                                                                                                                                                                                                                                                                                                                                                                                                                                                                                                                                                                                                                                                                                                                                                                                                                                                                                                                                                                                                                                                                                                                                                                                                                                                                                                                                                                                                                                                                                                                                                                                                                                                                                                                                                                                                                                                                                                                                                                                                                                                                                                                                                                                                                                                                                                        |                                         |

| 710 | deep water (Lo Monaco et al., 2005a) and in the SO (used as reference water, i.e. old water mass where $C_{ant} = 0$ ).                                           |
|-----|-------------------------------------------------------------------------------------------------------------------------------------------------------------------|
| 711 | $C_{0,obs}$ has been computed similarly as $A_{\pm}^{\theta}$ (Eq. 11):                                                                                           |
| 712 | $\mathcal{C}_{v,obs} = -0.0439P0 + 42.79S - 12.020 + 739.8, \tag{11}$                                                                                             |
| 713 | For more details inabout the $\underline{C^0}\underline{C^0}$ method, which has a final error of $\pm 6 \mu \text{mol.kg}^{-1}$ , especially on the determination |
|     |                                                                                                                                                                   |

of reference water terms and on the errors of this method, please see Lo Monaco et al. (2005a).

**715 4. Results**

The vertical distribution of hydrological and biogeochemical properties observed in deep and bottom waters and their evolution over the last 40 years are displayed in Fig. 3.2. The LCDW layer ( $\gamma^n = 28.15-28.27 \text{ kg.m}^{-3}$ ) is 717 718 characterized by maximum AOU values (Fig. 3c), maximum C4-minimum O2 concentrations (Fig. 3d2c), higher 719 CT (Fig. 2b) and minimumlower Cant concentrations than in the AABW (Fig. 3a2a). Cant concentrations were not 720 significant in the LCDW until the end of the 1990s (<6 µmol.kg-1), then our data show a suddenan increase in Cant 721 between January and Decemberthe two 1998 reoccupations, followed by relatively constant Cant concentrations 722  $(10\pm3 \ \mu\text{mol.kg}^{-1})$ . In the core of AABWLAABW ( $\gamma^n > 28.35 \ \text{kg.m}^{-3}$ ), well identified by low  $\Theta$ , low  $S_{\overline{3}}$  and high 723 O2-and low AOU, Cart concentrations are higher than in the overlying deep watersUAABW and LCDW (Fig. 3a) 724 and2a). The evolutions of the mean properties in the LAABW over 40 years are shown in Fig. 3. In this layer, Cant 725 concentrations increased from  $5\pm4 \ \mu\text{mol.kg}^{-1}$  in  $1978_{3}$  and  $7\pm4 \ \mu\text{mol.kg}^{-1}$  in the mid-1980s to  $13\pm2 \ \mu\text{mol.kg}^{-1}$  at 726 the end of the 1990s and up to  $19\pm 2 \mu \text{mol.kg}^{-1}$  in 2004 (Fig. 4a3a). Figure 4a3a also shows a very good agreement 727 between the TrOCA method and the C0 method for both the magnitude and variability of Cant in the core of 728 AABWLAABW. Our results show a mean Cant trend in AABWthe LAABW of +1.64 µmol.kg-1.decade-1 over the 729 full period and a maximum trend of the order of +6.5.2 µmol.kg-1.decade-1 over 1987-2004 (Table 2). These trends 730 are lower than the theoretical trend expected from the increase in atmospheric CO2. Indeed, assuming that the 731 surface ocean fCO2 follows the atmospheric growth rate (+1.8  $\mu$ atm.year-1 over 1978-2018), the theoretical Cant 732 trend at the AABW formation sites would be of the order of  $+8 \,\mu mol.kg^{-1}$ . decade-1. The observed slow Cant trends can be partly explained by the transit time for AABW to reach our study site and the mixing of AABWAABWs 733 734 with older CDW watersLCDW that contain less Cant over their transit (Fig. 32a).

Figure 3.2

To investigate changes in the accumulation of  $C_{aug}$  in AABW, Fig. 4 shows the evolution of  $C_{T_1}$   $A_{T_2}$   $\Theta_{2r}$   $\Theta$  and S 737 (properties used to estimate  $C_{\text{ant}}$ ), as well as the "natural" component of  $C_{T}$  ( $C_{\text{Trant}}$  calculated as the difference 738 between  $C_{T}$ -and  $C_{mat}$ ). Over the full period,  $C_{T}$  increased by 2.0±0.5 µmol.kg-1.decade-1, mostly due to the 739 accumulation of  $C_{ant}$  (Table 2). Our data also show a significant decrease in  $O_2$  concentrations by  $0.8\pm0.4 \mu$ mol.kg- 740 1.decade-1 over the 40-years period (Fig. 4e3c, Table 2) that could be caused by reduced ventilation, as suggested 741 by Schmidtko et al. (2017) who observed significant O2 loss in the global ocean. In the deep Indian SO sector, 742 these authors found a trend approaching -1 µmol.kg-1.decade-1 over 50 years (1960-2010), which is consistent with 743 our data. We did not detect any significant trend in  $A_T$ ,  $\Theta$  and S over the full period, but on shorter periods our 744 data show a significant decrease in AT from the mid 1980s to 2004 (Fig. 4d, Table 2) that is also observed in the 745 overlying deep waters (Fig. 3f). This. The low AT values observed over 2000-2004 (Fig. 3d) could suggest reduced 746 calcification in the upper ocean leading to less sinking of calcium carbonate tests and hence a decrease in CTnat (i.e. 747  $for A_T$  in deep and bottom waters over this period (Fig. 2d). For this period the increase in CT was lower than the 748 accumulation of Cant., but such feature is disputable in view of the uncertainty on the Cant calculation. This event is followed by an increase in  $C_{\text{Tunt}}$  the 'natural' component of  $C_{\text{T}}$  ( $C_{\text{nat}}$ , calculated as the difference between  $C_{\text{T}}$  and

 $C_{ant}$  since 2004 associated to a rapid decrease in O2 (and no increase in AOU) and a decrease in Cant (Table 2).

These recent trends were not associated with a small increase in  $\theta$  (Fig. 4e, Table 2), but no significant trend in  $\underline{\theta}$

or S (Fig. 4f3e, f, Table 2). The increase in  $C_{Trant}C_{nat}$  is thus unlikely originating solely from increased mixing with

LCDW during bottom waters transport-, confirming that our LAABW definition exclude mixing with the LCDW.

Enhanced organic matter remineralization is also unlikely since nitrateNO3 did not show any significant trend (Table 2).

Table 2<del>.</del>

Figure 4.3

Importantly, our data show substantial interannual variations in AABWLAABW properties, which could-759 significantly impact the trends estimated from limited reoccupations (e.g. Williams et al., 2015; Pardo et al., 2017; 760 Murata et al., 2019). For example, we found relatively higher Cant concentrations in 1985 (10 µmol.kg-1) compared 761 to 1978 and 1987 (5 µmol.kg-1)-) and 1987 (7 µmol.kg-1). This is linked to a signal of low S in 1985 (Fig. 3f) that 762 could be due to a larger contribution of fresher AABWwaters such as the WSDW or reduced mixing with saltier 763 LCDW (Fig. 3h). CDBW. This could also be related to the different sampling locations. Over the last decade (2009-764 2018), our data show large and rapid changes in S that are partly reflected on CT and O2, and that could explain 765 the relatively low Cant concentrations observed over this period. Indeed, the S maximum observed in 2012 (correlated to higher  $\theta$ ) is associated with a marked CT minimum (surprisingly almost as low as in 1987), as well 766 767 as low AT (hence low CTnat), and low nitrateNO3 concentrations. TheseSince these anomalies point to a change in 768 AABW characteristics rather than a change in mixing with the underlying deep waters, and since they were 769 associated with a decrease in Cant concentrations, one may argue for an increased contribution of bottom waters 770 ventilated far away from our study site (possibly from the Ross Sea due to higher S, Fig. 2). A few years later our 771 data show a S minimum (correlated to lower  $\theta$ ), associated with a rapid increase in CT and a rapid decrease in O2 772 between 2013 and 2016, suggesting the contribution of a closer AABW type such as the CDBW. The freshening 773 of -0.01006 decade-1 in S between 2004 and 2018 that we observed on the Westernwestern side of the Kerguelen 774 Plateau was also observed on the Easterneastern side of the Plateau by Menezes et al. (2017)-) over a similar 775 period. In this region, Menezes et al. (2017) evaluated a change in salinitys by about -0.008-decade-1 from 2007 776 to 2016 (against -0.002-decade-1 between 1994 and 2007), suggesting an acceleration of the AABW freshening in 777 recent years. However, they also reported a warming by +0.06 °C.decade-1, while we observed cooler temperature 778 in 2016-2018. This suggests that we sampled a different mixture of AABWs.

Figure 4

**780 5- Discussion**

**781 5.1 LAABW composition at OISO-ST11**

[revised manuscript text omitted]
 CT (increase 932 in Grand Crant Oral associated with an increase in Cant, and a decrease in O2 (as observed in recent years in Fig. 933 4a3a,b,c). Finally, it is also possible that the AABWLAABW observed in recent years at our location is the result 934 of a larger contribution of older RSBW-and/or, ALBW or even WSBW that have lower Cant and O2 concentrations 935 compared to CDBW formed at Cape Darnley and Prydz Bay.

**936 6- Conclusion**

The distribution and evolution of  $C_{ant}$  in the bottom layer of the SO are related to complex interactions between climatic forcing, air-sea  $CO_2$  exchange at formation sites, as well as biological and physical processes during AABWs circulation. The dataset that we collected regularly in the Enderby basin over the last 20 years (1998-2018) in the frame of the OISO project, together with historical observations obtained in 1978, 1985 and 1987

(GEOSECS and INDIGO cruises), allows the investigation of Cant changes in AABW over 40 years in this region. 942 The focus on the AABW variability is made by defining a Low Antarctic Bottom Water (LAABW) as described 943 in the Section 2.3. Our results suggest that the accumulation of Cant explains most-of (, but not all), of the observed 944 increase in  $C_T$ . We also detected a decrease in  $O_2$  that is consistent with the large-scale signal reported by 945 Schmidtko et al. (2017), possibly due to a decrease in AABWAABWs formation (Purkey and Johnson, 2012). Our 946 data further indicate rapid anomalies in some periods suggesting that for decadal to long-term estimates care have 947 to be taken when analyzing the change in Cant from data sets collected 10 or 20 years apart (e.g. Williams et al., 2015; Murata et al., 2019). Our results also show different Cant trends on short periods, with a maximum increase 948 949 of 6.5 µmol.kg-1.decade-1 between 1987 and 2004 and an apparent stability in the last 20 years (despite an increase 950 in CT). This suggests that AABWs have stored less Cant in the last decade, but our understanding of the processes 951 that explain this signal is not clear. This might be the result of the reduced CO2 uptake in the SO in the 1990s (Le Quéré et al., 2007; Landschützer et al., 2015), but this is not yet verified from direct CT or fCO2 observations in 952 953 AABW formation regions due to the lack of winter data and very large variability during summer. This calls for 954 more data collection and investigations in these regions. The apparent stability of Cant in AABWthe LAABW since 955 1998 could also be directly linked to a decrease in AABWAABWs formation in the 1990s (Purkey and Johnson, 956 2012) or a change in the contributions of AABWs from different sources, especially in the Prydz Bay region 957 (Williams et al., 2016). In these scenarios, an increased contribution of CT-rich and O2-poor older CDWLCDW 958 along AABWs transit would also explain the decoupling between  $C_{ant}$  and  $C_T$  (increase in  $C_{Trat}C_{nat}$ ) and decrease 959 in O2 concentrations observed in recent years, even if we tried to isolate this specific feature in our data selection. 960 The decoupling between Cant and CT is not a unique feature, as it was also reported along the SR03 section between 961 Tasmania and Antarctica, most probably due to advection of CT-rich waters (Pardo et al., 2017). This highlights 962 the importance of the ocean circulation in influencing the temporal CT and Cant inventories changes (De Vries et 963 al-., 2017) and the need to better separate anthropogenic and natural variability based on tim

---

## Author Response (AR2)

**Answer to the Topic Editor comments:**

Comments to the Author:

Dear Dr. Mahieu and co-authors,

Thank you for the revised submission. I am generally satisfied with the changes you made following the comments by the referees. Going through the manuscript myself, I have listed my comments below. Please prepare the final version of your manuscript taking into account these comments.

**Response: Dear Dr. Hoppema, we are thankful for your comments. Please find hereafter our responses.**

In the title: Antarctic Bottom Water (without –s) as this study measured only one type at one location.

**Response: we wanted to insist on the mix of AABW from different sources by writing it this way. This will be corrected as suggested.**

Section 2.1 is clearly part of the methods and should thus be moved to Section 3.

**Response: this section will be moved as suggested.**

I suggest to call the water mass defined here as Lower AABW, not as Low AABW. Just like other well-known water masses like Lower CDW, etc.

**Response: this will be corrected as suggested.**

Please place all 2 in CO2 in subscript.

**Response: this will be corrected in the title and the references.**

Please check the references because many are incomplete.

**Response: this will be done. DOIs will be updated and page numbers checked. The last references without page numbers do not mention any online.**

As to the data used in this study, there are several more OISO cruises (as also in the GLODAP tables). Please provide the arguments for including the cruises that the authors did, while excluding others.

**Response: the missing OISO cruises in this study correspond to the cruises when this station was not re-occupied. To clarify this, the following sentence will be added to section AABW sampling: 'In our analysis, we included all the data available for the OISO-ST11 location (which has not been sampled during each cruise for logistical reasons).'**

As to the supplement Table S1 (and discussion in the main text) with the adjustments from the different quality control efforts, it is shown that AT at the OISO cruises did not receive any adjustments. However, this is not the complete story. The GLODAP table says that there is not sufficient data for comparison in this region, upon which the OISO data did not get an adjustment because this could not be argued safely. This is actually the same as getting no quality control. This should be made clear in the manuscript.

**Response: we agree and will clarify this point as follows:**

'… this calls for great care before applying an adjustment. This is the case for $A_T$ data that did not
get an adjustment in GLODAP because this could not be argued safely due to the limited number of
data in this region.'.

L5 Shouldn't the University of Liverpool be mentioned?

**Response: this is missing indeed and will be added.**

L9 Antarctic bottom water (AABW) is known …

**Response: this will be corrected.**

L9 … but the sink is hardly quantified …

**Response: this will be added.**

L13 in the framework of …

**Response: this will be corrected.**

L16 At this location, the main sources of AABW are the low-saline … (fresh is not the word here,
because this is a saline water mass; I suggest to skip "younger" because: younger against what?)

**Response: we understand your concern and will correct this sentence as suggested.**

L20 SO has not been defined before

**Response: this will be added.**

L24 hydrographic (not: hydrological)

**Response: this will be corrected.**

L27 AABW

**Response: this will be corrected.**

L27-28 This sentence is trivial, and if not followed by which of these processes are important or how
they function, not necessary/useful.

**Response: we agree and will remove this sentence.**

L43 3% is more like the maximum. Mostly Cant is much less. I suggest to write here: less than 3%

**Response: we agree and will correct this.**

L53-55 "Thus, there is a need to better explore the CT and Cant temporal variability in the deep
ocean, especially in the SO where observations are relatively sparse." I cannot understand the
connection of this concluding sentence with the previous text in this paragraph. Please modify.

**Response: we agree that the sentence has no clear link with the previous statements. We suggest
to remove it.**

L56 AABW (without –s) Please change this throughout the manuscript.

**Response: this will be corrected.**

L58 … by covering a major part of the world ocean floor …

**Response: this will be corrected.**

L84 Study area

**Response: this will be corrected.**

L86 framework

**Response: this will be corrected.**

L98 is dominated by (instead of: is mainly governed)

**Response: this will be corrected.**

L111 … Lower Circumpolar …

**Response: this will be corrected.**

L112 I think HSSW is generally the abbreviation for High Salinity Shelf Water

**Response: this is correct, the abbreviation will be removed.**

L116-117 The PE deepest point of the PET is 3750 m, …

**Response: this will be corrected as 'The deepest point of the PET is 3750 m…'**

L158-160 "The accuracy of CT and AT measurements was ensured by daily analyses of Certified
Reference Materials (CRMs) provided by A.G. Dickson laboratory (Scripps Institute of
Oceanography)." This is indeed important to warrant the accuracy. For the interpretation it is also
important to know the accuracy. Please give the accuracy here.

**Response: A single accuracy value for all cruises is difficult to specify. Although we used the same**
**technic (and data processing) accuracy range between around 1.5 and 3 µmol/kg for both AT and**
**CT depending on the cruise. A complete list of CRMS batch number used during OISO cruise is**
**available at NCEI/OCADS with information on duplicates for each cruise**
**(https://www.nodc.noaa.gov/ocads/oceans/VOS_Program/OISO.html). As this information is**
**available at NCEI/OCADS (and the link recall in the section "Data Availability"), we think it was not**
**appropriate to list all CRM batch values for each cruise in the manuscript. We suggest to correct as**
**follows: 'The accuracy of $C_T$ and $A_T$ measurements (always better than ±3 µmol.kg$^{-1}$ for all cruises**
**since 1998) was ensured…'**

L164 silicate (no capital)

**Response: this will be corrected.**

L171 using (instead of: considering)

**Response: this will be corrected.**

L171 I do not understand why the value of 33 umol/kg was used, as the mean value from GLODAPv2
is 32.4 umol/kg. Even if the error because of this is small, it does increase it for no good reason.

**Response: this is correct. The value has been changed to 32.4 µmol.kg$^{-1}$. The change on the $C_{ant}$**
**values calculated with C° is -0.3 µmol.kg$^{-1}$.**

L187-188 from deep waters free of anthropogenic CO2 …

**Response: this will be corrected.**

L245-246 "the theoretical Cant trend at the AABW formation sites would be of the order of +8
µmol.kg-1.decade-1." How was this calculated? Only part of the AABW, when it is formed, contains
water that has been at the surface. Only that part could follow the atmospheric increase on CO2.
What percentage of surface water was assumed as contributing to AABW?

**Response: This value was listed to give a taste of the theoretical $C_T$ increase in Antarctic surface**
**waters assuming that ocean $fCO_2$ follows the atmospheric $CO_2$ increase. In the Prydz Bay region**
**Roden et al (2016) concluded that "surface waters in the seasonal ice zone track the atmospheric**
**increase in $fCO_2$". For our calculation we used the mean properties in Antarctic surface waters**
**observed in the Prydz Bay by Roden et al. (2016): SST=-1°C, SSS= 34.2, $A_T$=2291 µmol/kg and $fCO_2$=**
**376 µatm in 2006. Assuming that oceanic $fCO_2$ increased at a rate of 1.8 µatm we calculated $C_T$ and**
**we derived a trend in $C_T$ of +8 µmol/kg/decade in the Antarctic surface water (assuming no change**
**in temperature, salinity and alkalinity). Note that this value is close to the theoretical trend in $C_T$**
**calculated by Van Heuven et al. (2014) in the Weddell Sea (about +0.8 µmol/kg/yr, the red circle in**
**Figure 4a in Van Heuven et al., 2014). We suggest to revise following: "Due to the mixing of AABW**
**with old CDW (Cant free), these trends are lower than the theoretical trend expected from the**
**increase in atmospheric $CO_2$. Indeed, assuming that the surface ocean $fCO_2$ follows the**
**atmospheric growth rate (+1.8 µatm.year$^{-1}$ over 1978-2018) in the seasonal ice zone (Roden et al.,**
**2016), the theoretical $C_{ant}$ trend at the AABW formation sites would be of the order of +8 µmol.kg$^{-}$**
**$^1$.decade$^{-1}$ in the Antarctic surface water. This is close to the theoretical $C_T$ trend estimated for**
**freezing shelf water in the Weddell Sea (Van Heuven et al 2014)."**

L288 experiences

**Response: this will be corrected.**

L310 and ends … (instead of: and lasts)

**Response: this will be corrected.**

L315 … in the 1980s in the Indian sector of the Southern Ocean …

**Response: this will be corrected.**

L316 quality control (instead of: qualification)

**Response: this will be corrected.**

L427 "recognized freshening of AABWs over the last decades (Rintoul, 2007)." With a reference from
2007, this is not about the last decades. Please change wording or give a different reference.

**Response: we agree that there is a lack of consistency between the sentence and reference. We**
**suggest to change the sentence as follow: 'recognized freshening of the AABW (Rintoul, 2007;**
**Anilkumar et al., 2015).'**

L484 change to: GLODAPv2.2021

**Response: this will be corrected.**

L504 Please add info on what kind of this reference is and possibly where it can be found online.

**Response: we suggest to replace the current reference by the following:**
**Coverly, S. C., Aminot, A., and R. Kérouel, 2009. Nutrients in Seawater Using Segmented Flow**
**Analysis, In Practical Guidelines for the Analysis of Seawater, Ed. Oliver Wurl, CRC Press, June 2009,**
**doi: 10.1201/9781420073072.ch8.**

L506 Cycles (also in other cases where this journal is concerned)

**Response: this will be corrected.**

L538 pages: 205-206

**Response: this will be corrected.**

L551 pCO2

**Response: this will be corrected.**

L602 should be cited as: 18, GB1042, doi:10.1029/2002GB002017
L606 should be cited differently, similar as above
L624 pages: 346-349
L636 pages: 1221-1224
In many cases the references are incomplete, for example missing page numbers. Please go through
the references and correct them.

**Response: all the references will be checked and updated. The DOIs will be updated, and the page**
**numbers checked. The last references without page numbers do not mention any online.**

Figure 1 Please add that these are very rough transport paths. The dashed line for the ACC gives the
position, says the caption. What position? The ACC is wide; please explain. The path of the AABW in
the Weddell Sea is not correct. Neither is the path of the AABW from Prydz Bay and Cape Darnley,
which flows along the coast to the west and enters the Weddell circulation.

**Response: the mention will be added and the figure updated.**

Figure 2 The term is Hovmöller diagram.

**Response: this will be corrected.**

Thank you and best wishes

Mario Hoppema

[revised manuscript text omitted]
 $\mu mol.kg^{-1}$ | | $NO_3$ $\mu mol.kg^{-1}$ | | $O_2$ $\mu mol.kg^{-1}$ | | $A_T$ $\mu mol.kg^{-1}$ | | $C_T$ $\mu mol.kg^{-1}$ | | $C_{ant}$ TrOCA $\mu mol.kg^{-1}$ | |
|---|---|---|---|---|---|---|---|---|---|---|---|---|---|---|---|---|
| **1978-2018** | -0.001 | ± 0.001 | 0.01 | ± 0.01 | -1.2 | ± 0.9 | 0.2 | ± 0.2 | **-0.8** | **± 0.4** | -0.1 | ± 0.1 | **2.0** | **± 0.5** | **1.4** | **± 0.5** |
| **1987-2018** | -0.001 | ± 0.001 | 0.01 | ± 0.01 | -1.9 | ± 1.4 | 0.3 | ± 0.4 | -0.3 | ± 0.5 | 0.6 | ± 0.1 | **1.6** | **± 0.5** | 1.1 | ± 0.8 |
| **1987-2004** | -0.003 | ± 0.002 | 0.01 | ± 0.01 | **-6.5** | **± 1.8** | 0.9 | ± 0.9 | 1.7 | ± 1.0 | **-1.9** | **± 1.1** | **1.8** | **± 0.4** | **5.2** | **± 1.1** |
| **2004-2018** | -0.006 | ± 0.003 | 0.01 | ± 0.01 | -1.8 | ± 4.5 | -0.5 | ± 1.0 | **-3.9** | **± 0.7** | 3.4 | ± 0.2 | 1.7 | ± 1.9 | -3.5 | ± 1.5 |

[Figure]

**Figure 4. (a) Full Θ-S diagram of studied water masses and (b) zoomed on bottom waters. Values are from literature for the WSBW (Fukamachi et al., 2010; van Heuven, 2013; Pardo et al., 2014; Robertson et al., 2002), the WSDW (Carmack and Foster, 1975; Fahrbach et al., 1994; van Heuven, 2013; Robertson et al., 2002), the RSBW (Fukamachi et al., 2010; Gordon et al., 2015; Johnson, 2008; Pardo et al., 2014), the ALBW (Fukamachi et al., 2010; Johnson, 2008; Pardo et al., 2014), the CDBW (Ohshima et al., 2013) and the LCDW (Lo Monaco et al., 2005a; Pardo et al., 2014; Smith and Treguer, 1994), and from the OISO-ST11 dataset for the OISO-ST11 LAABW and OISO-ST11 LCDW. Error bars are calculated from the individual annual averaged values for the OISO-ST11 LAABW and from all data for the OISO-ST11 LCDW. For the OISO-ST11 LAABW, the grey cross are the GEOSECS (lowest Θ) and INDIGO-1 (highest Θ) values.**

Table 3. Compilation of $C_{ant}$ sequestration investigations in the AABWs ($\gamma^n \geq 28.25$ kg.m$^{-3}$) using the TrOCA method. The $C_{ant}$ estimation of Pardo et al. (2014) is calculated using theoretical AABW mean composition (with 3% of ALBW) and the carbon data from the GLODAPv1 and CARINA databases. Sandrini et al. (2007) values has been measured at the bottom in the Ross Sea and correspond to recently sink high salinity shelfurface water (HSSW). The mean values published by Roden et al. (2016) for the AABWs present WSDW characteristics but can be a mix of CDBW and LCDW.

| Source | Location | Water masses considered | Year | $C_{ant}$ μmol.kg$^{-1}$ |
|---|---|---|---|---|
| Pardo et al. (2014) Fig. 5 | Averaged AABW composition | WSBW-RSBW-ALBW | 1994 | 12 |
| Lo Monaco et al. (2005b) Fig. 4b | WOCE line I6 (30° E; 50°-70° S) | WSBW CDBW | 1996 | 15 20 |
| Sandrini et al. (2007) Fig. 4a | Ross Sea | HSSW (previous RSBW) | 2002/2003 | Max. of 30 |
| Shadwick et al. (2014) Table 2 | Mertz polynya and Adelie depression | ALBW | 2007/2008 | 15 |
| Roden et al. (2016) Table 2 | South Indian ocean (30°-80° E; 60°-69° S) | WSDW-LCDW-CDBW | 2006 | 25 |
| van Heuven et al. (2011) Fig.13 | Weddell gyre (0° E; 55°-71°S) | WSBW | 2005 | 16 |
| This study | Enderby basin (56.5° S/63° E) | LAABW (mix of WSDW-CDBW-RSBW-ALBW) | 1978-1987 | 8 ± 3 |
|  |  |  | 1987-1998 | 10 ± 4 |
|  |  |  | 1987-2004 | 13 ± 4 |
|  |  |  | 1998-2004 | 14 ± 2 |
|  |  |  | 2010-2018 | 13 ± 1 |
|  |  |  | **1978-2018** | **12 ± 3** |

---

## Author Response (AR3)

**Comments Referee 1**

General comments: This manuscript deals with temporal variations of anthropogenic CO2 in bottom waters in the Southern Ocean. The Southern Ocean is said to take up 40% of anthropogenic CO2 absorbed by the ocean. Thus, investigations of temporal variability of anthropogenic CO2 are very important to evaluate ocean's capacity of absorbing atmospheric CO2, information of which is indispensable for the projection of global warming. In terms of oceanic observation, the Southern Ocean is one of the regions, where the number of measurements, especially for chemical and biological properties, is scare. In this point also, it is worth of being published in the journal. The manuscript is well organized, and is easy to read. The approaches used in the study are not new, but traditional ones. It is not a problem. It would be necessary to adopt an approach, which has been demonstrated to be useful for the detection of small signals of anthropogenic CO2 variations. The authors attempt also to relate the variations to those of AABW formation, although not clearly found. As a whole, it seems that the manuscript is worthy of publication in the journal, but after a moderate revision. A few major comments are stated in the followings, and the minor ones are stated in the specific comments.

**Response: we are thankful for the quick answer provided by the reviewer. The concerns of the reviewer have been answered here after and have been valuable help to upgrade the manuscript.**

In this paper, temporal variability of anthropogenic CO2 is examined using historical data collected at OISO. The data have been quality controlled by some data synthesis activities such as GLODAP. Nevertheless, I have a question on this point; the data syntheses have been done with a purpose of obtaining data consistency of a basin-scale. By contrast, the authors examine temporal variability of a local scale. In addition, data consistency is usually confirmed by data in deep layers of > 2000 m. This paper deals with data in deep layers. From these points, it is necessary to show that results obtained in the present study is not influenced by the data synthesis. Furthermore, for the recent data, quality control is made independently. Is there any possibility that the Cant stability is caused by the quality control? I recommend the authors to conduct quality-control on OISO data independently.

**Response: The reviewer is correct. For most of the ocean basins, data consistency is generally based on data in deep layers (> 1500 or 2000 m). However, because in the Southern Ocean anthropogenic $CO_2$ is also found at depth (> 3000 m), comparison is investigated in "old" deep waters, say around 2000-3000m (LCDW) where Cant (and DIC) should be relatively stable from one year to the next (within error of measurements, 1-3 $\mu mol.kg^{-1}$). Following the reviewer's recommendation, we propose to add a figure in Supplement Material (Fig. S1) showing the consistency of our dataset at the two OISO stations where samples were collected down to the bottom, the OISO-ST11 presented in the manuscript and the OISO-ST17 sampled in the Subtropical Zone (30° S-66° E). This figure shows a limited number of measurements that are out of the range of tolerance, but one has to keep in mind that interannual (or multiannual) variations may occur and this calls for great care before applying an adjustment.**

**Since 1987 (when the cruise INDIGO3 was performed), a shift in $A_T$ is suggested at high latitudes by the comparisons of INDIGO3 data (unadjusted, following the GLODAPv1 and CARINA recommendations) with other cruises data (adjusted, following the GLODAPv2 recommendations). This comparison shows differences that range between -4 $\mu mol.kg^{-1}$ and +10 $\mu mol.kg^{-1}$ (Fig. S2). Most of the crossovers that suggest a positive offset for INDIGO3 data (between +6 $\mu mol.kg^{-1}$ and +10 $\mu mol.kg^{-1}$) are found south of 60°S, suggesting that $A_T$ may have decreased in deep waters at high latitudes since 1987. This is why we first decided for no adjustment in the submitted manuscript (as in the GLODAPv1 and CARINA data products, whereas the INDIGO3 data in GLODAPv2 were**

**45**  **corrected by -8 µmol.kg$^{-1}$). However, at the OISO-ST11, A$_T$ data from the INDIGO3 cruise are also**
**46**  **about 8 µmol.kg$^{-1}$ higher than the mean value in deep waters (2000-3000m), in good agreement with**
**47**  **the other crossovers at high latitudes. In order to reduce the potential bias that could result from**
**48**  **either over-adjusting the data (GLODAPv2 recommendation) or not adjusting the data (GLODAPv1**
**49**  **and CARINA recommendations), and because most of the crossovers at mid-latitudes suggest a small**
**50**  **positive offset, we propose to apply an intermediate adjustment of -4 µmol.kg$^{-1}$ in the revised**
**51**  **manuscript (the impact on C$_{ant}$ is +2 µmol.kg$^{-1}$). The uncertainty regarding this adjustment will be**
**52**  **discussed in Supplement Material. Fig. 3 (before Fig. 4) presenting the interannual variability of**
**53**  **LAABW properties and Table 2 presenting the calculated trends will be adjusted correspondingly.**

**54**  **Figure S1 also shows that the low A$_T$ values between late 1998 and 2004 are found both in the**
**55**  **Antarctic zone and the Subtropical zone. This is surprising, but there are no reason to believe that**
**56**  **the data are biased since CMRs were used for all OISO cruises, and the instrument and data**
**57**  **processing were the same during the first OISO cruise in January/February 1998 (showing A$_T$ values**
**58**  **close to the mean in Fig. S1) and the following cruises.**

**59**  In discussion, the authors attempt to relate variations of anthropogenic CO2 in AABW to changes in
**60**  AABW formation region. It is well discussed, but information of water mass age of AABW is lacking. It
**61**  is necessary to show that linkages between variations of AABW formation region and observed AABW
**62**  signals at OISO are appropriate in terms of water mass age. O2 and AOU are used simultaneously. I
**63**  think, it is enough for one of which, probably AOU.

**64**  **Response: we are sorry that there is no measurement related to water mass age in the available**
**65**  **data (I.e. no CFCs measured during OISO cruises), other than O$_2$ which is too sensitive to biological**
**66**  **activity to be used as a water mass age tracer. We agree that the mention of both O$_2$ and AOU is**
**67**  **unnecessary. This is a point also noticed by Reviewer 2. Because we are most discussing the O$_2$**
**68**  **concentration in the manuscript, we suggest to only present O$_2$ in Figure 3.**

**69**  Specific comments:

**70**  Line 18 (here around 460): "from about +7 µmol kg-1", increase from what?

**71**  **Response: We guess that what confused the two referees is the positive sign. We will delete the**
**72**  **positive sign and rephrase as follows: 'from the average concentration of 7 µmol.kg$^{-1}$ calculated for**
**73**  **the period 1978-1987 to the averaged concentration of 13 µmol.kg$^{-1}$ in the period 2010-2018.'**
**74**
**75**  Line 20 (here around 463): "CT", this is the first appearance in the abstract. Write it in full.

**76**  **Response: this will be added.**
**77**
**78**  Line 23 (here around 467): "$\theta$, S", they are the first appearance in the abstract. Write them in full.

**79**  **Response: this will be added.**
**80**
**81**  Lines 90-91 (here around 535-536): "station 430", depth?

**82**  **Response: the depth (4710 m) will be added.**

**83**  Line 91 (here around 535): "405 km and 465 km", away from where?

**84**  **Response: These are the distance away from the OISO-ST11 sampling site. This will be rephrased as**
**85**  **"located near the OISO-ST11 sampling site (405 km and 465 km away from it, respectively)"**
**86**
**87**  Line 109 (here around 556): "the PET sector", is it usually used? I do not understand where it is.

**Response: A short sentence will be added to the text, as well as the references mentioned here after**
**to clarify the use of this name. The PET, Princess Elizabeth Though, is also referred as the Balleny**
**Though in Orsi et al. (1999), even if more currently mentioned as PET. It corresponds to the ocean**
**section separating the Kerguelen Plateau from the Antarctic continent. Its deepest point is 3750 m,**
**deep enough to allow AABWs to flow between the Australian Antarctic Basin and the Enderby Basin**
**(Heywood et al., 1999). The work of Heywood et al. (1999; Fig. 1) revealed that in the northern part**
**of the PET the AABW flow from west to east, while in the southern part the flow is from east to west.**
Line 150 (here around 612): "AT", Probably this is the first appearance. Spell out here.

**Response: this will be added.**

Line 160 (here around 628): "$\theta$ and S", spell out here. ´

**Response: this will be added.**

Lines 163-165 (here around 633-635): according the description, it seems that the figures are not
accuracy but repeatability.

**Response: The referee is correct. The accuracy is given by the analysis of CRMs. This will be corrected.**

Line 236 (here around 714): "January", which year. In this paper, all the data are analyzed assuming
that seasonal variations in deep waters are negligible (lines 154-156). It is not appropriate to refer to
months.

**Response: the authors agree with the reviewer. This will be adjusted by mentioning the early and**
**late 1998 sampling.**

Line 276 (here around 762): "underlying", do you mean a water mass below AABW?

**Response: this is a mistake, we meant overlying the AABW (referring to LCDW). This will be corrected**
**by using 'LCDW' instead.**

**Comments Referee 2**

General comments

The study presents results from a time series in the Indian sector of the Southern Ocean, which together with historical relevant data span a 40-year period. Using this time series, the authors evaluate the evolution of anthropogenic $CO_2$ (Cant) in the Antarctic Bottom Waters (AABW). It is an interesting and generally well written work, and generally good figures and tables. There are some need for clarity in some parts and there is some concern of the treatment of data gaps, but most of this should be rather easily dealt with, and I recommend publication after minor revision. A detailed list of comments follows below.

**The authors are thankful for the fast answer and the positive interest given to the manuscript, as well as for the numerous valuable comments.**

My main comments are related to the definition and subsequent presentation of AABW, and, the data gap between 1987 and 1998 and how this is handled and presented. To start with the definition of AABW, this is not an issue in itself, since the denser definition has been used before, and also, since almost any definition can be accepted as long as it is clearly presented. The latter is the problem here, at least for someone not as familiar with the area and these water masses (I usually work in the high-northern latitudes). The definition and choice is clearly described in 2.3, but, then the reader is referred to Fig. 3, where AABW is noted in the layer above the focus of this study, while the data evaluated is in the layer annotated "Considered data". When then the results of the property evolution of AABW are further presented in Fig. 4, at least I got somewhat confused. Whether this is only me or not, this may call for some added clarity. I would suggest to annotate your AABW layer (hence at neutral density >28.35) as AABW (or AABW* or similar), to make this clear, and then make a distinction with the more common AABW.

**Response: the authors understand the concern of the reviewer. To solve this potential confusion, we suggest labelling the AABW as define in our manuscript (neutral density >28.35 kg.m$^{-3}$) Lower Antarctic Bottom Water (LAABW).**

Nevertheless, this mostly refers to Fig. 3, and I have several concerns with this figure, as detailed below. Hovmöller plot is a wonderful thing, and can be very illustrative. However, it can also be deceiving, especially when there are gaps in the data, and the gridding is allowed to interpolate over these gaps, which often can create features that give a false picture of actual evolution. Fig. 3 suffers from this when plotting the older data (1978–1987) together with the OISO time-series data starting from 1998. There are several peculiar features in Fig. 3, especially for Cant and AT. The fact that most of the other plotted parameters show overall stable layer properties, over the full period, may seem to reduce this concern, but I am not convinced. In addition, I'm not fully convinced about the benefit of showing depths from 1500 m, when almost all results and discussion are concerned with the layer below 4000 m. Even more so when the upper layers seems to show most of the strange features, for example the minimum in Cant in the older data (which may in part show the issue with the TrOCA method, with even negative concentrations, which are not realistic, in the most upper part of the deep waters).

**Response: The authors agree that the figure needs to be upgraded, clarified and simplified. The suggestions of the referee have been taken into account by redrawing the Fig. 3 (now Fig. 2) using only the OISO data (from 1998 to 2018). The extrapolations were very misleading indeed, so the figure is now drawn with weighted-average gridding (and limited extrapolation around the data point). The aim of this figure is to show the differences in AABW and LCDW characteristics before**

**focusing on the variability and trends observed in the bottom layer (it also shows that the neutral**
**density 28.35 is a better definition for a more homogeneous bottom layer that we now define as**
**LAABW). In addition, the control quality of the data is performed in the old deep waters (well**
**characterized in the figure by the maximum in $C_T$). Following the recommendation from the other**
**Referee, we propose to add a figure in Supplement Material (Fig. S1) showing the consistency of our**
**dataset at the two OISO stations where samples were collected down to the bottom, the OISO-ST11**
**presented in the manuscript and the OISO-ST17 sampled in the Subtropical Zone (30° S-66° E). This**
**figure shows a limited number of measurements that are out of the range of tolerance, but one has**
**to keep in mind that interannual (or multiannual) variations may occur and this calls for great care**
**before applying an adjustment.**

**Since 1987 (when the cruise INDIGO3 was performed), a shift in AT is suggested at high latitudes by**
**the comparisons of INDIGO3 data (unadjusted, following the GLODAPv1 and CARINA**
**recommendations) with other cruises data (adjusted, following the GLODAPv2 recommendations).**
**This comparison shows differences that range between -4 μmol.kg$^{-1}$ and +10 μmol.kg$^{-1}$ (Fig. S2). Most**
**of the crossovers that suggest a positive offset for INDIGO3 data (between +6 μmol.kg$^{-1}$ and +10**
**μmol.kg$^{-1}$) are found south of 60°S, suggesting that $A_T$ may have decreased in deep waters at high**
**latitudes since 1987. This is why we first decided for no adjustment in the submitted manuscript (as**
**in the GLODAPv1 and CARINA data products, whereas the INDIGO3 data in GLODAPv2 were**
**corrected by -8 μmol.kg$^{-1}$). However, at the OISO-ST11, $A_T$ data from the INDIGO3 cruise are also**
**about 8 μmol.kg$^{-1}$ higher than the mean value in deep waters (2000-3000m), in good agreement with**
**the other crossovers at high latitudes. In order to reduce the potential bias that could result from**
**either over-adjusting the data (GLODAPv2 recommendation) or not adjusting the data (GLODAPv1**
**and CARINA recommendation), and because most of the crossovers at mid-latitudes suggest a small**
**positive offset, we propose to apply an intermediate adjustment of -4 μmol.kg$^{-1}$ in the revised**
**manuscript (the impact on Cant is +2 μmol.kg$^{-1}$). The Fig. 3 (before Fig. 4) presenting the interannual**
**variability of the LAABW properties and the Table 2 presenting the calculated trends will be adjusted**
**correspondingly. Fig. S2 will be completed by the list of the cruises presented.**

**The Figure S1 also shows that the low $A_T$ values between late 1998 and 2004 are found both in the**
**Antarctic zone and the Subtropical zone. This is surprising, but there are no reason to believe that**
**the data are biased since CMRs were used for all OISO cruises, and the instrument and data**
**processing were the same during the first OISO cruise in January/February 1998 (showing $A_T$ values**
**close to the mean in Fig. S1) and the following cruises.**

The interpolation of this minimum patch leads to unfortunate wordings in the results, such as on line
236, with "a sudden increase. . . between January and December 1998" seems to refer to the low
values calculated for the 1987 data and the clearly higher concentrations calculated for the OISO data.
(I also don't really understand the "between Jan and Dec 1998" part, since the first OISO data were
sampled in Feb 1998, and the next in Dec the same year.) Apparently there are some need for
clarifications here, but also to be cautious when interpreting interpolated values over large gaps. One
way to solve this is of course to exclude the older data from the Hovmöller plots. These can still be
used in the comparison/evaluation, and included in Fig. 4.

**Response: the reviewer is right about the issue for the 1998 samplings mentioned (same as Reviewer**
**1). This is because the first OISO cruise started in January 1998, but the station 11 was actually**
**sampled in the beginning of February as mentioned in Table 1. This will be corrected. We also agree**
**that extrapolation can be misleading and we thank the Reviewer for pointing this issue. Having**
**removed the GEOSECS and INDIGO data from the Hovmöller plots (Fig 2., before was Fig. 3), the**
**extrapolation is no more an issue for interpreting the signal observed for the first OISO cruises, but**

**the increase in Cant between February 1998 and December 1998 remains (from < 6 μmol.kg$^{-1}$ to**
**about 10 μmol.kg$^{-1}$).**
To continue on this figure (Fig. 3), for the bottom layer, the fact that it is stretched below the deepest
samples seems to create at least the distinct maximum in mid-2000s. Perhaps this will be reduced if
the maximum depth/pressure is set to the deepest sample, to exclude extrapolations below that
depth.
**Response: Having removed the INDIGO1 data from the Hovmöller plots, this no more an issue**
**because the deepest sample is collected at the same depth for all cruises**
Specific comments
L18 (here around 460): Do the changes here (+7 and +13, respectively) refer to the whole period?
Please clarify.
**Response: these are not changes, but Cant concentrations. The following rephrasing is suggested:**
**'from the average concentration of 7 μmol.kg$^{-1}$ calculated for the period 1978-1987 to the average**
**concentration of 13 μmol.kg$^{-1}$ for the period 2010-2018.'**
L23 (here around 467): A rather tiny remark, but the use of "pluriannual" may be grammatically correct
(I'm not a native English speaker), but consider using "multiannual" (or multi-annual), which are more
common (I believe). The same is used on L360.
**Response: We agree that this is maybe not the best word to use. It will be replaced by 'multi-annual'.**
L59 (here around 502): I'm expecting a reference in the end of this sentence. This may be refer to the
reference in the previous line, but you may consider moving this to the end.
**Response: We agree that the reference is misplaced. It will be moved to the end of the sentence.**
L95 (here around 541): I can't find a definition of "AAC" anywhere. Please write out and define the first
time.
**Response: We agree that the definition of ACC is missing (Antarctic Circumpolar Current). That will**
**be corrected.**
L96-97 (here around 542-543): Unclear sentence. Need some rephrasing/re-writing. Suggestion: ". .
.Weddell Sea, where deep and bottom waters are produced. . .".
**Response: The sentence will be rephrased as suggested.**
L98-100 (here around 544-546): In the same sentence, there are several instances where the full water
mass name is not spelled out, for example "the Ross Sea (RSBW; . . .". This may be intuitive, but I don't
think the full names of some of these are written out at any place in the manuscript so would suggest
to consider doing that at some place.
**Response: The full names will be added explicitly.**
L100 (here around 546): Rephrase: In the Prytz Bay, AABW formation has also. . . This sentence is
overall quite unclear, especially the last part, so please consider rewriting for clarification.
**Response: It is indeed quite unclear. We propose the following rewriting : 'AABW formation has also**
**been observed in the Prydz Bay (Rodehacke et al., 2007; Yabuki et al., 2006). There, three polynyas**

**257** **and two ice shelves have been identified as Prydz Bay Bottom Water (PBBW) production hotspots**
**258** **from seal tagging data (Williams et al., 2016). This PPBW flows out the Prydz Bay through the Prydz**
**259** **Channel and get mixed with the CDBW.'**
**260**
**261** L105 (here around 552): The "Warm Deep Water" is not described, so not easy to follow without a
**262** previous knowledge of the area and the present water masses. Please clarify.

**263** **Response: we agree that it may be difficult to follow. The Warm Deep Water is slightly modified**
**264** **Circumpolar Deep Water (by mixing with surface waters when it enters the Weddell Basin). For**
**265** **simplification, we suggest rewriting as follows: The exported WSDW originates from the**
**266** **Circumpolar Deep Water (CDW) that enters the Weddell basin and mixes with WSBW and High**
**267** **Salinity Surface Water (HSSW) (see Fig.2 in van Heuven et al., 2011).**
**268**
**269** Section 2.4: Part of this section, and in particular from L133, deals with results of Cant from the
**270** methods not yet described. I would suggest to move this to the Result section, at least the Cant parts,
**271** or maybe part of the Discussion.

**272** **Response: the authors agree that this section does not fit in the material and method part of the**
**273** **manuscript, but rather in the discussion section as suggested.**
**274**
**275** L152 (here around 614): Since the "P" in GLODAP refers to "Project", the "project" after should be
**276** avoided (I think). You could rephrase this into something like: . . ..not yet qualified (or included in) the
**277** most recent GLODAPv2 product.

**278** **Response: the mention of GLODAP will be rephrased as suggested.**
**279**
**280** L161 (here around 629): The stated accuracy for temperature and salinity seems too low. The standard
**281** CTD accuracy, for example found at the GO-SHIP home page (Hydro-manual) is 0.002 for both. Please
**282** check.

**283** **Response: the authors agree and will correct the accuracy for temperature (0.002°C) and salinity**
**284** **(0.005 for measurements using a salinometer).**
**285**
**286** L161 (here around 629): As far as I can see, this is the first time "AT" is mentioned, but not defined.
**287** Please add this.

**288** L166 (here around 636): Same for "O2" as for AT above. Please define first time.

**289** **Response: AT and O2 will be defined here.**
**290**
**291** L170 (here around 641): You mean "onshore"?

**292** **Response: the reviewer is right about this mistake.**
**293**
**294** L184 (here around 656): Clarify which "Redfield ratio". You mean the C:O ratio? Please add this.

**295** **Response: Initially, only $C/O_2$ and $N/O_2$ ratios were involved in the definition of the parameter 'a'**
**296** **(Touratier and Goyet, 2004b; Lo Monaco et al., 2005b). In the latest definition of the method**
**297** **Touratier et al. (2007) presents an upgraded definition of this parameter by combining the Redfield**
**298** **equation coefficients for $CO_2$, $O_2$, $HPO_4^{2-}$ and $H^+$ and the same rules of construction as Broecker (1974)**
**299** **did for tracers NO or PO. Because we want to keep the explanation simple in the manuscript, we**
**300** **suggest to rephrase L184 as follows : 'where a is defined in Touratier et al. (2007) as combination of**

the Redfield equation coefficients for $CO_2$, $O_2$, $HPO_4^{2-}$ and $H^+$. For more details about the definition
and the calibration of this parameter, please refer to Touratier et al. (2007).'

L217 (here around 691): Either remove "after", so it reads ". . . and only impacted by. . .", or if more
correct, add "subduction", so it reads "and after subduction only impacted by. . .".

**Response: the word 'subduction' will be added as suggested.**
L233: "LCBW" is here mentioned for the first time, without definition or any description anywhere in
the manuscript, as far as I can see. Please add this.

**Response: LCDW refers to the Lower Circumpolar Deep Water laying above AABW in the entire**
**Southern Ocean. Details about this water mass will be added in Section 2.2 where it is first**
**mentioned.**
L235-236: This is what was commented on in the generall comments above, with the "sudden
increase". Please revise and clarify. It is more likely that there was a more gradual evolution, and none
of the other parameters calls for any sudden changes. Also, the data quality and methods between the
older data and the OISO data may differ, so extra causion is taken when comparing them.

**Response: we removed the older data form the Hovmoller plot, but the change in $C_{ant}$ in LCDW**
**remains (from <6 µmol.kg$^{-1}$ in Feb 1998 (similar as for the older data) to about 10 µmol.kg$^{-1}$ for the**
**following cruises).**
L240 (here around 719): The maximum in Cant in 2004 is one occasion, and followed by five (almost
six) years without any data. I would be cautious to over interpret this. However, it co-incides with a
maximum in oxygen, which could indicate a ventilation event.

**Response: we agree with the referee about being cautious with the measurements in 2004. Indeed**
**the maximum in $C_{ant}$ is due to the maximum in $O_2$ (not associated with a maximum in $C_T$).**
L256-260 (here around 739-743): The lower concentrations of AT in the years around 2000 at all depths
below (at least) 1500 m (have you checked the whole water column?) seems a bit odd. Especially when
this is not seen in any of the other parameters. Also, when comparing two years in the 1980s with data
more than a decade later, one should be extra cautious in the interpretation, not the least when the
two years/occasions in 1985/87 show the highest concentrations seen over the evaluated period.
Certainly the years after 2000 show much lower concentrations, which may be a phase due to a change
in different forcing, but to suggest reduced calcification from only a few years/occupations of data is
very speculative, and clearly something that change a few years later.

**Response: As mentioned in the general comments, the low $A_T$ values between late 1998 and 2004**
**are found both in the Antarctic zone and the Subtropical zone (Figure S1), but they are not observed**
**in the surface layer (this will be added in the revised manuscript). The hypothesis about reduced**
**calcification could explain this contrast between the surface waters and the deep ocean.**

L259-260 (here around 742-743): Is it realistic that the increase in CT is lower than the accumulation
of Cant?

**Response: The small increase in $C_T$ over the period 1987-2004 could be caused by a reduction in $C_{T,nat}$**
**around the year 2000 (associated with the low $A_T$ values). This said, we also have to keep in mind**
**the uncertainty on the $C_{ant}$ calculations. This will be clarified in the results and in the discussion.**

L261 (here around 744): While there is a rather clear trend in oxygen during this period – although I
would be careful in talking about trends over such short periods, especially when comparing to a year
with a maximum (2004) – there is no trend in Cant. Instead the latter shows some clear interannual
variability. Also, the "trend" in temperature is indeed very small, and even if not significant, the change,
or better, variability, in salinity is rather large. Consider these points when revising this part. Your
statement on L267-268 highlights this issue.

**Response: we agree with the reviewer that there is no clear trend in Cant over 2004-2018. We will**
**change "decrease in $C_{ant}$" for "no increase in $C_{ant}$". The same is true for temperature and salinity.**

L270-271 (here around 755): There is also a maximum in temperature in 1985, so this could indicate
more mixing with WSDW, which are both fresher and warmer.

**Response: we agree with the reviewer that more mixing with WSDW (or CDBW) could also explain**
**the higher $C_{ant}$ concentrations and lower S in 1985 (the signal in temperature is not well marked due**
**to the large error bars). This will be added in the text.**
L275-278 (here around 761-764): This is a very long sentence. I suggest to divide it, with period after
". . .the underlying deep waters." Then remove "and", and start on "Since", or change the start of the
sentence. For the last part of this sentence (L277-278), the suggestion of increased contribution from
the Ross Sea is not clear to me since the oxygen decrease, while the salinity goes up and down. Or are
you only referring to the one occupation in 2012? (If this is the case, it seems to detailed to explain a
single year taken out of a long time series.)

**Response: the suggestion made by the reviewer to shorten the sentence will be used. Our aim is to**
**discuss the variability in Cant concentrations that could reflect variations in the contribution of**
**different types of AABWs. We suggest that the lower Cant concentrations observed in 2011, 2012**
**and 2013 may be due to an increased contribution of older types of AABW. We agree that pointing**
**to RSBW as a possible candidate because salinity was higher in 2012 is too speculative. This will be**
**removed.**
L280 (here around 767): The stated freshening of 0.01, for which period is that observed? Please clarify.

**Response: The sentence will be corrected as follows: 'The freshening in S of -0.006 decade$^{-1}$ between**
**2004 and 2018 that we observed on the Western side of the Kerguelen Plateau was also observed**
**on the Eastern side of the Plateau by Menezes et al. (2017) over a similar period.'**

L312-313 (here around 823): ". . .(15 umol kg-1) due to mixing with older CDW."

**Response: the sentence will be corrected.**

L317 (here around 827): "that contain very high amounts of Cant . . ."

**Response: the sentence will be correct as suggested.**

L318-320 (here around 829-830): The last sentence of this paragraph basically repeats what have been
said above. Consider to remove.

**Response: the authors agree with the reviewer and will remove this sentence.**

L325 (here around 835): Here you write out "Southern Ocean" after having used the abbreviation
throughout the manuscript, even the sentence before. Consider to revise.

**Response: "Southern Ocean" will be changed to SO.**

L340 (here around 845): "evaluated" should here instead be "estimated", or "calculated", or "found"
(I think).

**Response: "evaluated" will be replaced by "calculated".**

L386 (here around 898): Consider rewording ". . .vary in a very large range. . .". Suggestion: "show a
very large variability", or maybe, "vary over a very large range".

**Response: the rewording 'vary over a very large range' will be used.**

L387-388 (here around 901): "(−221 mmol C m-2 d-1; Roden et al., 2016).

**Response: we will correct this according to the reviewer suggestion.**

L416 (here around 928): Both these water masses (RSBW and ALBW) have higher salinity, and while
oxygen show a reduced trend the salinity goes up and down, so this explanation does not hold for all
years during this period.

**Response: we understand the concern of the reviewer. The mention of the WSDW will be added, as
for the response of the comment L275-278.**

L424 (here around 937): "explains most, but not all, of the observed. . ."

**Response: the sentence will be corrected.**

L463 (here around 977): GLODAPv2 version are written as "GLODAPv2.2021 (.2020 is soon to be
released). You do mean 2021 and not 2020?

**Response: the data will not be included in GLODAP in the 2020 version, but in the following one.**
L851-853 (here around 1353-1354): Table 2 (and in general): You may want to consider if you want to
keep AOU as parameter, when you mostly refer to oxygen. The trends are almost exactly thesame (but
opposite of course), and gives the same message.

**Response: we agree with the reviewer. AOU will be removed from Table 2 and from Figure 2 (and
from the corresponding parts in the text).**
Technical comments

L22: This is, however, modulated. . .

**Response: the comas will be added.**

L35: The references should, typically, be chronologically ordered. Please check throughout the
manuscript. (There are more examples of this, but I won't comment on this more.)

**Response: we agree with the referee. We will check for other occurrences.**

L71: This is, however, not the. . .

**Response: the comas will be added.**

L91: ". . .(405 and 465 km, respectively)."

**Response: the coma will be added.**

L107-113: Examplified with ". . .East of the Kerguelen. . .", this section has many of these
"directions/locations" (east/west/. . .) spelled with a large letter, even not part of a name. I think this
is not correct, and if so, please change.

**Response: this will be corrected.**

L118-119: . . .28.27-bottom, respectively. . .

**Response: the coma will be added.**

L172-173: Change font; the part of the sentence from "for deep samples. . ." are in a different font
(maybe "Cambria").

**Response: the font will be changed.**

L220: Change font for "value for".

**Response: the font will be changed.**

L306: Add a comma: "2018 (Fig 3a), probably . . ."

**Response: the coma will be added.**

L340: Add a ".": Pardo et al. (2017)

**Response: the dot will be added.**

L347: For consistency, change "South-Western" to "South-western" (similar as on L325).

**Response: this will be corrected.**

L449: Remove "." for consistency: (e.g. Frölicher et al., 2014).

**Response: the coma will be deleted.**

L451: References in chronological order.

**Response: we agree with the referee. This will be corrected.**

**Answer to the Topic Editor comments:**

Comments to the Author:

Dear Dr. Mahieu and co-authors,

Thank you for the revised submission. I am generally satisfied with the changes you made following the comments by the referees. Going through the manuscript myself, I have listed my comments below. Please prepare the final version of your manuscript taking into account these comments.

**Response: Dear Dr. Hoppema, we are thankful for your comments. Please find hereafter our responses.**

In the title: Antarctic Bottom Water (without –s) as this study measured only one type at one location.

**Response: we wanted to insist on the mix of AABW from different sources by writing it this way. This will be corrected as suggested.**

Section 2.1 is clearly part of the methods and should thus be moved to Section 3.

**Response: this section will be moved as suggested.**

I suggest to call the water mass defined here as Lower AABW, not as Low AABW. Just like other well-known water masses like Lower CDW, etc.

**Response: this will be corrected as suggested.**

Please place all 2 in CO2 in subscript.

**Response: this will be corrected in the title and the references.**

Please check the references because many are incomplete.

**Response: this will be done. DOIs will be updated and page numbers checked. The last references without page numbers do not mention any online.**

As to the data used in this study, there are several more OISO cruises (as also in the GLODAP tables). Please provide the arguments for including the cruises that the authors did, while excluding others.

**Response: the missing OISO cruises in this study correspond to the cruises when this station was not re-occupied. To clarify this, the following sentence will be added to section AABW sampling: 'In our analysis, we included all the data available for the OISO-ST11 location (which has not been sampled during each cruise for logistical reasons).'**

As to the supplement Table S1 (and discussion in the main text) with the adjustments from the different quality control efforts, it is shown that AT at the OISO cruises did not receive any adjustments. However, this is not the complete story. The GLODAP table says that there is not sufficient data for comparison in this region, upon which the OISO data did not get an adjustment because this could not be argued safely. This is actually the same as getting no quality control. This should be made clear in the manuscript.

**Response: we agree and will clarify this point as follows:**

'… this calls for great care before applying an adjustment. This is the case for $A_T$ data that did not
get an adjustment in GLODAP because this could not be argued safely due to the limited number of
data in this region.'.

L5 Shouldn't the University of Liverpool be mentioned?

**Response: this is missing indeed and will be added.**

L9 Antarctic bottom water (AABW) is known …

**Response: this will be corrected.**

L9 … but the sink is hardly quantified …

**Response: this will be added.**

L13 in the framework of …

**Response: this will be corrected.**

L16 At this location, the main sources of AABW are the low-saline … (fresh is not the word here,
because this is a saline water mass; I suggest to skip "younger" because: younger against what?)

**Response: we understand your concern and will correct this sentence as suggested.**

L20 SO has not been defined before

**Response: this will be added.**

L24 hydrographic (not: hydrological)

**Response: this will be corrected.**

L27 AABW

**Response: this will be corrected.**

L27-28 This sentence is trivial, and if not followed by which of these processes are important or how
they function, not necessary/useful.

**Response: we agree and will remove this sentence.**

L43 3% is more like the maximum. Mostly Cant is much less. I suggest to write here: less than 3%

**Response: we agree and will correct this.**

L53-55 "Thus, there is a need to better explore the CT and Cant temporal variability in the deep
ocean, especially in the SO where observations are relatively sparse." I cannot understand the
connection of this concluding sentence with the previous text in this paragraph. Please modify.

**Response: we agree that the sentence has no clear link with the previous statements. We suggest
to remove it.**

L56 AABW (without –s) Please change this throughout the manuscript.

**Response: this will be corrected.**

L58 … by covering a major part of the world ocean floor …

**Response: this will be corrected.**

L84 Study area

**Response: this will be corrected.**

L86 framework

**Response: this will be corrected.**

L98 is dominated by (instead of: is mainly governed)

**Response: this will be corrected.**

L111 … Lower Circumpolar …

**Response: this will be corrected.**

L112 I think HSSW is generally the abbreviation for High Salinity Shelf Water

**Response: this is correct, the abbreviation will be removed.**

L116-117 The PE deepest point of the PET is 3750 m, …

**Response: this will be corrected as 'The deepest point of the PET is 3750 m…'**

L158-160 "The accuracy of CT and AT measurements was ensured by daily analyses of Certified
Reference Materials (CRMs) provided by A.G. Dickson laboratory (Scripps Institute of
Oceanography)." This is indeed important to warrant the accuracy. For the interpretation it is also
important to know the accuracy. Please give the accuracy here.

**Response: A single accuracy value for all cruises is difficult to specify. Although we used the same**
**technic (and data processing) accuracy range between around 1.5 and 3 µmol/kg for both AT and**
**CT depending on the cruise. A complete list of CRMS batch number used during OISO cruise is**
**available at NCEI/OCADS with information on duplicates for each cruise**
**(https://www.nodc.noaa.gov/ocads/oceans/VOS_Program/OISO.html). As this information is**
**available at NCEI/OCADS (and the link recall in the section "Data Availability"), we think it was not**
**appropriate to list all CRM batch values for each cruise in the manuscript. We suggest to correct as**
**follows: 'The accuracy of $C_T$ and $A_T$ measurements (always better than ±3 µmol.kg$^{-1}$ for all cruises**
**since 1998) was ensured…'**

L164 silicate (no capital)

**Response: this will be corrected.**

L171 using (instead of: considering)

**Response: this will be corrected.**

L171 I do not understand why the value of 33 umol/kg was used, as the mean value from GLODAPv2
is 32.4 umol/kg. Even if the error because of this is small, it does increase it for no good reason.

**Response: this is correct. The value has been changed to 32.4 µmol.kg$^{-1}$. The change on the $C_{ant}$**
**values calculated with C° is -0.3 µmol.kg$^{-1}$.**

L187-188 from deep waters free of anthropogenic CO2 …

**Response: this will be corrected.**

L245-246 "the theoretical Cant trend at the AABW formation sites would be of the order of +8
µmol.kg-1.decade-1." How was this calculated? Only part of the AABW, when it is formed, contains
water that has been at the surface. Only that part could follow the atmospheric increase on CO2.
What percentage of surface water was assumed as contributing to AABW?

**Response: This value was listed to give a taste of the theoretical $C_T$ increase in Antarctic surface**
**waters assuming that ocean $fCO_2$ follows the atmospheric $CO_2$ increase. In the Prydz Bay region**
**Roden et al (2016) concluded that "surface waters in the seasonal ice zone track the atmospheric**
**increase in $fCO_2$". For our calculation we used the mean properties in Antarctic surface waters**
**observed in the Prydz Bay by Roden et al. (2016): SST=-1°C, SSS= 34.2, $A_T$=2291 µmol/kg and $fCO_2$=**
**376 µatm in 2006. Assuming that oceanic $fCO_2$ increased at a rate of 1.8 µatm we calculated $C_T$ and**
**we derived a trend in $C_T$ of +8 µmol/kg/decade in the Antarctic surface water (assuming no change**
**in temperature, salinity and alkalinity). Note that this value is close to the theoretical trend in $C_T$**
**calculated by Van Heuven et al. (2014) in the Weddell Sea (about +0.8 µmol/kg/yr, the red circle in**
**Figure 4a in Van Heuven et al., 2014). We suggest to revise following: "Due to the mixing of AABW**
**with old CDW (Cant free), these trends are lower than the theoretical trend expected from the**
**increase in atmospheric $CO_2$. Indeed, assuming that the surface ocean $fCO_2$ follows the**
**atmospheric growth rate (+1.8 µatm.year$^{-1}$ over 1978-2018) in the seasonal ice zone (Roden et al.,**
**2016), the theoretical $C_{ant}$ trend at the AABW formation sites would be of the order of +8 µmol.kg$^{-}$**
**$^1$.decade$^{-1}$ in the Antarctic surface water. This is close to the theoretical $C_T$ trend estimated for**
**freezing shelf water in the Weddell Sea (Van Heuven et al 2014)."**

L288 experiences

**Response: this will be corrected.**

L310 and ends … (instead of: and lasts)

**Response: this will be corrected.**

L315 … in the 1980s in the Indian sector of the Southern Ocean …

**Response: this will be corrected.**

L316 quality control (instead of: qualification)

**Response: this will be corrected.**

L427 "recognized freshening of AABWs over the last decades (Rintoul, 2007)." With a reference from
2007, this is not about the last decades. Please change wording or give a different reference.

**Response: we agree that there is a lack of consistency between the sentence and reference. We**
**suggest to change the sentence as follow: 'recognized freshening of the AABW (Rintoul, 2007;**
**Anilkumar et al., 2015).'**

L484 change to: GLODAPv2.2021

**Response: this will be corrected.**

L504 Please add info on what kind of this reference is and possibly where it can be found online.

**Response: we suggest to replace the current reference by the following:**
**Coverly, S. C., Aminot, A., and R. Kérouel, 2009. Nutrients in Seawater Using Segmented Flow**
**Analysis, In Practical Guidelines for the Analysis of Seawater, Ed. Oliver Wurl, CRC Press, June 2009,**
**doi: 10.1201/9781420073072.ch8.**

L506 Cycles (also in other cases where this journal is concerned)

**Response: this will be corrected.**

L538 pages: 205-206

**Response: this will be corrected.**

L551 pCO2

**Response: this will be corrected.**

L602 should be cited as: 18, GB1042, doi:10.1029/2002GB002017
L606 should be cited differently, similar as above
L624 pages: 346-349
L636 pages: 1221-1224
In many cases the references are incomplete, for example missing page numbers. Please go through
the references and correct them.

**Response: all the references will be checked and updated. The DOIs will be updated, and the page**
**numbers checked. The last references without page numbers do not mention any online.**

Figure 1 Please add that these are very rough transport paths. The dashed line for the ACC gives the
position, says the caption. What position? The ACC is wide; please explain. The path of the AABW in
the Weddell Sea is not correct. Neither is the path of the AABW from Prydz Bay and Cape Darnley,
which flows along the coast to the west and enters the Weddell circulation.

**Response: the mention will be added and the figure updated.**

Figure 2 The term is Hovmöller diagram.

**Response: this will be corrected.**

Thank you and best wishes

Mario Hoppema

[revised manuscript text omitted]